# Coupling enzymatic activity and gating in an ancient TRPM chanzyme and its molecular evolution

Yihe Huang[1,2], Sushant Kumar ●[1], Junuk Lee[1], Wei Lü ●[1]✉ & Juan Du ●[1]✉

Channel enzymes represent a class of ion channels with enzymatic activity directly or indirectly linked to their channel function. We investigated a TRPM2 chanzyme from choanoflagellates that integrates two seemingly incompatible functions into a single peptide: a channel module activated by ADP-ribose with high open probability and an enzyme module (NUDT9-H domain) consuming ADP-ribose at a remarkably slow rate. Using time-resolved cryogenic-electron microscopy, we captured a complete series of structural snapshots of gating and catalytic cycles, revealing the coupling mechanism between channel gating and enzymatic activity. The slow kinetics of the NUDT9-H enzyme module confers a self-regulatory mechanism: ADPR binding triggers NUDT9-H tetramerization, promoting channel opening, while subsequent hydrolysis reduces local ADPR, inducing channel closure. We further demonstrated how the NUDT9-H domain has evolved from a structurally semi-independent ADP-ribose hydrolase module in early species to a fully integrated component of a gating ring essential for channel activation in advanced species.

Channel enzymes (chanzyme) form a unique group of ion channels that unite two key biological functions—ion permeation and enzymatic activity—into a single peptide. The best-known chanzyme is cystic fibrosis transmembrane conductance regulator (CFTR), which is the only ATP-binding cassette transporter known to be a chloride channel[1,2]. The binding of ATP to the nucleotide-binding domain in CFTR triggers channel opening[3], while ATP hydrolysis catalyzed by the nucleotide-binding domain followed by product release leads to channel closure[4]. Therefore, the catalytic cycle is strictly coupled to the gating cycle in CFTR, as agonist binding and agonist digestion take place in the same active site.

A second group of chanzymes are members of the transient receptor potential cation channel melastatin (TRPM) subfamily. TRPM chanzymes are characterized by the enzymatic activity conferred by their C-terminal domain (enzyme module) that is covalently linked to the channel module. For example, TRPM6 and TRPM7 are $Mg^{2+}$-permeable nonselective cation channels with a C-terminal protein kinase module[5].

The kinase module, however, is decoupled from channel gating; instead, it undergoes proteolytic cleavage and is translocated to the nucleus to phosphorylate histones and regulate gene expression[6,7]. Another TRPM chanzyme, the Nudix-linked TRPM2-like protein (hereinafter referred to as TRPM2 chanzyme), is present in early species including chordates, molluscs and ancient organisms beyond the bilateral lineage-like sea anemones (cnidaria) and even in unicellular protists, and is the only type of TRPM channel found in early species[8]. Its channel module is similar to the mammalian TRPM2 in amino acid sequence and overall architecture, forming a $Ca^{2+}$-permeable nonselective cation channel coactivated by ADP-ribose (ADPR) and $Mg^{2+}$ (or $Ca^{2+}$) (refs. 9,10). Its C-terminal enzyme module is named NUDT9-H because of its homology to the NUDT9 protein[11], a mitochondrial ADPR hydrolase that hydrolyzes ADPR into adenosine monophosphate (AMP) and ribose-5-phosphate (R5P) in the presence of $Mg^{2+}$ (ref. 12).

The TRPM2 chanzyme is unique among chanzymes in two aspects. First, it harbors two binding sites for its agonist, ADPR, one in the

[1]Van Andel Institute, Grand Rapids, MI, USA. [2]Present address: Department of Biochemistry, University of Nebraska-Lincoln, Lincoln, NE, USA. ✉e-mail: wei.lu@vai.org; juan.du@vai.org

**Table 1 | Cryo-EM data collection, refinement and validation statistics**

| | srTRPM2–WT–apo | srTRPM2–WT–Ca$^{2+}$ | srTRPM2–WT–Mg$^{2+}$ | srTRPM2–WT–ADPR |
|---|---|---|---|---|
| **Data collection and processing** | | | | |
| Microscope | Titan Krios (FEI) | Titan Krios (FEI) | Titan Krios (FEI) | Titan Krios (FEI) |
| Voltage (kV) | 300 | 300 | 300 | 300 |
| Defocus range (μM) | −1.0–2.5 | −0.9–1.9 | −0.9–1.9 | −0.9–1.9 |
| Exposure time (s) | 8 | 1.5 | 1.5 | 1.5 |
| Dose rate (e$^-$/Å$^2$/s) | 6.8 | 0.86 | 0.86 | 0.86 |
| Number of frames | 40 | 75 | 75 | 75 |
| Pixel size (Å) | 1.048 | 0.826 | 0.826 | 0.826 |
| Particles cleaned | 1,902,236 | 797,499 | 484,184 | 677,227 |
| Particles refined | 168,854 | 207,704 | 199,037 | 11,628 |
| Maps | Relion consensus | Relion consensus | Relion consensus | Relion consensus |
| Symmetry imposed | *C4* | *C4* | *C4* | *C4* |
| Model resolution (Å) | 3.02 | 3.03 | 3.16 | 3.30 |
| FSC threshold | 0.143 | 0.143 | 0.143 | 0.143 |
| Model resolution range (Å) | 2.54–9.89 | 2.64–5.69 | 2.80–6.55 | 2.98–12.10 |
| **Model statistics** | | | | |
| | Tetramer | Tetramer | Tetramer | Tetramer |
| Number of atoms | 39,248 | 36,696 | 38,176 | 39,384 |
| Protein residues | 5,412 | 5,392 | 5,408 | 5,396 |
| Water | 0 | 0 | 0 | 0 |
| Ligand | 12 | 20 | 20 | 16 |
| R.m.s. deviations | | | | |
| Bond lengths (Å) | 0.011 | 0.010 | 0.010 | 0.011 |
| Bond angles (°) | 1.584 | 1.527 | 1.563 | 1.651 |
| Ramachandran plot | | | | |
| Favored (%) | 97.17 | 97.44 | 97.47 | 96.94 |
| Allowed (%) | 2.83 | 2.56 | 2.53 | 3.06 |
| Disallowed (%) | 0 | 0 | 0 | 0 |
| Rotamer outlier (%) | 0 | 0 | 0 | 0.1 |

N-terminal MHR1/2 domain of the channel module and the other in the C-terminal NUDT9-H enzyme module[13]. While the N-terminal ADPR site is absolutely required for channel activation[13–16], the C-terminal NUDT9-H module hydrolyzes ADPR[10], thus separating channel activation from agonist consumption. This is in sharp contrast to CFTR, where agonist ATP is consumed directly in the agonist binding site. Second, the NUDT9-H domain seems to have undergone major evolutionary changes, transitioning from an ADPR hydrolase that indirectly regulates channel gating by depleting agonist in early species, to an ADPR-sensing domain that is indispensable for channel gating in advanced species. This is evidenced by studies on cnidarian *Nematostella vectensis* (*nv*) TRPM2 chanzyme, where the NUDT9-H is an active ADPR hydrolase[10] and removal of this domain does not eliminate ADPR-dependent currents[15]. In contrast, vertebrate TRPM2 lacks ADPR hydrolase activity and is therefore not considered a channel enzyme[17]; mutations in the NUDT9-H domain that affect ADPR binding render the vertebrate TRPM2 insensitive to ADPR[13,14,18,19]. However, it is unclear how the enzymatic activity of the NUDT9-H domain is coupled to channel gating in TRPM2 chanzyme and how NUDT9-H converts its function from a hydrolyase to an indispensable component of channel gating along the evolutionary process.

While seven of the eight members of the TRPM family emerged late in evolution, TRPM2 is the only one that is present from unicellular choanoflagellates to human and may thus represent the evolutionary ancestor of all metazoan TRPM channels[8]. Therefore, dissecting how the gating cycle and enzymatic cycle of the ancient TRPM2 chanzyme are integrated with each other into a complex molecular machine helps us understand the evolution of TRPM channels into different branches with distinct biophysical properties and physiological functions[20], such as temperature sensation and pain, as well as the functional evolution of their C-terminal NUDT9-H domain.

## Results

### The complete catalytic and gating cycles of TRPM2 chanzyme

The key to studying the coupling mechanism between enzymatic activity and channel gating of chanzymes is to capture the structures of all intermediate states in the catalytic and gating cycles. This is challenging if the enzymatic reaction is fast, such as the well-characterized *nv*TRPM2 chanzyme[10], because the substrate will be depleted in the time frame required for preparing cryo-EM samples using state-of-the-art instrumentation, which typically takes at least 5 to 10 seconds. Phylogenetic analysis suggested that choanoflagellate *Salpingoeca rosetta* (*sr*) TRPM2 is also one of the earliest TRPM channels[10] (Supplementary Fig. 1). Functional studies suggested that it is a chanzyme with slower ADPR hydrolysis kinetics than *nv*TRPM2 (ref. 10). To accurately measure the enzyme kinetics of *sr*TRPM2, we developed a highly sensitive and accurate fluorescence-based enzyme kinetics

**Table 2 | Cryo-EM data collection, refinement and validation statistics, continued**

| | srTRPM2–WT–Ca²⁺–ADPR | | srTRPM2–WT–Mg²⁺–AMP–R5P | | | srTRPM2–WT–Mg²⁺–ADPR–4m | |
|---|---|---|---|---|---|---|---|
| **Data collection and processing** | | | | | | | |
| Microscope | Titan Krios (FEI) | | Titan Krios (FEI) | | | Titan Krios (FEI) | |
| Voltage (kV) | 300 | | 300 | | | 300 | |
| Defocus range (µM) | −1.0–2.5 | | −0.9–1.9 | | | −0.9–1.9 | |
| Exposure time (s) | 8 | | 1 | | | 1.5 | |
| Dose rate (e⁻/Å²/s) | 6.8 | | 1 | | | 0.86 | |
| Number of frames | 40 | | 50 | | | 75 | |
| Pixel size (Å) | 1.048 | | 0.826 | | | 0.826 | |
| Particles cleaned | 1,053,242 | | 1,572,396 | | | 634,404 | |
| Particles refined | 741,258 | | 952,710 | | | 148,230 | |
| Maps | Relion consensus | NUDT9-H focused | cryoSPARC consensus | Relion consensus | NUDT9-H focused | Relion consensus | NUDT9-H focused |
| Symmetry imposed | *C4* | *C1* | *C4* | *C4* | *C1* | *C4* | *C1* |
| Resolution (Å) | 2.99 | 3.69 | 1.97 | 2.16 | 2.80 | 3.08 | 3.83 |
| FSC threshold | 0.143 | 0.143 | 0.143 | 0.143 | 0.143 | 0.143 | 0.143 |
| Model resolution range (Å) | 2.60–5.30 | 3.30–10.84 | 1.73–7.09 | 1.94–4.92 | 2.67–7.51 | 2.72–6.64 | 3.54–12.25 |
| **Model statistics** | | | | | | | |
| | Tetramer | NUDT9-H | Tetramer | NUDT9-H | | Tetramer | |
| Number of atoms | 40,281 | 2,086 | 43,021 | 2,228 | | 39,976 | |
| Protein residues | 5,464 | 289 | 5,528 | 292 | | 5,464 | |
| Water | 4 | 0 | 52 | 2 | | 0 | |
| Ligand | 37 | 3 | 37 | 4 | | 36 | |
| R.m.s. deviations | | | | | | | |
| Bond lengths (Å) | 0.011 | 0.013 | 0.011 | 0.004 | | 0.011 | |
| Bond angles (°) | 1.598 | 1.344 | 1.508 | 0.980 | | 1.616 | |
| Ramachandran plot | | | | | | | |
| Favored (%) | 97.35 | 96.82 | 97.52 | 97.22 | | 97.42 | |
| Allowed (%) | 2.65 | 3.18 | 2.48 | 2.78 | | 2.58 | |
| Disallowed (%) | 0 | 0 | 0 | 0 | | 0 | |
| Rotamer outlier (%) | 0 | 0 | 0 | 0 | | 0.2 | |

assay capable of detecting ADPR at the nanomolar level (Methods and Extended Data Fig. 1a–e). Our data showed that the *sr*TRPM2 is among the slowest known enzymes with the turnover number ($k_{cat}$) an order of magnitude lower than those of *nv*TRPM2 (as detailed in the section '*sr*NUDT9-H modulates agonist availability via slow hydrolysis'), making it a promising candidate for capturing catalytic intermediates by performing time-resolved cryo-EM studies on time scales of seconds to minutes. Indeed, by precisely controlling the ligand condition and timing of ADPR hydrolysis during the preparation of cryo-EM samples (Methods), we determined a series of structural snapshots of *sr*TRPM2 in the ADPR hydrolysis cycle, as well as various functional states in the channel gating cycle, including a total of 14 structures (Tables 1–5). These structures were determined at high resolutions (up to 1.97 Å) (Fig. 1a and Extended Data Figs. 2–4), which allowed us to unambiguously define the binding of substrates, lipids, and ions, as well as the closed and open conformations of the ion-conducting pore (Fig. 1b and Supplementary Video 1), thus revealing the complete catalytic and gating cycles of the TRPM2 chanzyme.

**The TRPM2 chanzyme is regulated by five ligand binding sites**
In the presence of ADPR, *sr*TRPM2 can be activated by Ca²⁺ or Mg²⁺, yielding similar voltage-independent currents with high

channel open probabilities (Fig. 1c and Extended Data Fig. 5), whereas ADPR hydrolysis in NUDT9-H occurs only in the presence of Mg²⁺ but not Ca²⁺ (refs. 10,21). By contrast, vertebrate TRPM2 opens only in the presence of Ca²⁺ with a lower open probability[22], and cannot hydrolyze ADPR[17].

The structures of *sr*TRPM2 exhibit a characteristic TRPM architecture (Figs. 1d and 2a), from top to bottom, a transmembrane domain (TMD) layer containing the ion-conducting pore, a signal transduction layer consisting of MHR3/4 domain and the C-terminal rib helix, and an ADPR-sensing layer consisting of the N-terminal MHR1/2 domain and the C-terminal NUDT9-H domain. In *sr*TRPM2, the NUDT9-H domain adopts a 'vertical' pose, as opposed to the 'horizontal' pose observed in human and zebrafish TRPM2 (Extended Data Fig. 6a). This difference accounts for the markedly longer but slimmer shape of *sr*TRPM2 in comparison to its vertebrate orthologs and, more importantly, is closely linked to the molecular evolution of the NUDT9-H domain, which has let to distinct functions of NUDT9-H in early and advanced species TRPM2 (detailed in the last two sections).

Obvious conformational differences were observed between the *sr*TRPM2 structures in ligand-free and ligand-bound states, which is mainly manifested by the rearrangement of the MHR1/2 and NUDT9-H

**Table 3 | Cryo-EM data collection, refinement and validation statistics, continued**

| | srTRPM2–WT–Mg$^{2+}$–ADPR–10s | | | | | | |
|---|---|---|---|---|---|---|---|
| **Data collection and processing** | | | | | | | |
| Microscope | Titan Krios (FEI) | | | | | | |
| Voltage (kV) | 300 | | | | | | |
| Defocus range (µM) | −0.9–1.9 | | | | | | |
| Exposure time (s) | 1 | | | | | | |
| Dose rate (e$^-$/Å$^2$/s) | 1 | | | | | | |
| Number of frames | 50 | | | | | | |
| Pixel size (Å) | 0.826 | | | | | | |
| Particles cleaned | 1,385,278 | | | | | | |
| Gate conformation | Open | | Closed | | | | |
| Particles refined | 35,103 | | 971,893 | 1,272 | | 1,761 | |
| ADPR$_C$ status | NA | | Total | Intact | | Hydrolyzed | |
| Maps | Relion consensus | NUDT9-H focused | Relion consensus | Relion consensus | NUDT9-H focused | Relion consensus | NUDT9-H focused |
| Symmetry imposed | C4 | C1 | C4 | C4 | C1 | C4 | C1 |
| Model resolution (Å) | 2.93 | 3.13 | 2.27 | 3.87 | 2.80 | 3.71 | 2.80 |
| FSC threshold | 0.143 | 0.143 | 0.143 | 0.143 | 0.143 | 0.143 | 0.143 |
| Model resolution range (Å) | 2.64–7.09 | 3.05–5.03 | 2.04–4.99 | 3.19–11.17 | 2.70–5.26 | 3.07–10.92 | 2.69–5.37 |
| **Model statistics** | | | | | | | |
| | Tetramer | NUDT9-H | Tetramer | | NUDT9-H | Tetramer | NUDT9-H |
| Number of atoms | 42,686 | 2,204 | 42,781 | | 2,260 | 42,789 | 2,252 |
| Protein residues | 5,472 | 285 | 5,508 | | 292 | 5,508 | 292 |
| Water | 9 | 0 | 12 | | 3 | 4 | 0 |
| Ligand | 41 | 4 | 41 | | 4 | 45 | 5 |
| R.m.s. deviations | | | | | | | |
| Bond length (Å) | 0.011 | 0.012 | 0.012 | | 0.004 | 0.012 | 0.006 |
| Bond angle (°) | 1.746 | 1.164 | 1.786 | | 1.028 | 1.719 | 1.001 |
| Ramachandran plot | | | | | | | |
| Favored (%) | 94.83 | 96.42 | 97.46 | | 95.49 | 96.79 | 96.53 |
| Allowed (%) | 5.17 | 3.58 | 2.54 | | 4.51 | 3.21 | 3.47 |
| Disallowed (%) | 0 | 0 | 0 | | 0 | 0 | 0 |
| Rotamer outlier (%) | 0 | 0.46 | 0 | | 0 | 0 | 0 |

domains (Fig. 1d). We identified five ligand binding sites, including two sets of ADPR-binding sites (one of which also binds the hydrolysis products AMP and R5P) and three sets of cation binding sites (Figs. 1d and 2a). Some of these ligand binding sites are conserved among TRPM2 orthologs, while others are not.

The agonist ADPR was found in both the N-terminal MHR1/2 domain of the channel module (ADPR$_N$) and C-terminal NUDT9-H enzyme module (ADPR$_C$) domains (Fig. 2a). The ADPR binding to the MHR1/2 domain is conserved across all the TRPM2 orthologs (Extended Data Fig. 6b). Two conserved residues, F268 for π–π stacking with the adenine group of ADPR and R275 for hydrogen bonding with the ribose group of ADPR, are essential for ADPR binding in srTRPM2 (Fig. 2b). Replacement of either of these residues with alanine rendered srTRPM2 insensitive to ADPR (Extended Data Fig. 6c,d). These data confirmed that the function of the MHR1/2 domain in activating TRPM2 is preserved from early to advanced species[13–16]. By contrast, the ADPR binding to the NUDT9-H domain differs between TRPM2 in early and advanced species. Specifically, ADPR in srNUDT9-H adopts an entirely different conformation from that in human NUDT9-H (Protein

Data Bank (PDB) ID 6PUS), and is coordinated by cations (3× Mg$^{2+}$ or 2× Ca$^{2+}$) that are absent in human NUDT9-H (Fig. 2c,d).

The cation binding site in the S1-S4 domain of TMD is conserved in all the Ca$^{2+}$-activated TRPM channels (TRPM2/4/5) and is required for channel activation (Extended Data Fig. 4d)[13,14,23–27]. However, only srTRPM2 can also be activated by binding of Mg$^{2+}$ at this site (Figs. 1c and 2e). A cation binding site is found between the MHR3/4 domain and the Rib helix of each subunit, termed Mg$_{MHR}$ (Fig. 2f). This site does not distinguish between Ca$^{2+}$ or Mg$^{2+}$ either (Extended Data Fig. 7a,b) and is unique to srTRPM2 as the key residue, E1114, is not conserved (Supplementary Fig. 2). Replacing E1114 with alanine eliminated cation binding at Mg$_{MHR}$, resulting in a reduced rotational movement of the intracellular domain around the symmetry axis when ligands were bound, as compared to the wild-type (WT) protein (Extended Data Fig. 7c). The E1114A mutation had no obvious effect on macroscopic currents. However, single-channel analysis showed a small decrease in channel open probability and a small increase in channel close rate (Extended Data Fig. 7d,e and Supplementary Fig. 3a–c), revealing a regulatory role of this cation binding site on channel gating.

**Table 4 | Cryo-EM data collection, refinement and validation statistics, continued**

| | srTRPM2–E1114A–Mg²⁺-ADPR-5s | | | |
|---|---|---|---|---|
| **Data collection and processing** | | | | |
| Microscope | Titan Krios (FEI) | | | |
| Voltage (kV) | 300 | | | |
| Defocus range (μM) | −0.9–1.9 | | | |
| Exposure time (s) | 1.5 | | | |
| Dose rate (e⁻/Å²/s) | 0.86 | | | |
| Number of frames | 75 | | | |
| Pixel size (Å) | 0.826 | | | |
| Particles cleaned | 1,122,170 | | | |
| Gate conformation | Open | | Closed | |
| Particles refined | 8,279 | | 379,553 | |
| Maps | Relion consensus | NUDT9-H focused | Relion consensus | NUDT9-H focused |
| Symmetry imposed | *C4* | *C1* | *C4* | *C1* |
| Model resolution (Å) | 3.87 | 4.08 | 2.82 | 3.27 |
| FSC threshold | 0.143 | 0.143 | 0.143 | 0.143 |
| Model resolution range (Å) | 3.48–10.09 | 4.18–6.77 | 2.48–5.57 | 2.97–11.78 |
| **Model statistics** | | | | |
| | Tetramer | | Tetramer | |
| Number of atoms | 39,468 | | 40,636 | |
| Protein residues | 5,464 | | 5,472 | |
| Water | 0 | | 0 | |
| Ligand | 36 | | 36 | |
| R.m.s. deviations | | | | |
| Bond lengths (Å) | 0.009 | | 0.011 | |
| Bond angles (°) | 1.272 | | 1.637 | |
| Ramachandran plot | | | | |
| Favored (%) | 95.61 | | 96.76 | |
| Allowed (%) | 4.39 | | 3.24 | |
| Disallowed (%) | 0 | | 0 | |
| Rotamer outlier (%) | 0.1 | | 0.3 | |

## Snapshots of ADPR hydrolysis

In the presence of Mg²⁺, srNUDT9-H is catalytically active, which allowed us to capture five distinct states of the ADPR hydrolysis cycle (Fig. 3a). These structures reveal the detailed molecular mechanisms underlying ADPR hydrolysis. Before the onset of hydrolysis (substrate-bound state), both MHR1/2 and NUDT9-H domains are occupied by ADPR. Within NUDT9-H, the adenine group of ADPR is sandwiched by the indole group of W1264 and the phenyl group of F1372 via π–π stacking (Fig. 2c). The double alanine mutation of the corresponding residues in NUDT5—another ADPR hydrolase from the Nudix hydrolase family like NUDT9 (ref. 28)—reduced its catalytic efficiency by four orders of magnitude[29]. The terminal ribose group of ADPR forms multiple hydrogen bonds with the side chains of D1330, D1426 and R1428 (Fig. 2c). Three Mg²⁺ cofactors (Mg₁, Mg₂ and Mg₃) are bound between the two phosphate groups of ADPR and two acidic residues on the Nudix helix, E1386 and E1390 (Fig. 2c). A glutamate residue corresponding to E1386 is conserved in NUDT5, NUDT9 and TRPM2 chanzymes, but is replaced by isoleucine in vertebrate TRPM2 channels (Supplementary Fig. 2); replacing the corresponding residue in NUDT5 or NUDT9 with glutamine or isoleucine, respectively, reduced the catalytic efficiency by four or three orders of magnitude[29,30], indicating its key role in metal binding and ADPR

hydrolysis. The β-phosphate (Pβ) of ADPR forms salt bridge interactions with the conserved R1360 and R1428 (Fig. 2c).

The extensive interactions between ADPR and the NUDT9-H domain strategically anchor both termini of the ADPR molecule, exposing only its α-phosphate (Pα) to solvent and making it readily accessible for nucleophilic attack (Fig. 2c). Indeed, we observed a putative water molecule bridging Mg₂ and Mg₃, as previously found in the crystal structure of NUDT5 (Fig. 3b and Extended Data Fig. 4c)[29]. On activation by Mg²⁺ ions, the water molecule is deprotonated by the catalytic base D1460 located in a loop close to the Nudix helix, and is poised for nucleophilic attack on the phosphorus atom of Pα (Fig. 3c). Mutations in the equivalent residues of NUDT5 and NUDT9 have been shown to reduce the catalytic efficiency of the enzymes by two orders of magnitude relative to the WT form[29,30]. The water molecule breaks down the phosphoester bond, resulting in the production of AMP and R5P, as observed in intermediate state I (Fig. 3a,d and Extended Data Fig. 4c). The release of the hydrolysis products triggers a conformational change of NUDT9-H in preparation for the next catalytic cycle, as seen in intermediate state II (Fig. 3a). It is noteworthy that the MHR1/2 domain remains bound to an intact ADPR in all structures except for one, which was determined using protein samples incubated with ADPR for an extended period. In this instance, the ADPR molecules are

presumably depleted, leaving the MHR1/2 domain unoccupied while the NUDT9-H domain bound to the hydrolysis products (intermediate state III; Fig. 3a,e and Extended Data Fig. 4c). Following release of these hydrolysis products, the protein returns to the substrate-free state (Fig. 3a).

Although both $Ca^{2+}$ and $Mg^{2+}$ activate the channel, the $Mg^{2+}$-dependent ADPR hydrolysis of srTRPM2 is inhibited by $Ca^{2+}$ (Extended Data Fig. 1f). This is because in the $Mg^{2+}$-bound active site, the three $Mg^{2+}$ cofactors are closely coordinated around the α-phosphate (Fig. 3c), likely inducing a favorable geometric distortion around the phosphorus atom for hydrolysis[31]. In contrast, in the $Ca^{2+}$-bound active site, only two $Ca^{2+}$ occupy the equivalent positions of $Mg_2$ and $Mg_3$ (Extended Data Fig. 8a,b), which presumably does not induce a sufficient genometric distortion around the α-phosphorus atom necessary for hydrolysis. Therefore, $Ca^{2+}$ inhibits the enzyme by competing with $Mg^{2+}$ for the metal binding sites in NUDT9-H.

### The srNUDT9-H modulates channel activity via tetramerization

In the apo state, the ADPR-sensing layer, particularly the NUDT9-H enzyme module of srTRPM2, is flexible due to a lack of interface with the channel module (Fig. 4a and Extended Data Fig. 9a). This layer underwent a marked rearrangement on binding of ADPR (or its hydrolysis products) and divalent cations, and is stabilized by the formation of extensive interactions between NUDT9-H and the rest of the protein (Extended Data Fig. 9b,c). We noticed that the stabilization of NUDT9-H was caused, at least in part, by a peptide recruited from the adjacent subunit (Fig. 4b). This intersubunit binding was confirmed by cross-linking experiments (Fig. 4c,d). This peptide, which is situated in the linker region connecting the C-terminal pole helix and NUDT9-H, is composed of 11 residues and forms a short helix. As it buckles the NUDT9-H from the adjacent subunit and tightens all four NUDT9-H domains together, we named it a buckle helix. The sequence of the buckle helix is not conserved (Supplementary Fig. 2), and the corresponding peptide is disordered in the available vertebrate TRPM2 structures[13,14,26,27,32].

The tetramerization of the NUDT9-H domain through the buckle helix remodels the interface between the enzymatic module and the channel module (Extended Data Fig. 9b,c). This raises the question of whether the NUDT9-H enzyme module plays a direct role in channel gating of srTRPM2 on ADPR binding. To this end, we generated a truncated construct by removing NUDT9-H, srTRPM2-ΔNUDT9-H. Despite reduced protein expression level at the plasma membrane, the truncation construct still responded to ADPR and $Ca^{2+}$, generating currents with similar characteristics to the WT (Extended Data Fig. 10a–c). This is distinct to human TRPM2 (hsTRPM2), in which removal of NUDT9-H abolished ADPR-induced current[13]. Single-channel analysis of srTRPM2-ΔNUDT9-H revealed channel gating kinetics similar to the WT, albeit with a reduced open rate (Extended Data Fig. 10d and Supplementary Fig. 4a–c). Notably, this construct exhibited long-lived closed states between bursts of channel opening, a pattern not observed in the WT protein (Figs. 1c and 4e). In agreement with the functional data, the overall structure of srTRPM2-ΔNUDT9-H closely resembles the channel module of srTRPM2, but with some minor conformational changes that may have contributed to the subtle differences in gating kinetics (Extended Data Fig. 10e–h). Our result indicates that the NUDT9-H domain is dispensable for the channel activation of srTRPM2, which is consistent with the nvTRPM2 chanzyme[15], but plays a role in the regulation of the channel activity.

We then set out to understand how NUDT9-H directly modulates channel activity. When srTRPM2 was incubated with ADPR and $Mg^{2+}$ for a long enough time before cryo-EM sample preparation, ADPR was completely hydrolyzed in the structure. We found that the MHR1/2 domain became ligand-free and adopted the same conformation as in the apo structure. On the other hand, NUDT9-H was occupied by the hydrolysis products AMP, R5P and $Mg^{2+}$ (Extended Data Fig. 8d,e),

**Table 5 | Cryo-EM data collection, refinement and validation statistics, continued**

| | srTRPM2–ΔNUDT9-H–apo | srTRPM2–ΔNUDT9-H–Ca²⁺/ADPR |
|---|---|---|
| **Data collection and processing** | | |
| Microscope | Talos Arctica (FEI) | Talos Arctica (FEI) |
| Voltage (kV) | 200 | 200 |
| Defocus range (μM) | −1.1–2.5 | −1.1–2.5 |
| Exposure time (s) | 8 | 8 |
| Dose rate (e⁻/Å²/s) | 1.61 | 1.61 |
| Number of frames | 40 | 40 |
| Pixel size (Å) | 1.16 | 1.1.6 |
| Particles cleaned | 867,924 | 375,251 |
| Particles refined | 191,126 | 156,178 |
| Maps | Relion consensus | Relion consensus |
| Symmetry imposed | C4 | C4 |
| Resolution (Å) | 3.74 | 4.14 |
| FSC threshold | 0.143 | 0.143 |
| Model resolution range (Å) | 3.52–6.65 | 3.64–10.40 |
| **Model statistics** | | |
| | Tetramer | Tetramer |
| Number of atoms | 26,156 | 27,328 |
| Protein residues | 4,296 | 4,284 |
| Water | 0 | 0 |
| Ligand | 12 | 24 |
| R.m.s. deviations | | |
| Bond length (Å) | 0.008 | 0.008 |
| Bond angle (°) | 1.133 | 1.223 |
| Ramachandran plot | | |
| Favored (%) | 97.10 | 96.95 |
| Allowed (%) | 2.90 | 3.05 |
| Disallowed (%) | 0 | 0 |
| Rotamer outlier (%) | 0.3 | 0 |

but adopted almost the same conformation as when it was complexed with intact ADPR and $Mg^{2+}$. Therefore, by comparing this structure with the apo structure, we can approximately analyze how the binding of ADPR and $Mg^{2+}$ to NUDT9-H affects the conformation of the channel module of the chanzyme. In the apo structure, the NUDT9-H enzyme module formed only one interface with the MHR1/2 domain of the same subunit and was thus flexible and unlikely to have a major impact on the conformation of the channel module (Fig. 4a and Extended Data Fig. 9b). By contrast, in the presence of ligands, the NUDT9-H layer underwent tetramerization via the buckle helix, causing a clockwise rotation viewed from the intracellular side and a contraction of the NUDT9-H layer (Fig. 4b). This created multiple additional interfaces of the NUDT9-H enzyme module with cognate and adjacent subunits (Extended Data Fig. 9c), ultimately leading to a rotational movement of the signal transduction layer—the MHR3/4 domain and the rib helix (Extended Data Fig. 8f). Our previous studies on vertebrate TRPM2 have shown that rotational movement of the MHR3/4 domain is a key factor in transducing the signal from agonist binding to channel gating[13,14]. We therefore propose that binding of ADPR (or its hydrolysis products) to the NUDT9-H enzyme module leads to buckle helix-mediated self-tetramerization, which

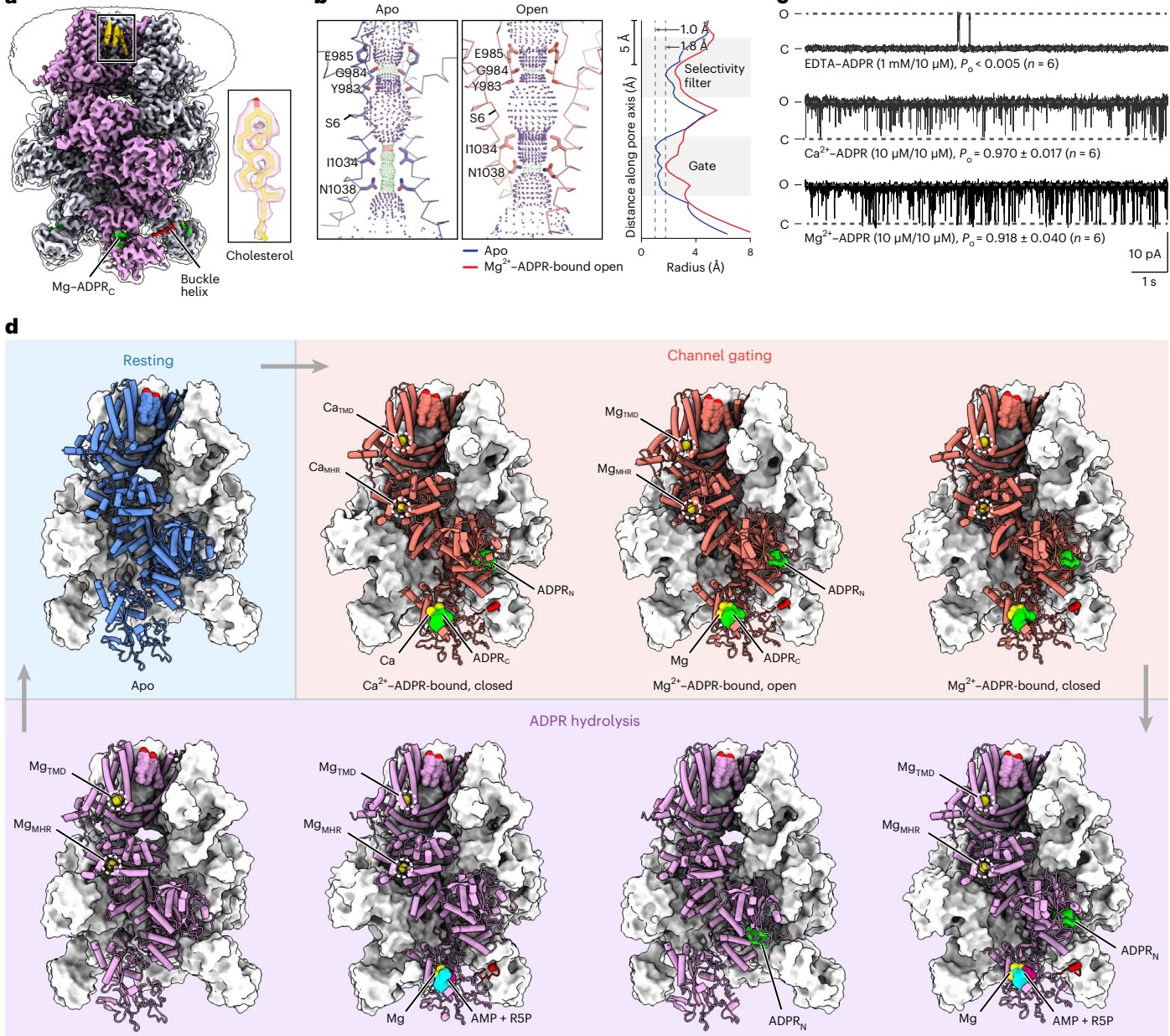

**Fig. 1 | The complete gating and catalytic cycles of the _sr_TRPM2 chanzyme.**
**a**, The unsharpened composite cryo-EM map of _sr_TRPM2–WT–Mg²⁺–AMP–R5P viewed parallel to the membrane. The density of a representative bound cholesterol molecule is shown in the box. **b**, The ion-conducting pore in the apo (left) and Mg²⁺–ADPR-bound open (middle) states viewed parallel to the membrane, with plots of the pore radius along the pore axis (right). The pore region (shown as a cartoon), residues (shown with sticks) forming the gate and the selectivity filter in two subunits are shown. Purple, green and red spheres define radii of >2.3, 1.2–2.3 and <1.2 Å, respectively. **c**, Example traces of inside-out patch recordings of _sr_TRPM2 in the presence of EDTA–ADPR (_n_ = 6, each lasting 60–120 s), Ca²⁺–ADPR (_n_ = 6, each lasting 50–100 s) and

Mg²⁺–ADPR (_n_ = 6, each lasting 30–100 s), respectively, clamped at +60 mV. The channel open probability ($P_\mathrm{o}$) is reported as mean ± s.d. **d**, Structural snapshots of the gating and catalytic cycles of the _sr_TRPM2 chanzyme. Of note, the intermediate state II of the ADPR hydrolysis was approximated by the _sr_TRPM2 structure bound with ADPR. The only difference between this structure and the true intermediate state II is the lack of Mg²⁺ binding in the Mg_TMD_ and Mg_MHR_ sites. However, our analysis indicates that the absence of Mg²⁺ binding in these sites does not cause major conformational changes in the MHR1/2 and NUDT9-H domains. Therefore, we consider this structure to be a reasonable approximation of the intermediate state II.

facilitates the movement of the signal transduction layer, thereby regulating channel function. Consistent with this notion, replacing residues of the buckle helix with alanine (_sr_TRPM2–BH2A) or truncating the buckle helix (_sr_TRPM2–ΔBH), both of which are expected to weaken the tetramerization of NUDT9-H, resulted in the occurrence of long-lived closed states, similar to those observed in _sr_TRPM2–ΔNUDT9-H (Fig. 4e).

**_sr_NUDT9-H modulates agonist availability via slow hydrolysis**

As an ADPR hydrolase, the NUDT9-H domain of _sr_TRPM2 also couples indirectly to channel gating by regulating the local concentration of agonist. This indirect coupling, however, requires that the kinetics of the hydrolysis must be substantially slower than the kinetics of the channel activation, otherwise the agonist would be depleted before it has a chance to induce channel opening. Indeed, _sr_TRPM2 is among the

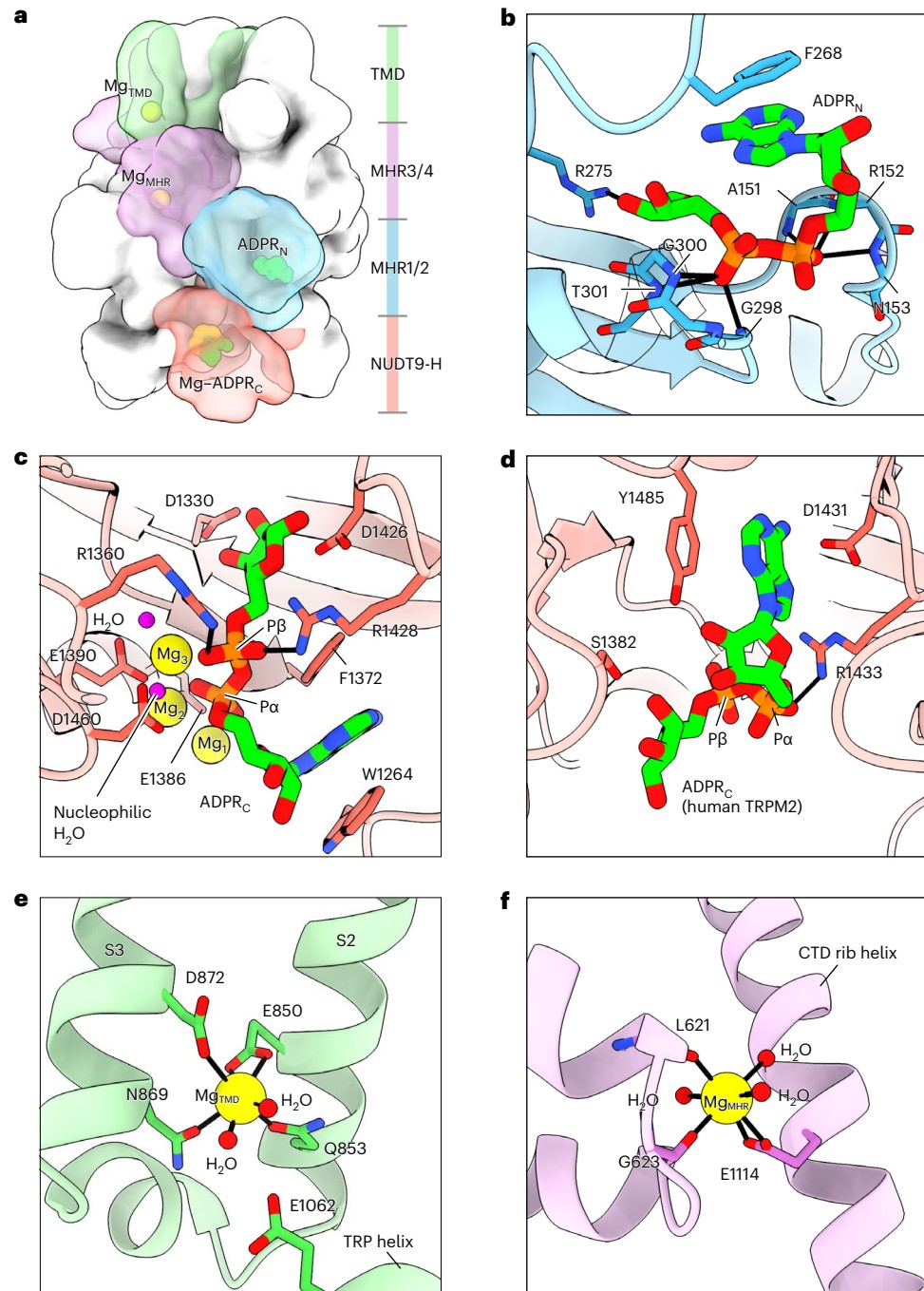

**Fig. 2 | Ligand binding sites. a**, The overall structure of *sr*TRPM2–WT–Mg²⁺–ADPR in surface representation viewed parallel to the membrane. One subunit is highlighted, with the four domains colored differently and the ligands shown as spheres. **b**, The ADPR_N site. The ADPR_N molecule and interacting residues are shown as sticks. Polar interactions are indicated by thick black bars. **c,d**, The ADPR_C site in *sr*TRPM2 (**c**) and *hs*TRPM2–WT–Ca²⁺–ADPR (PDB ID 6PUS) (**d**).

The ADPR molecules and interacting residues are shown as sticks. The Mg²⁺ cofactors and water molecules are shown as spheres. Note that the orientation of ADPR_C molecules in **c** and **d** is reversed, despite the two NUDT9-H domains have the same orientation. **e,f**, The Mg_TMD site (**e**) and Mg_MHR site (**f**). The Mg²⁺ cations and water molecules are shown as spheres. Polar interactions are indicated by thick black bars.

slowest enzymes with a $k_{cat}$ of 3 (Fig. 5a,b), similar to RuBisCo involved in carbon fixation[33].

To understand how the *sr*TRPM2 chanzyme has evolved an extremely slow enzyme module to accommodate the function of the channel module, we compared the active site in the ligand-free state and when bound to ADPR (or the hydrolysis products). We expect different conformations, as the free enthalpy of the hydrolysis reaction provides the driving force to overcome the stability of the protein and disassemble the active site, allowing the products to leave. Indeed, an AlphaFold-predicted[34] model of the *sr*NUDT9-H domain in the ligand-free form—which is otherwise challenging to obtain in high resolution due to its high flexibility in the full-length protein—has an open clamshell conformation, in contrast to the closed clamshell conformation bound to ADPR (or the hydrolysis products) in the cryo-EM structure, as indicated by the change in distance between the arginine pair R1360–R1428 involved in ADPR binding (Fig. 5c). Structural comparison further revealed that the opening of the clamshell requires the movement of the Nudix helix

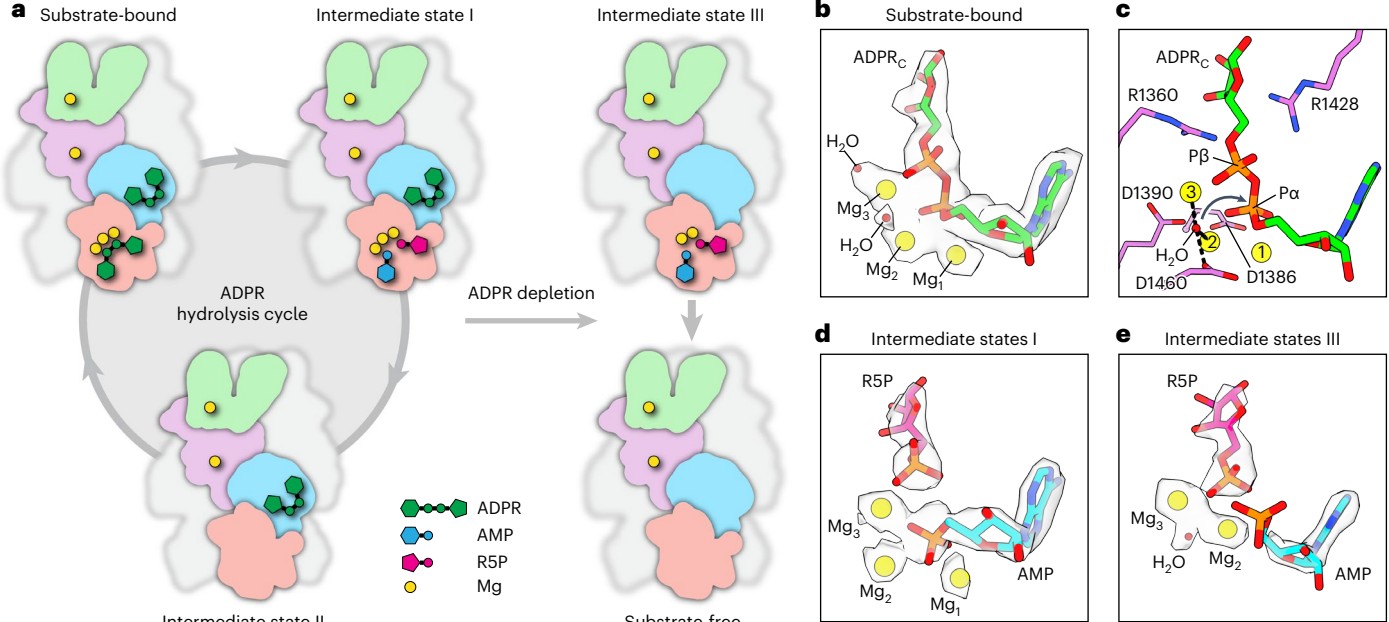

**Fig. 3 | Snapshots of ADPR hydrolysis. a**, Cartoon representation of the structural snapshots during ADPR hydrolysis. Note that we were unable to obtain the structure of a bona fide intermediate state II. However, the structure in the presence of EDTA–ADPR provides a reasonable approximation. In this structure, the ADPR$_N$ site in the channel module is occupied, while the enzyme module is unoccupied and adopts an apo-like conformation. The only difference between this structure and a bona fide intermediate state II is the lack of Mg$_{TMD}$ and Mg$_{MHR}$ (Fig. 1d), two cations that cause only minor conformational changes. Note that ADPR binding to the N- and C-terminal sites occurs independently, and each is in a dynamic equilibrium with the environmental ADPR pool. **b**, Cryo-EM densities of the intact ADPR$_C$, Mg$^{2+}$ cofactors and putative water molecules in the substrate-bound NUDT9-H from $sr$TRPM2–WT–Mg$^{2+}$–ADPR–10s. The ADPR molecule is shown as sticks. The Mg$^{2+}$ cofactors and putative water molecules are shown as spheres. **c**, Key residues that coordinate the Mg$^{2+}$

cofactors and nucleophilic water molecule, as well as the two arginine residues that interacts with Pβ of ADPR. The arrow indicates the nucleophilic attack of the water molecule on Pα of ADPR for hydrolysis. **d,e**, Cryo-EM densities of the hydrolysis products (AMP and R5P) and Mg$^{2+}$ cofactors in NUDT9-H of intermediate state I from $sr$TRPM2–WT–Mg$^{2+}$–ADPR–10s (**d**) and intermediate state III from $sr$TRPM2–WT–Mg$^{2+}$–AMP–R5P (**e**). The ADPR molecule is shown with sticks. The Mg$^{2+}$ cofactors and water molecules are shown as spheres. Note that in intermediate state III, AMP adopts a different conformation from that in intermediate state I, with its phosphate group dissociated from R5P, rendering it flexible, as evidenced by the lack of resolved density of AMP's phosphate group. This change leads to the loss of Mg$_1$, which interacts with AMP's phosphate group in intermediate state I, and the movement of Mg$_2$ to fill the position of AMP's phosphate group in intermediate state I.

that forms part of the active site (Fig. 5c, black arrow). While the Nudix helix is free to move in an independent monomeric NUDT9-H domain, it is constrained in the $sr$TRPM2 chanzyme when loaded with ADPR, as the NUDT9-H domains tetramerize via the buckle helix, which is inserted into the cleft between the Nudix helix and a nearby helix (Fig. 5c), preventing the movement of the Nudix helix to release the products. To release hydrolysis products, the buckle helix must disassociate from the adjacent NUDT9-H domain, an additional step that slows down the enzymatic reaction. In support of this idea, successive steps of detetramerization of the NUDT9-H domain were observed in the cryo-EM data of ADPR being completely hydrolyzed (Fig. 5d). Our data suggest that buckle helix-mediated tetramerization exerts an inhibitory effect on the enzyme module. This is consistent with measurements of enzyme kinetics, which showed that the $k_{cat}$ of the full-length $sr$TRPM2 is more than twofold lower than that of the isolated $sr$NUDT9-H domain. Furthermore, as expected, replacing residues of the buckle helix with alanine ($sr$TRPM2–BH2A) or truncating the buckle helix ($sr$TRPM2–ΔBH) resulted in a $k_{cat}$ between those of the full-length $sr$TRPM2 and the isolated $sr$NUDT9-H domain (Fig. 5b).

In summary, the enzymatic NUDT9-H domain modulates the channel gating in two different ways. Directly, it up-regulates the channel activation by facilitating signal transduction from the ADPR-sensing layer to the channel gate. Indirectly, it down-regulates the channel activation by reducing the available ADPR through its slow hydrolase activity.

### Loss-of-function of NUDT9-H as hydrolase in vertebrates

To understand the molecular basis of how the NUDT9-H domain loses ADPR hydrolase activity in advanced species, we analyzed the amino acid sequences and performed structural comparisons of $sr$TRPM2 and $hs$TRPM2, focusing on two sets of important residues responsible for ADPR hydrolysis and binding, respectively.

Residues responsible for ADPR hydrolysis include those that coordinate the three Mg$^{2+}$ cofactors and the catalytic water molecule in the active site, which vary considerably between early and advanced species[10] (Supplementary Fig. 2). For instance, D1460, responsible for deprotonating the catalytic water molecule in $sr$TRPM2, becomes S1469 in $hs$TRPM2, and E1386, responsible for coordinating the catalytic Mg$_2$, becomes I1405 in $hs$TRPM2. As a result, human NUDT9-H can no longer bind Mg$^{2+}$ and activate the nucleophilic water, becoming catalytically inactive[10,17].

Residues responsible for ADPR binding primarily include those that interact with both termini of the ADPR molecule, which also differ considerably between early and advanced species[29,30] (Supplementary Fig. 2). In $sr$TRPM2, the adenine moiety (head) of ADPR is sandwiched by W1264 and F1372 (Fig. 2c), which are either absent (former) or replaced by a serine residue (latter) in $hs$TPRPM2. The terminal ribose moiety (tail) of ADPR interacts with D1330 of $sr$TRPM2 (Fig. 2c), which is replaced in $hs$TRPM2 by a tyrosine residue whose bulky sidechain extends into the binding pocket, thus competing for space with the ribose moiety. These differences reversed the pose of the ADPR in $hs$TRPM2 (PDB ID 6PUS) from head to tail (Fig. 2c,d). As a result, a

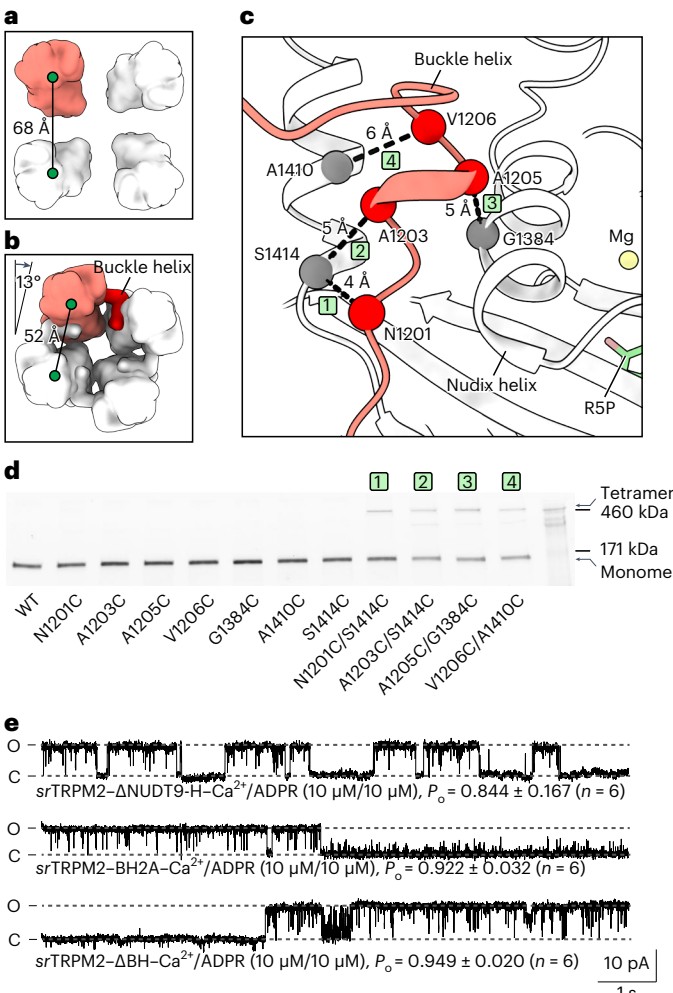

**Fig. 4 | The srNUDT9-H modulates channel activity via cation/ADPR-induced tetramerization. a,b**, Organization of the NUDT9-H layer in the apo state (**a**) and substrate-bound state (**b**), viewed from cytoplasmic side. The NUDT9-H domains are shown in surface representation, with one subunit highlighted in red. The center-of-mass distances of adjacent NUDT9-H domains, as well as the rotation of the NUDT9-H layer in the substrate-bound state relative to the apo state, are indicated. **c**, Interactions of the buckle helix (red) with NUDT9-H from adjacent subunit (white). The buckle helix and NUDT9-H are shown in a cartoon representation. The Cα atoms of residues selected for disulfide cross-linking experiments are shown as spheres, with dashed lines marking Cα distances between residue pairs that potentially form disulfide bonds if mutated to cysteine. The IDs of residue pairs match those in **d**. **d**, Assessing site-directed disulfide cross-linking assessed through in-gel fluorescence signals. GFP-tagged srTRPM2–WT and single or double cysteine mutants of selected residues shown in **d** were analyzed by nonreducing SDS–PAGE for three times. The cross-linked tetramer bands were consistently observed. **e**, Example traces of inside-out patch recordings of srTRPM2–ΔNUDT9-H (with both buckle helix and NUDT9-H domain were truncated) (n = 6, each lasting 50–100 s), srTRPM2–BH2A (with all buckle helix residues mutated to alanines) (n = 6, each lasting 90–140 s) and srTRPM2–ΔBH (with the buckle helix truncated) (n = 6, each lasting 30–110 s) clamped at +60 mV.

conserved arginine residue switches its interaction partner from the β-phosphate group of ADPR in srTRPM2 (R1428) to the α-phosphate group of ADPR in hsTRPM2 (R1433), protecting the α-phosphate group from nucleophilic attack (Fig. 2c,d).

## Gain-of-function of NUDT9-H as gating module in vertebrates

The NUDT9-H domain of TRPM2 in advanced species loses its function as an ADPR hydrolase during evolution, but becomes an ADPR-binding module essential for channel activation[13,14,17]. While we have shown that

the loss of enzymatic function is caused by changes in the active site, we hypothesize that NUDT9-H domain gains function in channel gating by enhancing its interaction with the channel module, which would allow the local conformational changes induced by ADPR binding to propagate to the distal ion channel conduction pore.

In apo srTRPM2, the interface between the enzyme module and the channel module is formed by the core region of the NUDT9-H domain and the MHR2 domain of the same subunit, mainly contributed by charged interactions (Fig. 6a). This type of interaction is known to play an important role in protein–protein interactions[35]. Notably, the NUDT9-H domain does not form additional interfaces with adjacent subunits in apo srTRPM2 (Fig. 6c), which, together with the charged intrasubunit interactions, suggests that NUDT9-H may operate as a semi-independent module. Supporting this idea, both the channel module and enzyme module can be expressed and purified separately, and each is functional. In the intrasubunit interface, the positively charged R1347 in the P-loop of the core region[30] is inserted into a negatively charged notch in the MHR2 domain (Fig. 6e). When the NUDT9-H domain was fully loaded with ligands, the core region established three additional contacts with adjacent subunits (Extended Data Fig. 9c).

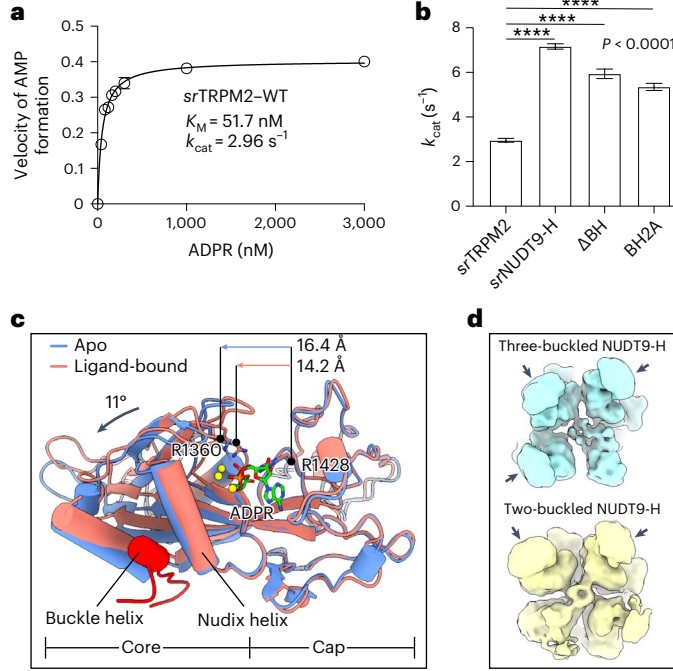

**Fig. 5 | The srNUDT9-H is a slow ADPR hydrolase. a**, Plot of the rate of AMP formation as a function of substrate concentration, representing the rate of ADPR hydrolysis by the WT srTRPM2. Each enzymatic reaction was performed three times independently and the converted ε-AMP was measured by fluorescence detection. The solid line indicates the fit to the Michaelis–Menten equation (n = 3) with Michaelis constant ($K_M$) indicated, and the circles and error bars represent mean ± s.d. **b**, Estimated $k_{cat}$ values for WT srTRPM2 (n = 3), isolated srNUDT9-H (n = 3), srTRPM2–ΔBH (n = 3) and srTRPM2–BH2A (n = 3) from the experiments in **a**. The bars and error bars represent mean ± s.d. and the P value was derived from two-tailed analysis. **c**, Superposition of NUDT9-H in the apo (predicted by AlphaFold) and ligand-bound states using the cap region of NUDT9-H. The NUDT9-H and buckle helix are shown in a cartoon representation. ADRP, R1360 and R1428 are shown as sticks. The movement of the core region of NUDT9-H on ligand release is manifested by an increased Cα distance between R1360 and R1428, as well as the rotation of the Nudix helix. **d**, Cytoplasmic view of the 3D classes in the cryo-EM data of ADPR being completely hydrolyzed, showing successive steps of detetramerization of the NUDT9-H domain. The unbuckled NUDT9-H domains disassociate from other subunits, thus becoming flexible and less well defined in the cryo-EM map. By contrast, the buckled NUDT9-H domains as well as the channel module remains well defined, and the channel module maintains C4 symmetry.

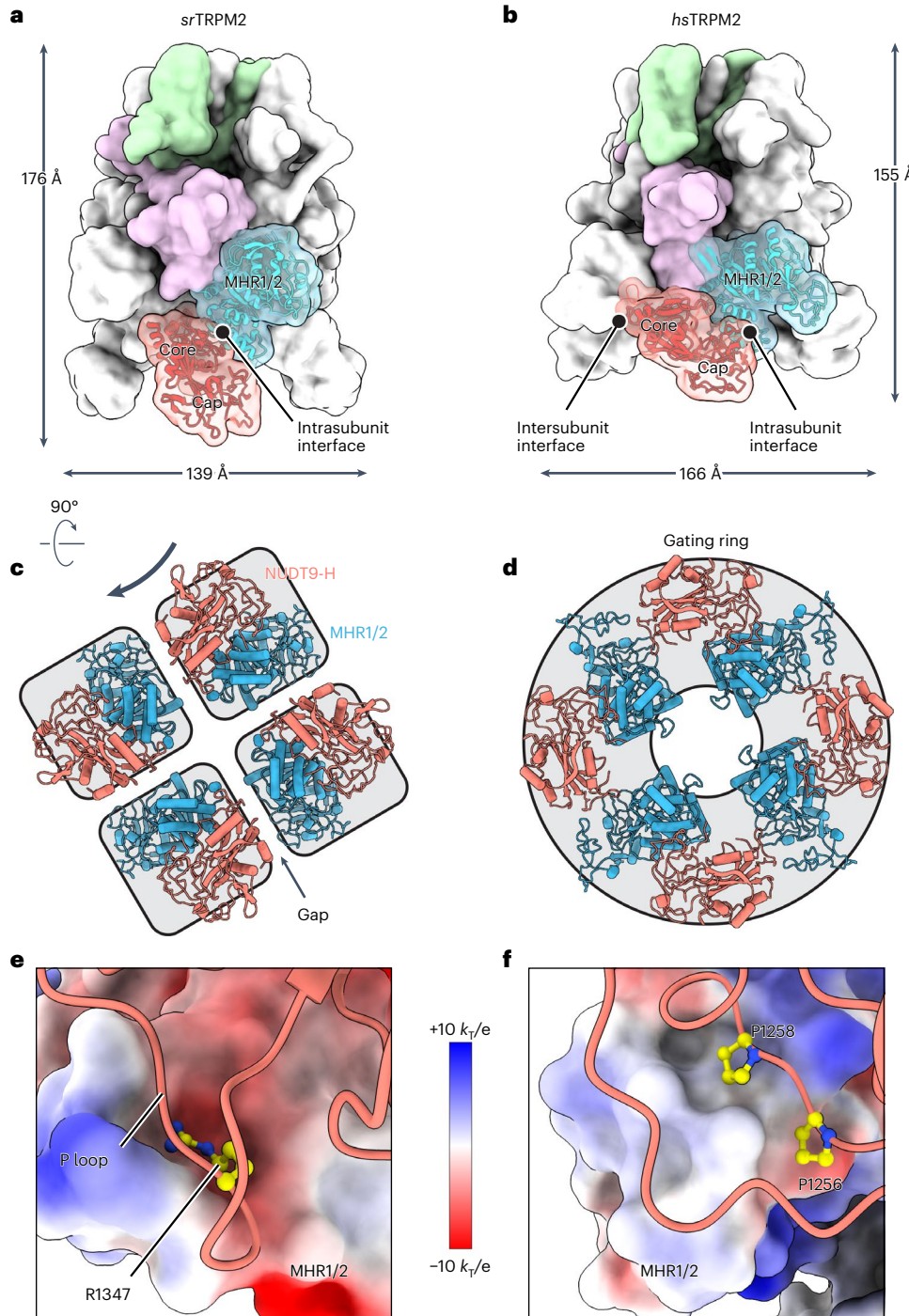

**Fig. 6 | Gain-of-function of the NUDT9-H domain as a gating module by forming a gating ring in vertebrate TRPM2. a,b**, The overall apo structures of *sr*TRPM2 (**a**) and *hs*TRPM2 (**b**) in surface representation viewed parallel to the membrane. One subunit is highlighted, with the four domains colored differently. The MHR1/2 and NUDT9-H domains are also shown in a cartoon representation. The core and cap regions of the NUDT9-H domain as well as its intrasubunit and intersubunit interfaces are indicated. **c,d**, Intracellular view of the organization of the ADPR-sensing layer in *sr*TRPM2 (**c**) and *hs*TRPM2 (**d**), consisting of the N-terminal MHR1/2 domain (blue) and C-terminal NUDT9-H domain (red). The black arrow represents a hypothetical movement of the NUDT9-H domain in *sr*TRPM2, which would be necessary to transition the ADPR-sensing layer in *sr*TRPM2 to a conformation similar to that of *hs*TRPM2. **e,f**, Intrasubunit interface between NUDT9-H and MHR1/2 domains in *sr*TRPM2 (**e**) and *hs*TRPM2 (**f**). The NUDT9-H domains are shown in a cartoon representation, with selected residues shown as sticks. The MHR1/2 domains are shown in a surface representation colored according to the electrostatic surface potential.

However, the cap region of the NUDT9-H domain, regardless of the ligand state, did not participate in any interactions with the channel module. This reinforces the notion that the NUDT9-H domain operates as a semi-independent module in *sr*TPRM2, thereby limiting its capacity to affect the conformation and gating of the channel module.

In contrast, the NUDT9-H domain in *hs*TPRM2 becomes a fully integrated part of the channel module by forming extensive interactions with the channel module through both core and cap regions (Fig. 6b), which is consistent with the reported insolubility of isolated *hs*NUDT9-H (ref. 36). Specifically, the intrasubunit interface is no longer

formed by the MHR2 domain with the core region, but with the cap region. The interactions also become more hydrophobic, partially contributed by a $^{1256}$P-V-P$^{1258}$ sequence of the cap region (Fig. 6f). The P-V-P sequence is conserved in vertebrate TRPM2, but is absent in early species including *sr*TRPM2 or *nv*TRPM2, where the corresponding residues and their surroundings are more hydrophilic (Supplementary Fig. 2). The core region, on the other hand, turns to interact with the adjacent subunit (Fig. 6b)—either the MHR1 domain in the apo state or the MHR2 domain in the $Ca^{2+}$/ADPR-bound state—through its positively charged P-loop. To accommodate these interactions, both the MHR1 and MHR2 domains in *hs*TRPM2 have evolved negative electrostatic surface potentials. In conclusion, the NUDT9-H domain forms extensive interfaces with the channel module in vertebrate TRPM2, forming a gating ring that allows local conformational changes induced by ADPR binding to propagate to the distal TMD and affect channel gating (Fig. 6d). This ADPR gating ring is reminiscent of the $Ca^{2+}$ gating ring in $Ca^{2+}$-gated potassium channel that uses the free energy of $Ca^{2+}$-binding to open the pore[37–39]. Our structural analysis thus supports that the role of the NUDT9-H domain in channel gating is obtained through the coevolution of the interfaces between the NUDT9-H and MHR1/2 domains.

## Discussion

Early species such as choanoflagellate *S. rosetta* are unicellular eukaryotes living in $Ca^{2+}$- and $Mg^{2+}$-rich seawater (~10 and 50 mM, respectively), thus facing much higher pressure on the $Ca^{2+}$ and $Mg^{2+}$ gradients than cells of advanced species. They are often exposed to ultraviolet radiation and hazardous chemicals, which can lead to oxidative stress and the accumulation of ADPR[40–42]. Perhaps to accommodate their harsh living environment, the choanoflagellate has developed a unique Nudix-linked TRPM protein, the TRPM chanzyme, that covalently links two functional modules with seemingly incompatible functions into a single peptide, a channel module activated by ADPR with high open probability and a Nudix enzyme module consuming ADPR at a remarkably slow rate. Activation of the channel module opens an ion permeation pathway for $Ca^{2+}$ and $Mg^{2+}$ influx, while the hydrolysis reaction in the enzyme module reduces the local ADPR concentration, thus closing the channel. Therefore, the enzyme module provides negative feedback that allows the channel module to cycle rapidly between open and closed states, avoid overloading of $Mg^{2+}$ and $Ca^{2+}$. This sophisticated channel–enzyme coupling is made possible by the ultra-slow kinetics of the enzyme module, which is ingeniously achieved through its integration with the channel module to form a chanzyme.

The Nudix-linked TRPM chanzyme is the only TRPM channel found in early species and is considered the evolutionary origin of the metazoan TRPM family, which contains eight members with diverse physiological roles. Although it shares similar overall architecture with TRPM2 channel in advanced species, TPRM2 channel in advanced species has evolved three main features that may be required for physiological functions specific to the advanced species. First, the enzyme module loses the ADPR hydrolase activity, which we showed is due to the evolution of key residues in the active center, resulting the loss of the $Mg^{2+}$ cofactors that is essential for hydrolase activity. Second, NUDT9-H in the TRPM2 channel becomes an ADPR-sensing module that forms a gating ring in conjunction with the N-terminal MHR1/2 domain, which is necessary for TRPM2 activation. A plausible structural basis for this functional evolution is that the loss of $Mg^{2+}$ cofactors may lead to a structural rearrangement of the binding site, forcing ADPR to adopt a different conformation, which in turn alters the way NUDT9-H couples to the channel module. Last, activation of TRPM2 channel becomes insensitive to $Mg^{2+}$ and strictly dependent on $Ca^{2+}$.

In summary, the Nudix-linked TRPM2 chanzyme exploits both channel module and enzyme module to control the channel activity. In contrast, TRPM2 in advanced species has evolved a strictly $Ca^{2+}$-dependent gating mechanism that incorporates desensitization[10],

thus eliminating the need for ADPR depletion to close the channel. We speculate that these differences can be attributed to the distinct environments that the cells inhabit. In advanced species such as humans, the intracellular concentration of free $Mg^{2+}$ remains relatively stable, whereas free $Ca^{2+}$ is dynamically regulated and is the most important cation in cellular signaling. Moreover, ADPR and its derivatives are important cellular metabolites involved in various physiological processes, many of which are found only in advanced species. Preserving ADPR, rather than depleting it for channel closure, reflects an adaptation to the metabolic needs of advanced organisms. Our study highlights the ability of organisms to adapt to their environments at the molecular level, and how the evolution of a protein such as TRPM2 can be shaped by the environment and physiological roles it serves in different species.

## Online content

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

## Methods

### Cell lines

Sf9 cells and tsA201 cells were purchased from the American Type Culture Collection. These cells were purchased and routinely maintained in our laboratory, and were not authenticated experimentally in this study. Cells were tested negative for mycoplasma contamination. No commonly misidentified cell lines were used.

### Phylogenetic analysis of TRPM2 channels

The amino acid sequences of putative TRPM2 channels were obtained by the basic local alignment search tool (BLAST)[43]. A local TRPM2 database was constructed with 130 predicted protein sequences including all the species from invertebrates, and representative ones from bony fish, birds and mammals. The multiple sequences were aligned using Clustl Omega[44] with default settings. The phylogeny was estimated by the maximum likelihood method and visualized by Interactive Tree Of Life[45]. The phylogenetic tree was manually colored and annotated.

### Constructs design and molecular cloning

Full-length *sr*TRPM2 gene (UniProtKB accession codes F2UB89) were synthesized by GenScript and subcloned into the pEG BacMam vector containing a TwinStrepII tag, an His$_8$ tag and enhanced green fluorescent protein (eGFP) with thrombin cleavage site at the N terminus of the gene[46]. The NUDT9-H domains of *sr*TRPM2 was subcloned into the same vector by Gibson Assembly[47] and the thrombin cleavage site was replaced by a tobacco etch virus (TEV) cleavage site. Mutants of buckle helix were generated by Gibson Assembly[47]. Single cysteine and double cysteine mutants used for cross-linking studies were generated by site-directed mutagenesis (Supplementary Table 1).

### *sr*TRPM2 expression and purification

The full-length *sr*TRPM2 and *sr*TRPM2–ΔNUDT9-H constructs were transformed into DH10Bac cells for bacmid generation. The purified bacmid was used to transfect Sf9 cells by Cellfectin II (ThermoFisher Scientific) for baculovirus production. P2 virus was used to infect tsA201 suspension cells that incubated at 37 °C. To boost the expression, sodium butyrate was added to a final concentration of 10 mM after 12 h postinfection and the temperature was decreased to 30 °C. The mammalian cells were collected and washed with cold buffer containing 20 mM Tris-HCl pH 8.0, 150 mM NaCl (TBS buffer) after 72 h postinfection.

The expressed cells were lysed in cold 10 mM Tris-HCl pH 8.0 buffer supplemented with 1 mM phenylmethylsulfonyl fluoride (PMSF), 2 mM pepstatin, 0.8 μM aprotinin and 2 μg ml$^{-1}$ leupeptin by gentle stirring. One hour after hypotonic lysis, Tris-HCl pH 8.0 and NaCl were added to final concentrations of 20 and 150 mM, respectively, and incubated for additional 10 min. Cell debris and unbroken cells were removed by centrifugation at 2,000$g$ for 10 min and membranes were collected by ultracentrifugation at 186,000$g$ for 1 h at 4 °C with a 45 Ti rotor (Beckman Coulter). The collected membrane was homogenized using a Dounce homogenizer in cold TBS buffer containing 1 mM PMSF, 2 mM pepstatin, 0.8 μM aprotinin, 2 μg ml$^{-1}$ leupeptin and 2 mM 2-mercaptoethanol (β-ME). The homogenized membrane was solubilized using 10 mM glycol-diosgenin (GDN) for 1 h at 4 °C by gentle stirring. Insolubilized debris was removed by ultracentrifugation at 186,000$g$ for 30 min. The supernatant was incubated with Talon resin (Clontech) for 2 h by gentle rotating at 4 °C and then washed with six bed volumes of TBS buffer containing 0.2 mM GDN, 10 mM imidazole and 2 mM β-ME. The protein was eluted with TBS buffer containing 0.2 mM GDN, 250 mM imidazole and 2 mM β-ME. The protein elution was concentrated and loaded onto a Superose 6 column (Cytiva) that was pre-equilibrated in buffer containing 20 mM Tris-HCl pH 8.0, 150 mM NaCl, 0.2 mM GDN, 5 mM β-ME (TBS-GDN buffer). The peak fractions were pooled and concentrated to 8–9 mg ml$^{-1}$ using a 100 kDa concentrator (Millipore) for cryo-EM studies.

For enzymatic studies, the elution from affinity resin was mixed with thrombin (HTI) with a mass ratio of 20:1 (*sr*TRPM2 elution:thrombin) and incubated at 4 °C for overnight. To remove the cleaved TwinStrepII-His$_8$-eGFP tag and uncleaved protein, the overnight sample was reapplied to Talon resin (Clontech). The flow-through was collected, concentrated and further purified by a Superose 6 column (Cytiva) in TBS-GDN buffer. The peak fractions were combined and concentrated for enzymatic studies.

### Nanodisc reconstitution

Brain total lipid extract (Avanti) in chloroform was dried using nitrogen gas and the chloroform residue was further removed by placing the dried lipids in a vacuum concentrator overnight (SpeedVac plus, Savant). The completely dried brain total lipid was resuspended into TBS-GDN buffer by bath sonication to a final concentration of 10 mM. The MSP2N2 scaffold protein was expressed and purified as previously described[48]. The eluted MSP2N2 protein from affinity resin was mixed with TEV protease at a mass ratio of 10:1 (MSP2N2:TEV) and dialysis against TBS buffer overnight. The overnight dialyzed sample was reapplied to affinity resin to remove the His-tag and uncleaved MSP2N2. The follow-through was collected and concentrated for use in nanodisc reconstitution.

The protein elution from affinity resin was directly used for nanodisc reconstitution. The resuspended brain total lipid was added to the protein elution and incubated for 30 min first. Then MSP2N2 was added and incubated for another 1 h. The final molar ratio of *sr*TRPM2:MSP2N2: Lipids was 1:4:600. Detergent was removed by adding preactivated Bio-Beads (SM2, Bio-Rad) to the final concentration of 100 mg ml$^{-1}$. After 2 h of gentle rotation at 4 °C, Bio-Beads were replaced with a new batch of beads and thrombin (HTI) was added to a final mass ratio of 20:1 (*sr*TRPM2 elution:thrombin). The mixture was further incubated overnight by gentle rotating. Following the removal of the Bio-Beads, the reconstituted sample was concentrated and loaded on a Superose 6 column (Cytiva) in TBS buffer supplemented with 5 mM β-ME. The peak fractions were collected and concentrated to 8–9 mg ml$^{-1}$ for cryo-EM studies.

### NUDT9-H domain expression and purification

Plasmid DNA with *sr*NUDT9-H domain was transformed into DH5α competent cells. The fresh transformed DH5α cell was expanded into a large-scale Luria-Bertani medium and further cultured for 12–16 h. Plasmid DNA was isolated to high purity using EndoFree Plasmid kits (Qiagen). Purified plasmid DNA was mixed with PEI 25K (Polysciences) in a mass ratio of 3:1 (PEI:DNA) and incubated at room temperature for 30 min. The PEI–DNA mixture was added to suspension tsA201 cells at 37 °C. After 16 h posttransfection, sodium butyrate was added to a final concentration of 10 mM and the temperature was decreased to 30 °C to boost the protein expression. Seventy-two hours after transfection, the mammalian cells were collected and washed with cold TBS buffer.

The collected mammalian cells were resuspended in a buffer composed of 20 mM Tris-HCl pH 8.0, 300 mM NaCl, 10% glycerol (v/v), 2 mM β-ME (buffer A). The cells were lysed with 10 mM *n*-dodecyl-β-D-maltoside (DDM) in the presence of 1 mM PMSF, 2 mM pepstatin, 0.8 μM aprotinin and 2 μg ml$^{-1}$ leupeptin by gentle stirring at 4 °C for 1 h. Cell debris and unbroken cells were removed by centrifugation at 186,000$g$ for 30 min. The supernatant was mixed with Talon resin for 2 h by gentle rotating and washed with buffer A supplemented with 1 mM DDM and 10 mM imidazole. The affinity resin was further washed with buffer A supplemented with 20 mM imidazole. The *sr*NUDT9-H domain was eluted with buffer A containing 250 mM imidazole. Protein elution was mixed with TEV protease in a mass ratio of 10:1 (*sr*NUDT9-H elution:TEV protease) and dialysis against TBS buffer supplemented with 2 mM β-ME overnight. The overnight dialyzed sample was reapplied to affinity resin to remove the His-tag and uncleaved *sr*NUDT9-H domain. The follow-through was collected,

concentrated and further purified by a Superdex 75 column (Cytiva) in TBS buffer containing 5 mM β-ME. The peak fractions were combined and concentrated for enzymatic studies.

## Enzymatic activity visualization by thin-layer chromatography

The enzymatic reaction mixtures including 20 mM ADPR, 0.44 µM purified full-length $sr$TRPM2, 16 mM $MgCl_2$ in TBS-GDN buffer were incubated at room temperature for 1 h. AMP and ADPR controls were treated the same way except that no enzyme was added. To visualize the inhibition effect of $CaCl_2$, a serial concentration of $CaCl_2$ (0.01, 0.03, 0.1, 0.3, 1, 3 and 10 mM) was coapplied with the $MgCl_2$. Then 2 µl of the reaction mixtures were spotted on Silica gel 60G F254 25 Glass Plates (EMD Millipore), dried and developed by solvents ethanol:water (70:30, v/v) supplemented with 200 mM $NH_4HCO_3$ (ref. 10). AMP and ADPR positions were visualized by ultraviolet light (ChemiDoc, Bio-Rad).

## Enzyme kinetic assay

Our primary studies have shown that $sr$TRPM2 has a very low $K_M$. To detect the low concentration of the AMP product, AMP was converted to 1,$N^6$-etheno-AMP ($\varepsilon$-AMP), which produces a fluorescence signal with an excitation wavelength of 230 nm and an emission wavelength of 410 nm, enabling the highly sensitive detection of AMP at nanomolar range[49,50]. Reversed-phase high-performance liquid chromatographic equipment with fluorescence detection was used to separate and detect $\varepsilon$-AMP.

$MgCl_2$ and ADPR were mixed first in TBS-GDN buffer ($MgCl_2$ at the concentration of 20 mM and ADPR at the concentrations of 80 nM, 160 nM, 240 nM, 320 nM, 400 nM, 600 nM, 2 mM and 6 mM), then 25 µl $MgCl_2$–ADPR mixture was added to 25 µl purified $sr$TRPM2 enzymes (0.2 nM $sr$TRPM2–WT, 0.1 nM $sr$TRPM2–BH2A, 0.1 nM $sr$TRPM2–ΔBH, 0.05 nM $sr$NUDT9-H) to initiate the reactions. The reactions were carried out at 25 °C controlled by a thermal cycler (T100, Bio-Rad) for 40 s for $sr$TRPM2–WT, $sr$TRPM2–BH2A, $sr$TRPM2–ΔBH and 30 s for $sr$NUDT9-H. The reactions were stopped by adding 75 µl freshly prepared solution containing 18.2 mM EDTA, 908.7 mM sodium acetate, 593.7 mM choroacetaldehyde, pH 4.5. The final mixture was incubated at 60 °C for 1 h for converting AMP to $\varepsilon$-AMP. The reaction was stopped by placing the PCR tubes on ice. The samples were centrifuged and 10 µl of samples were analyzed by a high-performance liquid chromatography machine (Shimadzu) equipped with an ACQUITY UPLC BEH Shield RP18 Column (130 Å, 1.7 µM, 2.1 × 150 mm, Waters). The mobile phase contains 10 mM ammonium acetate, 0.5% methanol, pH 5.0. $\varepsilon$-AMP was quantified using an excitation wavelength of 230 nm with an output emission wavelength of 410 nm.

## Disulfide cross-linking analysis

tsA201 cells were transfected with the WT, double cysteine mutants and the corresponding single cysteine mutants. After incubation at 37 °C for 16 h, sodium butyrate was added to a final concentration of 10 mM, and plates were moved to 30 °C to boost the expression. Forty-eight hours after transfection, the cells were collected and washed with cold TBS buffer. The washed cells were resuspended to TBS buffer supplemented with 5 mM β-ME. To induce the transient interaction between the buckle helix and NUDT9-H domain, $CaCl_2$ and ADPR were added to final concentrations of 1 and 1 mM, respectively. The cells were lysed with 10 mM GDN by gentle rotating at 4 °C for 2 h. Then the sample was cleared by centrifuge and the supernatant was mixed with 2× SDS-loading buffer (without any reducing agents) and analyzed by SDS–PAGE and in-gel fluorescence (Chemidoc, Bio-Rad).

## Electrophysiology

tsA201 cells were plated in 24-well plates and transfected using Lipofectamine 2000 (ThermoFisher) according to the manufacturer's

protocol. The transfected cells were incubated at 37 °C for 12–24 h before electrophysiological measurements. Patch-clamp recordings were acquired at 10 kHz using both a HEKA EPC-10 amplifier with Patchmaster software (HEKA) and a Multiclamp 700B with pCLAMP 11 (Molecular Devices), and were digitally filtered at 1 kHz.

For macroscopic recordings, glass pipettes were pulled to 4–6 MΩ and filled with an internal solution containing 10 mM HEPES, 150 mM NaCl, 3 mM KCl, pH 7.4 (adjusted by NaOH). Inside-out patches were pulled, and recordings were carried out at room temperature (~24 °C) with a holding potential of +60 mV. The bath solution was the same as the internal solution. A bath solution supplemented with ADPR (10 µM) alone, EDTA–ADPR (1 µM:1 mM), $CaCl_2$–ADPR (10:10 µM) or $MgCl_2$–ADPR (10:10 µM) was used for $sr$TRPM2 channel activation. The solution change was performed using a two-barrel theta-glass pipette controlled manually.

For single-channel recordings, glass pipettes were pulled to 10–16 MΩ and recordings were performed 3–8 h after transfection with the membrane potential clamped to +60 mV. Single-channel recordings were analyzed and open probability was determined using Nest-o-Patch v.2.1 (https://sourceforge.net/projects/nestopatch/) and Clampfit Software. Statistical analysis was done by GraphPad Prism (GraphPad Software) data were reported as mean ± s.d. and analyzed using an unpaired $t$-test.

## Electron microscopy sample preparation and data acquisition

The electron microscopy grids were prepared by a Vitrobot Mark VI held at 18 °C and 100% humidity. Freshly purified $sr$TRPM2 protein in GDN or nanodisc was mixed with EDTA (1 mM), $CaCl_2$ (1 mM), $MgCl_2$ (10 mM), EDTA–ADPR (1:1 mM), $CaCl_2$–ADPR (1:1 mM), $MgCl_2$–AMP–R5P (10:1:1 mM) or $MgCl_2$–ADPR (10:1 mM) and incubated for 30 min before grid preparation except for $MgCl_2$–ADPR conditions. For the $MgCl_2$–ADPR conditions, we performed time-resolved cryo-EM grid preparation by adding $MgCl_2$ directly to the mixture of $sr$TRPM2 and ADPR on the grid to initiate the enzyme reaction while controlling the timing. Using the Vitrobot Mark VI, the shortest time interval we could effectively manage was approximately 5 s. Therefore, we chose two time points, 5 and 10 s, in hopes of capturing intermediate states of the hydrolysis cycle. To capture the conformation of the NUDT9-H domain in the posthydrolyzed state, with only AMP and R5P bound, we extended the incubation time to 4 min. While this prolonged incubation led to most particles adopting an apo-like state with a flexible NUDT9-H domain, we were able to obtain a substantial number of particles that were refined to a high-resolution structure, with well-resolved NUDT9-H domains bound to the hydrolysis products AMP and R5P. In more detail, $MgCl_2$ was added to the preincubated $sr$TRPM2–ADPR mixture (30 min) and incubated for 5 s, 10 s or 4 min before the plunge-frozen. For nanodisc samples, 0.5 mM (1$H$,1$H$,2$H$, 2$H$-perfluorooctyl)-β-D-maltopyranoside (Anatrace) was added for improving particles distribution and contrast. Quantifoil holey carbon grids (gold, 2/1 µm size/hole space, 300 mesh) were glow-discharged for 30 s, then 2.5 µl of sample was applied to the carbon side of the grids and blotted for 1.5 s. The grids were plunge-frozen in liquid ethane cooled by liquid nitrogen.

Images were obtained using a FEI Titan Krios transmission electron microscope operating at 300 kV with a nominal magnification of 130,000. The $sr$TRPM2–WT–EDTA (GDN) and $sr$TRPM2–WT–Ca²⁺– ADPR (GDN) datasets were recorded by a Gatan K2 Summit direct electron detector in super-resolution mode with a binned pixel size of 1.074 Å. Each image was dose fractionated to 40 frames for 8 s with a total dose of 54.4 e⁻/Å⁻². The $sr$TRPM2–WT–EDTA–ADPR (GDN), $sr$TRPM2–WT–Ca²⁺ (GDN), $sr$TRPM2–WT–Mg²⁺ (GDN), $sr$TRPM2–WT– Mg²⁺–ADPR–4m (GDN) and $sr$TRPM2–E1114A–Mg²⁺–ADPR–5s (nanodisc) datasets were collected by a Gatan K3 direct electron detector in super-resolution mode with a binned pixel size of 0.826 Å. Each image was dose fractionated to 75 frames for 1.5 s with a total dose of

49 e$^-$/Å$^2$. The srTRPM2–WT–Mg$^{2+}$–ADPR–10s and srTRPM2–WT–Mg$^{2+}$–AMP–R5P datasets were collected by a Gatan K3 direct electron detector in super-resolution mode with a binned pixel size of 0.826 at Pacific Northwest Center for Cryo-EM. Each image was dose fractionated to 50 frames with a total dose of 50 e$^-$/Å$^2$. The images were recorded using the automated acquisition program SerialEM[51]. Nominal defocus values varied from −0.9 to −1.9 µm for the K3 camera, and −1.0 to −2.5 µm for the K2 camera. For the srTRPM2–ΔNUDT9-H constructs, the images were obtained using a FEI Talos Arctica transmission electron microscope operating at 200 kV. The data were recorded by a K2 direct electron detector operated in super-resolution mode with a binned pixel size of 1.16 Å. Each image was dose fractionated to 40 frames for 8 s with a total dose of 64.4 e$^-$/Å$^{-2}$. Nominal defocus values varied from −1.1 to −2.5 µM.

## Image processing

Super-resolution image stacks were motion-corrected, and 2 × 2 binned in Fourier space using MotionCor2 (ref. [52]). The values of the contrast transfer function (CTF) parameters were estimated by Gctf[53]. Particles were then picked by Gautomatch (https://github.com/JackZhang-Lab/Gautmatch), Relion autopicking[54] and Topaz[55] independently. Junk particles were removed by heterogeneous refinement in CryoSPARC[56]. The particles after cleanup were merged and deduplicated. Initial reconstruction was obtained using cryoSPARC. The deduplicated particles, together with the cryoSAPRC initial reconstruction, were submitted to Relion v.3.1 or v.4.0 (ref. [54]) for three-dimensional (3D) classification with C1 symmetry.

For the datasets in apo (EDTA) or partial ligands bound (Mg$^{2+}$, Ca$^{2+}$ and EDTA–ADPR) states, very dynamic NUDT9-H domains were observed. All the classes with good TMD, MHR1-4 domains were combined and refined with C4 symmetry. Then the particles were symmetry expanded and single subunits were analyzed by 3D classification with C1 symmetry. The single subunit classes with visual NUDT9-H domain were refined and then used to trace the tetrameric particles containing four copies of these subunits. Additionally, the quality of the single subunit map was further improved by multibody refinement and focused refinement on TMD plus MHR3/4 domain (body1) and MHR1/2 plus NUDT9-H domain (body2), respectively. Finally, the composite maps were generated by Phenix using the focused maps, single subunit consensus map and whole particle consensus map.

Similar strategies were used for the full ligands bound conditions, except that the NUDT9-H domain was further refined and used for generating the Phenix composite maps. To seperate the open and closed conformations in the Mg$^{2+}$–ADPR–10s dataset, we performed classification with C4 symmetry. Simultaneous binding of Mg$^{2+}$ or Ca$^{2+}$ with ADPR to the NUDT9-H domain increased its resolution compared to both the apo state and the partially ligand-bound states. However, the resolution of the NUDT9-H domain remained low in contrast to the rest of the protein, due to its distal location and the lack of extensive interactions with the rest of the protein. To improve the resolution of the NUDT9-H domain, thereby facilitating the discrimination of hydrolysis intermediates and aiding in de novo model building, we implemented additional steps. Specifically, we subtracted the NUDT9-H domain and proceeded with C1 symmetry-based refinement using all particles, followed by 3D classification without image alignment. We then conducted final refinements with selected classes representing different ligand-bound states. For Mg$^{2+}$–AMP–R5P data, the final refined particles from Relion after multiple rounds of CTF refinement and Bayesian polishing were further refined in cryoSPARC by nonuniform refinement and followed by two rounds of CTF refinement. The final consensus map has a resolution of 1.97 Å, which was validated by both cryoSAPRC and Relion. For the srTRPM2–ΔNUDT9-H constructs, initial 3D reconstruction was obtained using cryoSPARC and the selected particles from two-dimensional classification were subjected to 3D classification in Relion v.3.1, with the initial reconstruction low pass filtered to 50 Å as a reference model. Particles from classes showing high-resolution features were combined and refined with C4 symmetry in Relion v.3.1 and were further refined by CTF refinements and Bayesian polishing.

The detailed image processing steps were summarized and illustrated in Extended Data Fig. 2. For all datasets, the Gold-standard Fourier shell correlation (FSC) 0.143 criteria were used to provide the map resolution estimate.

## Model building

Models for srTRPM2 were built in Coot using zebrafish TRPM2 and human NUDT9 structures as references (PDB ID 6DRK and 1Q33, respectively)[14,30]. The initial models were then subjected to real-space refinement in Phenix with secondary-structure restraints[57]. The refined models were further manually examined and adjusted in Coot[58]. For validation of the refined structure, FSC curves were applied to calculate the difference between the final model and electron microscopy map by Phenix comprehensive validation (cryo-EM)[59]. The validation of atomic models was performed by MolProbity[60] in the Phenix suite[61]. All figures were prepared using UCSF ChimeraX[62] and PyMOL (https://pymol.org). The NUDT9-H domain in apo conformation was generated by AlphaFold[34] and rigid fitted into the cryo-EM maps.

## Reporting summary

Further information on research design is available in the Nature Portfolio Reporting Summary linked to this article.

## Data availability

The cryo-EM density maps and coordinates of srTRPM2–WT–apo, srTRPM2–WT–Ca$^{2+}$, srTRPM2–WT–Mg$^{2+}$, srTRPM2–WT–ADPR, srTRPM2–WT–Ca$^{2+}$–ADPR, srTRPM2–WT–Mg$^{2+}$–AMP–R5P, srTRPM2–WT–Mg$^{2+}$–ADPR–4m, srTRPM2–WT–Mg$^{2+}$–ADPR–10s_open, srTRPM2–WT–Mg$^{2+}$–ADPR–10s_closed_intact, srTRPM2–WT–Mg$^{2+}$–ADPR–10s_closed_hydrolyzed, srTRPM2–E1114A–Mg$^{2+}$/ADPR/5s_open, srTRPM2–E1114A–Mg$^{2+}$–ADPR–5s_closed, srTRPM2–ΔNUDT9-H–apo, srTRPM2–ΔNUDT9-H–Ca$^{2+}$–ADPR were deposited in the Electron Microscopy Data Bank (EMD) under the accession numbers EMD-40722, EMD-40724, EMD-40723, EMD-40725, EMD-40726, EMD-40721, EMD-40730, EMD-40727, EMD-40728, EMD-40729, EMD-40731, EMD-40732, EMD-40733 and EMD-40734 respectively. Atomic models for srTRPM2–WT–apo, srTRPM2–WT–Ca$^{2+}$, srTRPM2–WT–Mg$^{2+}$, srTRPM2–WT–ADPR, srTRPM2–WT–Ca$^{2+}$–ADPR, srTRPM2–WT–Mg$^{2+}$–AMP–R5P, srTRPM2–WT–Mg$^{2+}$–ADPR–4m, srTRPM2–WT–Mg$^{2+}$–ADPR–10s_open, srTRPM2–WT–Mg$^{2+}$–ADPR–10s_closed_intact, srTRPM2–WT–Mg$^{2+}$–ADPR–10s_closed_hydrolyzed, srTRPM2–E1114A–Mg$^{2+}$–ADPR–5s_open, srTRPM2–E1114A–Mg$^{2+}$–ADPR–5s_closed, srTRPM2–ΔNUDT9-H–apo, srTRPM2–ΔNUDT9-H–Ca$^{2+}$–ADPR were deposited in the Research Collaboratory for Structural Bioinformatics PDB under accession codes 8SR8, 8SRA, 8SR9, 8SRB, 8SRC, 8SR7, 8SRG, 8SRD, 8SRE, 8SRF, 8SRH, 8SRI, 8SRJ and 8SRK, respectively. Source data are provided with this paper.

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

## Acknowledgements

We thank G. Zhao and X. Meng for the support with data collection at the David Van Andel Advanced Cryo-Electron Microscopy Suite. We appreciate the high-performance computing team of VAI for computational support. W.L. is supported by National Institutes of Health (NIH) grants (nos. R01HL153219 and R01NS112363). J.D. is supported by a McKnight Scholar Award, a Klingenstein-Simon Scholar Award, a Sloan Research Fellowship in neuroscience, a Pew Scholar in the Biomedical Sciences award and an NIH grant (no. R01NS111031). A portion of this research was supported by NIH grant no. U24GM129547 and performed at the Pacific Northwest Center for Cryo-EM at Oregon Health & Science University (OHSU) and accessed through the Environmental Molecular Sciences Laborator (EMSL) (grid.436923.9), a Department of Environment Office of Science User Facility sponsored by the Office of Biological and Environmental Research.

## Author contributions

W.L. and D.J. supervised the project. Y.H. carried out all the experiments. S.K. participated in refining the atomic models. J.L. participated in single-channel recordings. Y.H., S.K., J.L., W.L. and J.D. analyzed the data. Y.H., W.L. and J.D. wrote the manuscript.

## Competing interests

The authors declare no competing interests.

## Additional information

**Extended data** is available for this paper at https://doi.org/10.1038/s41594-024-01316-4.

**Correspondence and requests for materials** should be addressed to Wei Lü or Juan Du.

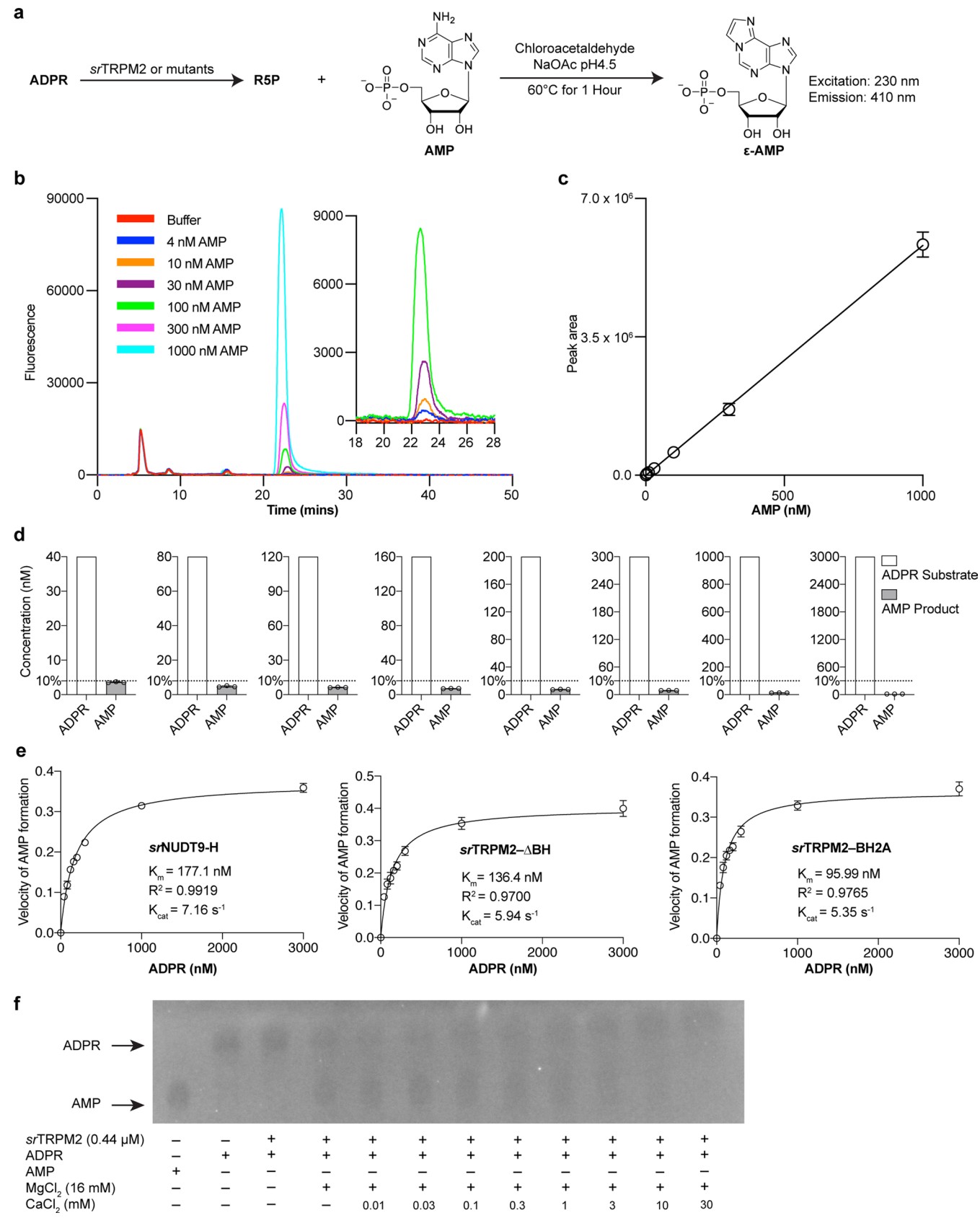

**Extended Data Fig. 1 | See next page for caption.**

**Extended Data Fig. 1 | Enzymatic assays of *sr*TRPM2 chanzyme. a**, Schematic diagram illustrating the principle of the enzymatic kinetic assay for ADPR hydrolysis by detecting the product AMP. In this assay, the AMP was converted to ε-AMP, which produces a fluorescence signal with an excitation wavelength of 230 nm and an emission wavelength of 410 nm, enabling the highly sensitive detection of AMP at nanomolar concentrations. Each enzymatic reaction was performed for three times independently and the converted ε-AMP was measured. **b**, **c**, Fluorescence detection of ε-AMP at various concentrations (**b**), with the area of the resulting peaks plotted in **c**. This generates a standard curve for ε-AMP, which shows a linear relationship between its concentration and the fluorescence signal (peak area) in the range of 4–1000 nM (n = 3). The circles and error bars represent mean ± s.d. **d**, To measure the initial velocity of the ADPR hydrolysis reaction, the reaction was stopped when less than 10% of the substrate has been converted to product, with substrate concentrations ranging from 40 to 3000 nM (n = 3). The bars and error bars represent mean ± s.d. **e**, Plot of the rate of AMP formation as a function of substrate concentration, representing the rate of ADPR hydrolysis by the isolated *sr*NUDT9-H domain, *sr*TRPM2 lacking the buckle helix (ΔBH), and *sr*TRPM2 with the buckle helix residues replaced by alanine (BH2A). The solid line indicates the fit to the Michaelis–Menten equation (n = 3), and the circles and error bars represent mean ± s.d. **f**, Thin layer chromatography analysis of the ADPR hydrolysis products by purified *sr*TRPM2. The reaction was performed in the presence of 16 mM $MgCl_2$ and different concentrations of $CaCl_2$, showing a seemingly dose-dependent inhibition of enzyme activity by $Ca^{2+}$.

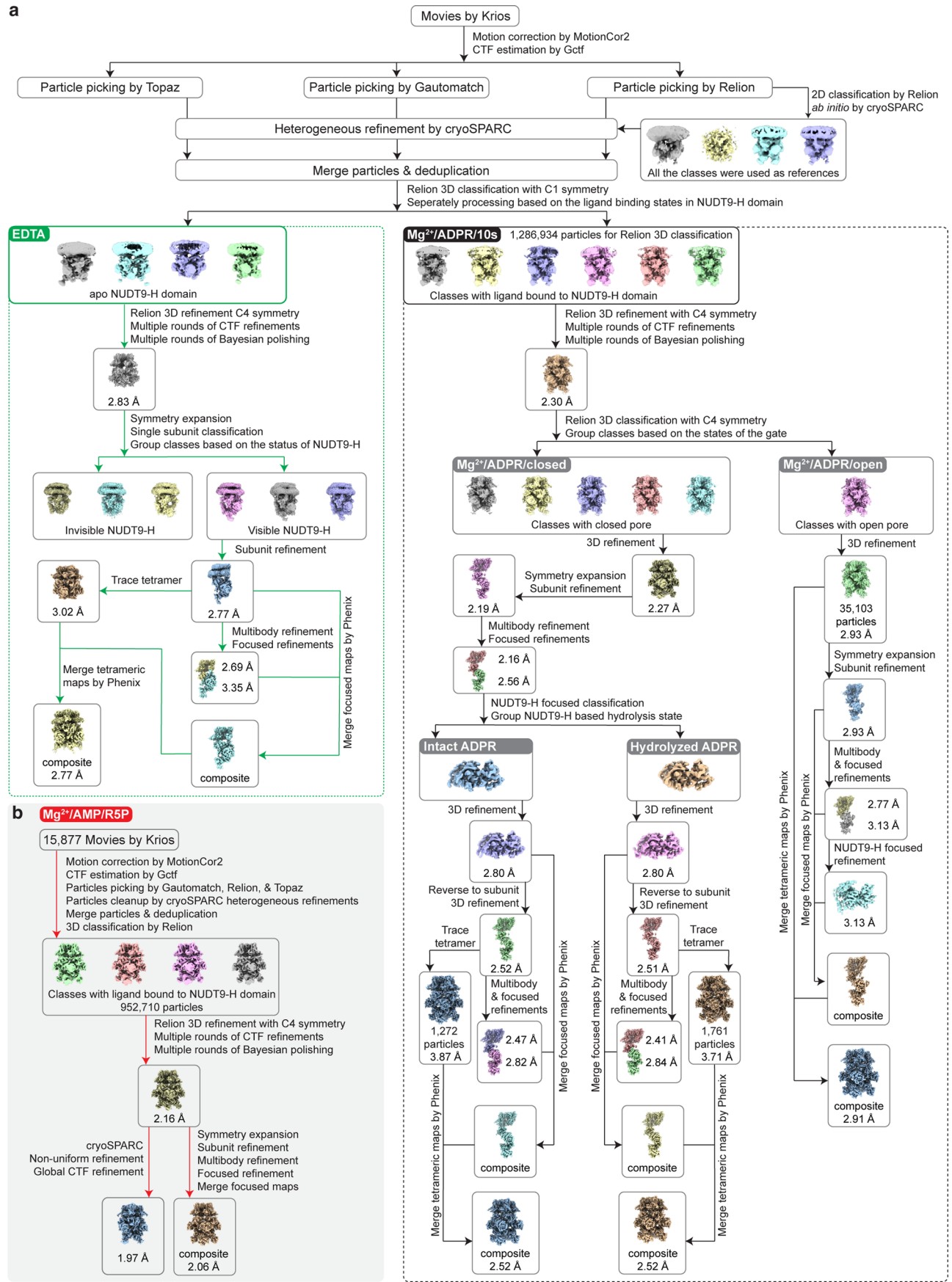

**Extended Data Fig. 2 | Workflow of cryo-EM data analysis. a**, Cryo-EM data analysis procedures using the apo (EDTA) and Mg²⁺/ADPR/10 s datasets as examples. **b**, Processing steps used to achieve sub-2 Å resolution for the Mg²⁺/AMP/R5P dataset. After the final consensus refinement in Relion, further refinement of the particles was carried out in cryoSPARC.

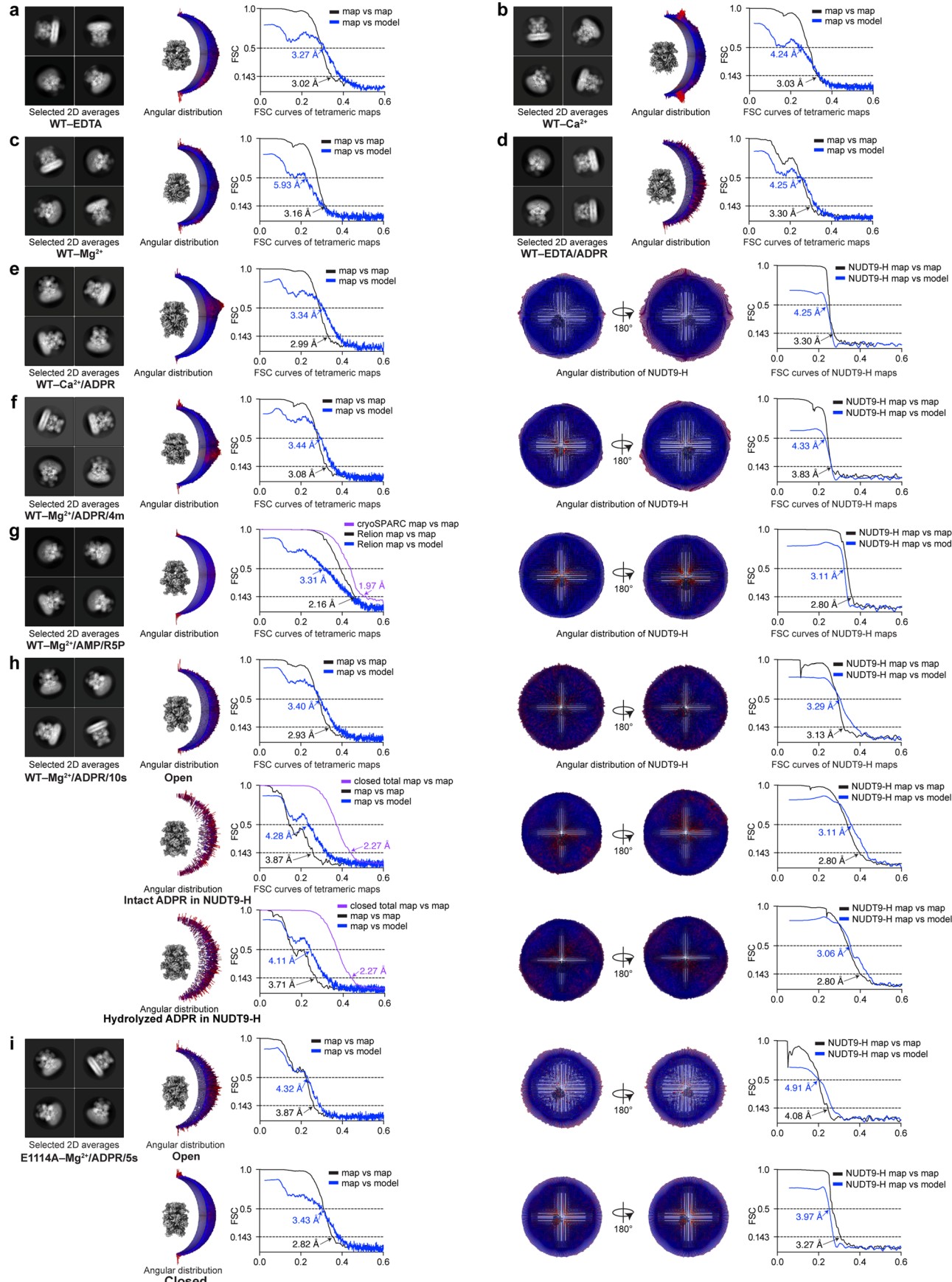

**Extended Data Fig. 3 | Cryo-EM data analysis.** Representative 2D class averages, angular distribution, and FSC curves of the data sets of WT-EDTA (**a**), WT-Ca²⁺ (**b**), WT-Mg²⁺ (**c**), WT-EDTA-ADPR (**d**), WT-Ca²⁺-ADPR (**e**), WT-Mg²⁺-ADPR-4m (**f**), WT-Mg²⁺-AMP-R5P (**g**), WT-Mg²⁺-ADPR-10s (**h**) and E1114A-Mg²⁺-ADPR-5s (**i**).

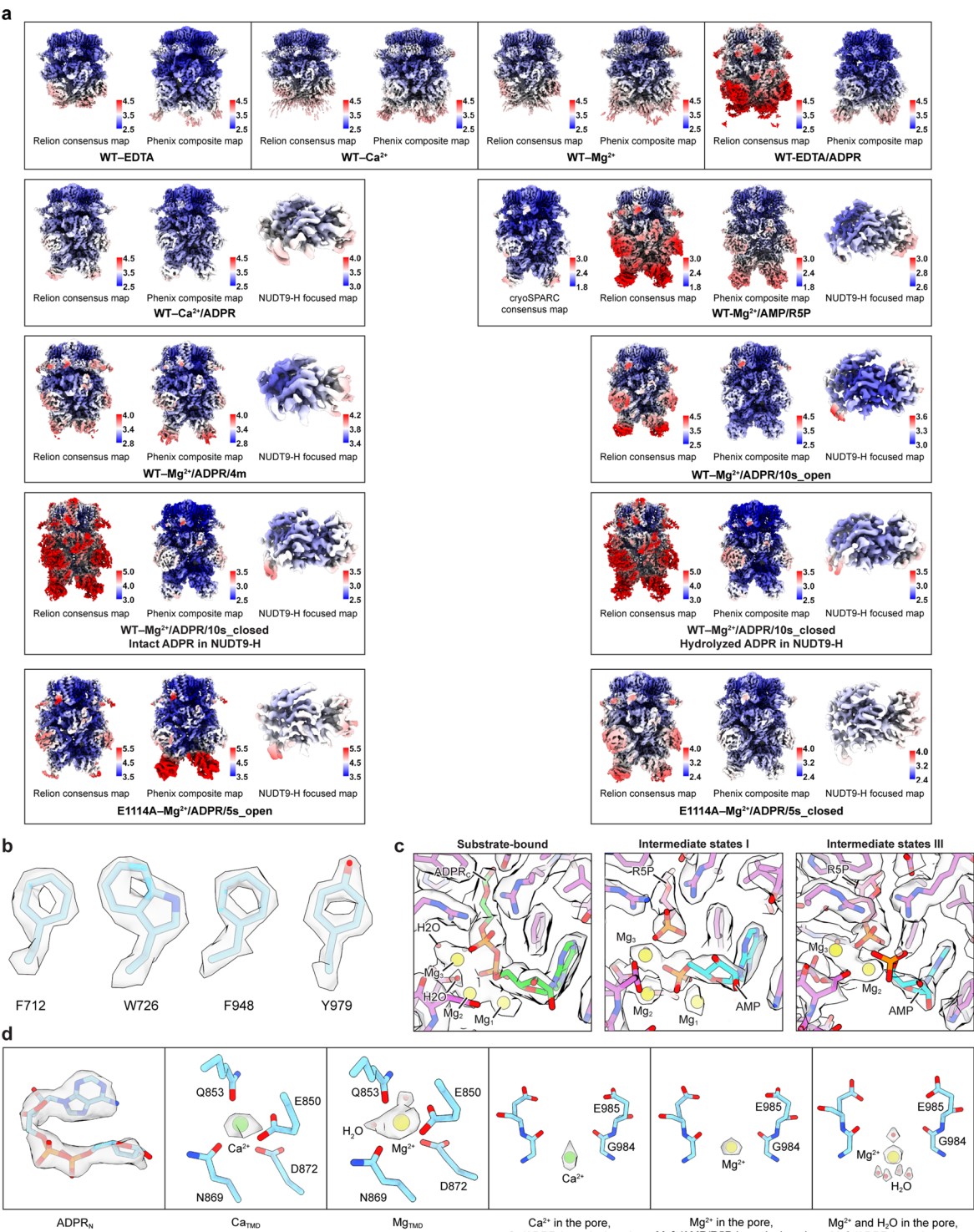

**Extended Data Fig. 4 | Local resolution estimation and representative densities. a**, Local resolution estimation of Relion consensus maps, Phenix composite maps, and NUDT9-H focused maps. The unit of the color bar is angstrom. **b**, Representative densities from the 1.97 Å Mg²⁺-AMP-R5P map.

**c**, Densities of ligand binding sites at NUDT9-H domain in substrate-bound, intermediate states I, and intermediate states III, respectively. **d**, Densities of the ligand binding sites including ADPR$_N$ in the MHR1/2 domain, Ca$_{TMD}$, Mg$_{TMD}$, as well as putative cations (Ca²⁺ or Mg²⁺) and water molecules in the ion conducting pore.

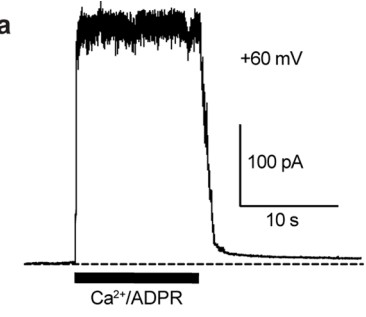

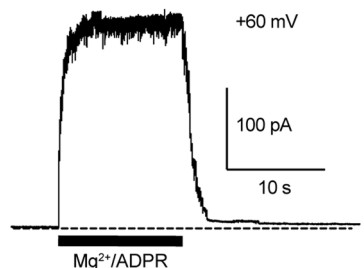

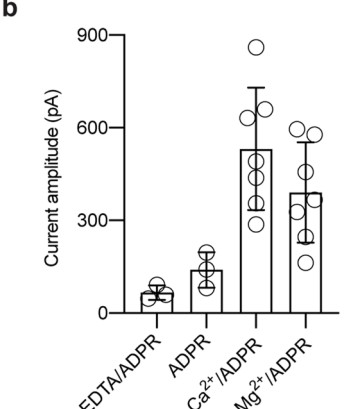

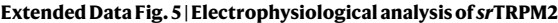

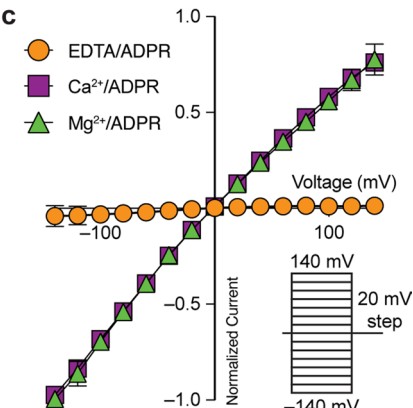

**Extended Data Fig. 5 | Electrophysiological analysis of *sr*TRPM2.**
**a**, Representative traces from membrane patches excised from tsA201 cells overexpressing *sr*TRPM2 in the presence of Ca²⁺/ADPR, and Mg²⁺/ADPR, respectively, recorded in the inside-out patch-clamp configuration at +60 mV. **b**, Current amplitudes of the experiments in **a**. Each point represents a single membrane patch from independent cells. The bars and error bars denote mean value and s.d., respectively. **c**, Normalized currents from membrane patches excised from tsA201 cells overexpressing *sr*TRPM2 in the presence of EDTA/ADPR, Ca²⁺/ADPR, and Mg²⁺/ADPR, respectively, recorded in the inside-out patch-clamp configuration. Voltage clamps were imposed from –140 to +140 mV in steps of 20 mV. The error bars represent s.d.

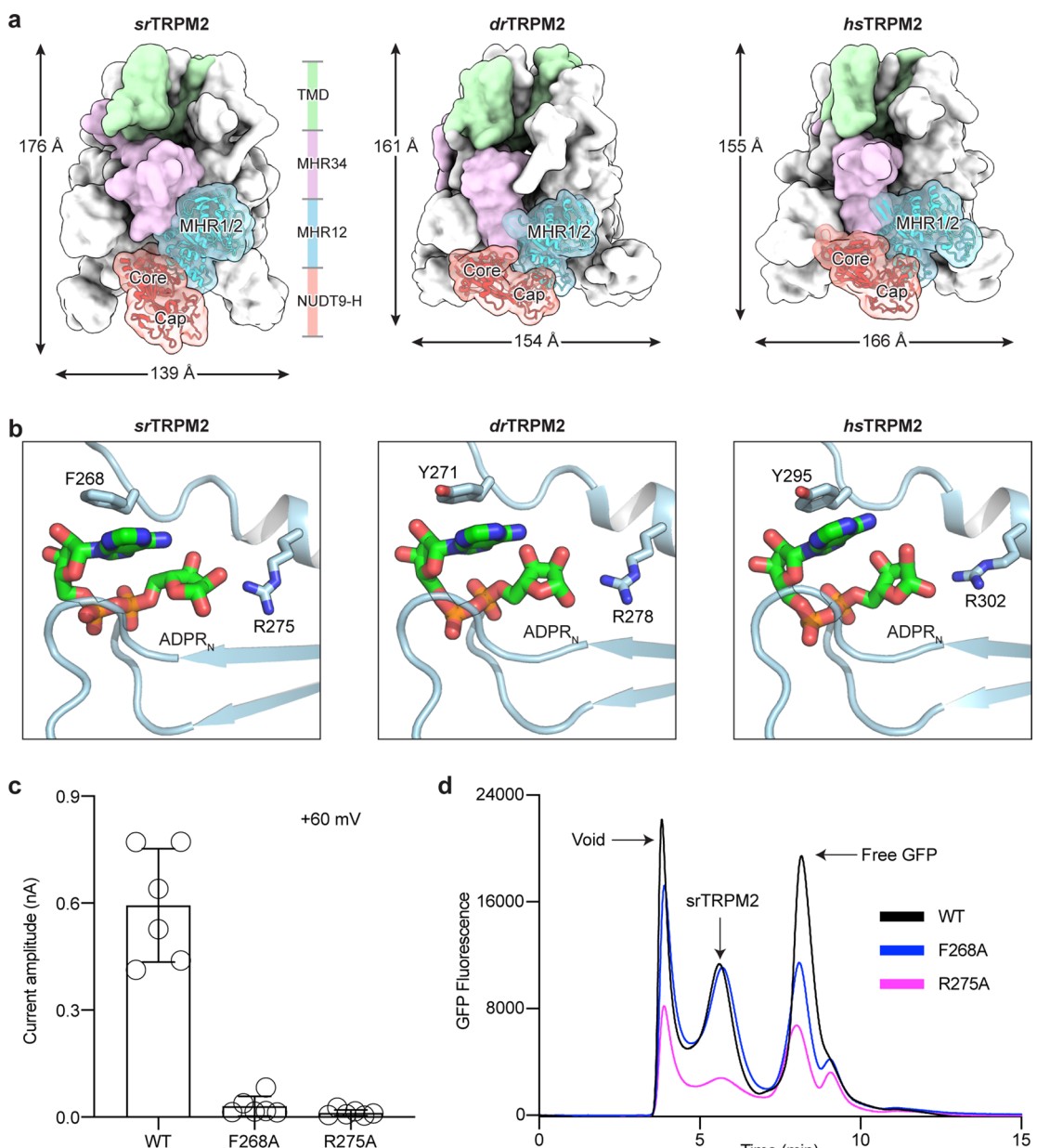

**Extended Data Fig. 6 | The conserved ADPR binding in MHR1/2 domain.**
**a**, The overall structures of *sr*TRPM2, *dr*TRPM2 (PDB ID: 6DRJ) and *hs*TRPM2
(PDB ID: 6PUS) in surface representation viewed parallel to the membrane. One
subunit is highlighted, with the four domains colored differently. The MHR1/2
and NUDT9-H domains are also shown in cartoon representation. **b**, The ADPR$_N$
binding site in the MHR1/2 domain of *sr*TRPM2, *dr*TRPM2 and *hs*TRPM2.
Two conserved key residues involved in ligand binding are shown in sticks.

**c**, Ca$^{2+}$/ADPR evoked current amplitudes from inside-out patch-clamp
measurements of wild-type *sr*TRPM2 and mutants of two conserved key residues
in the ADPR$_N$ binding site, F268A and R275A (n = 6 cells). The bars and error bars
represent mean ± s.d. **d**, Expression profile of wild-type *sr*TRPM2 (black), F268A
(blue), and R275A (pink). While F268A has a similar expression level compared to
wild type, the R275A mutant has a reduced expression level.

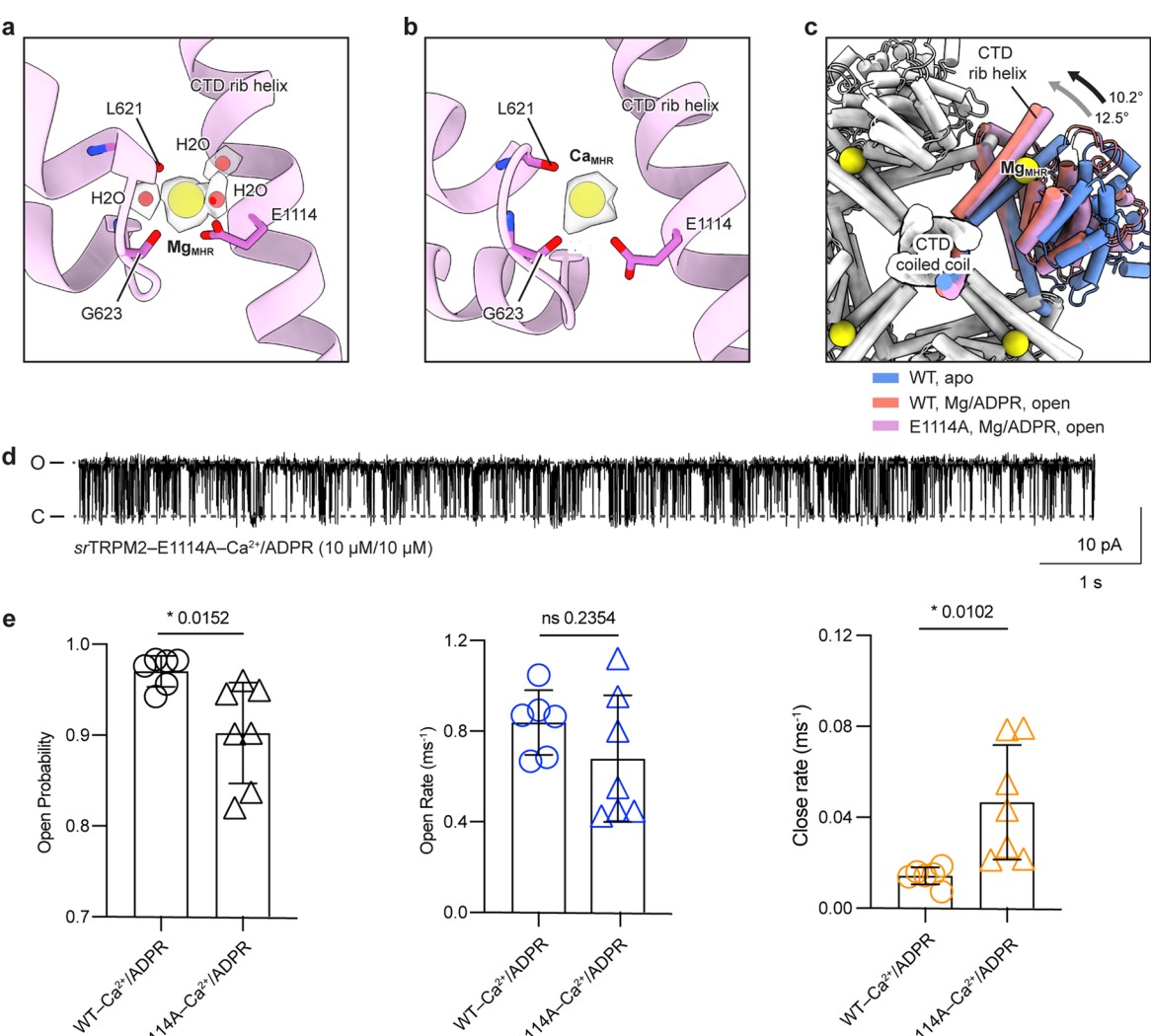

**Extended Data Fig. 7 | Structural and electrophysiological analysis of the cation binding site between the MHR4 domain and rib helix. a, b**, Cryo-EM density of $Mg_{MHR}$ and coordinating water molecules (**a**), and $Ca_{MHR}$ (**b**). The cations and water molecules are shown in yellow and red spheres, respectively. Key residues involved in cation binding are shown in sticks. **c**, Binding of $Mg^{2+}$ (or $Ca^{2+}$) to this site facilitates the rotational movement of the MHR3/4 domain and rib helix induced by the binding of ADPR. This is evidenced by the observation that the $Mg_{MHR}$-deficient mutant, E1114A, caused a smaller rotational movement (the black arrow) when bound with $Mg^{2+}$ (or $Ca^{2+}$) and ADPR than

the wild-type (the gray arrow) relative to the apo structure. The structures are superimposed using the C-terminal coiled-coil. The MHR3/4 domain, pole helix and C-terminal coiled-coil are shown in cartoon. The $Mg_{MHR}$ cations are shown in spheres. **d**, Example traces of inside-out patch recordings of $sr$TRPM2–E1114A (n = 7, each lasting 40–110 seconds) clamped at +60 mV. **e**, Single channel kinetics of $sr$TRPM2 WT and E1114A mutant. Each point represents a single measurement from independent cells. The channel kinetics are reported as mean ± s.d. and $p$ value was derived from two-tailed analysis.

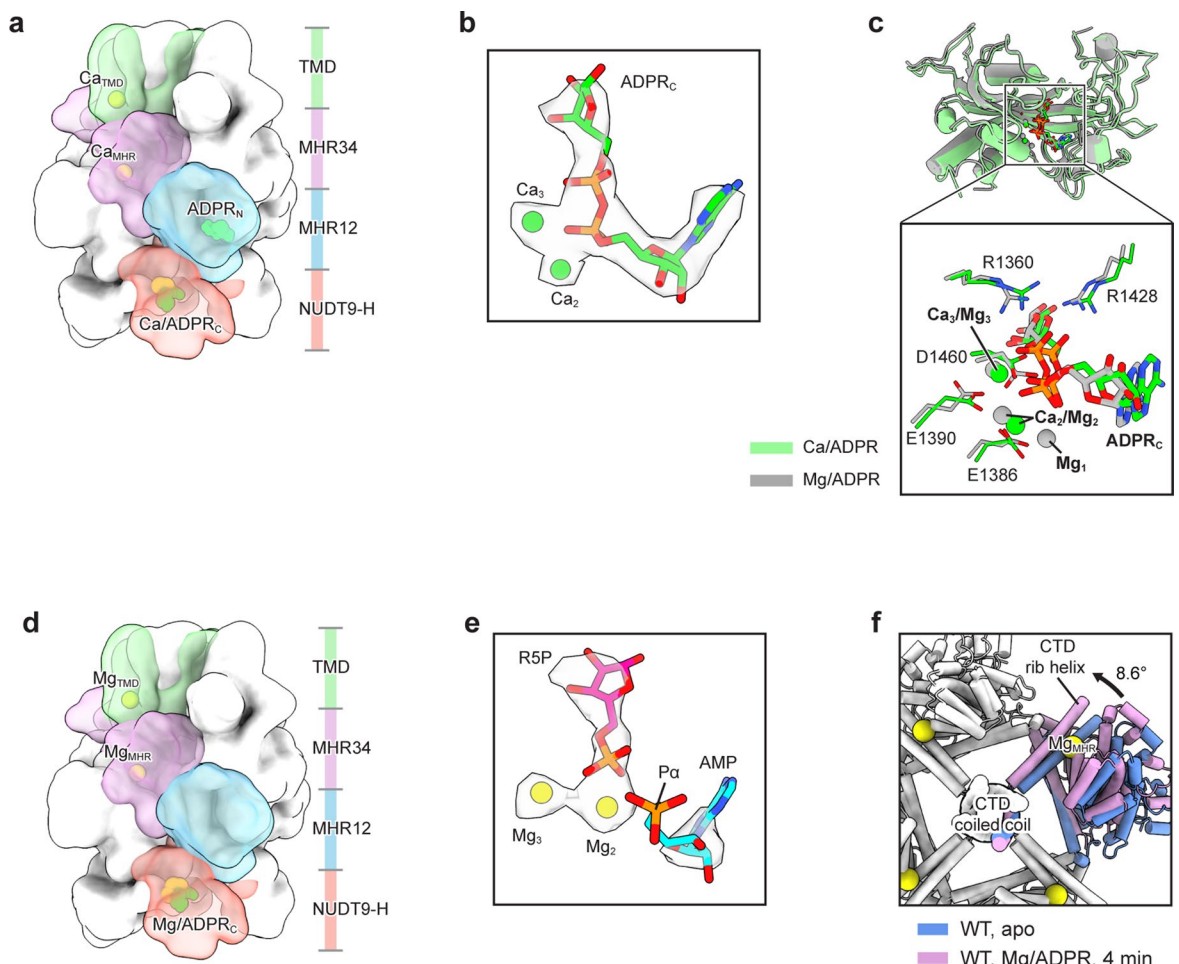

**Extended Data Fig. 8 | Structures analysis of *sr*TRPM2 bound with Ca²⁺/ADPR (a–c), as well as *sr*TRPM2 incubated with Mg²⁺/ADPR for 4 minutes (d–f).** **a**, The overall structure in surface representation viewed parallel to the membrane. One subunit is highlighted, with the four domains colored differently and the ligands shown as spheres. **b**, Cryo-EM densities of the ADPR$_C$ and two bound Ca²⁺ cations in NUDT9-H. The ADPR molecule is shown in sticks, while the Ca²⁺ cations are shown in spheres. **c**, Comparison of the NUDT9-H domains bound with Ca²⁺/ADPR (green) and Mg²⁺/ADPR (gray), respectively, with a zoomed-in view of the active site shown in the black box. The ADPR molecules and surrounding residues are shown in sticks, while the cations are shown in spheres. **d**, The overall structure in surface representation viewed parallel to the membrane. One subunit is highlighted, with the four domains colored differently and the ligands shown as spheres. The ADPR molecules were depleted, resulting in the binding of hydrolysis products to the NUDT9-H domain and an empty MHR1/2 domain. The overall structure showed high similarity to the structure of *sr*TRPM2 incubated directly with the hydrolysis products, AMP and R5P, along with Mg²⁺. **e**, Cryo-EM densities of AMP, R5P, and two Mg²⁺ cofactors (Mg$_2$ and Mg$_3$) in the active site of NUDT9-H. The ADPR molecule is shown in sticks, while the Mg²⁺ cofactors are shown in spheres. The cofactor Mg$_1$, which coordinates the α-phosphate (Pα) group, was not observed and was likely released after hydrolysis, causing Pα to become flexible. **f**, Ligand binding to NUDT9-H resulted a rotational movement of the MHR3/4 domain and rib helix relative to the apo structure, positively modulating channel activation. The structures are superimposed using the C-terminal coiled-coil. The MHR3/4 domain, pole helix and C-terminal coiled-coil are shown in cartoon. The Mg$_{MHR}$ cations are shown in spheres.

**a**

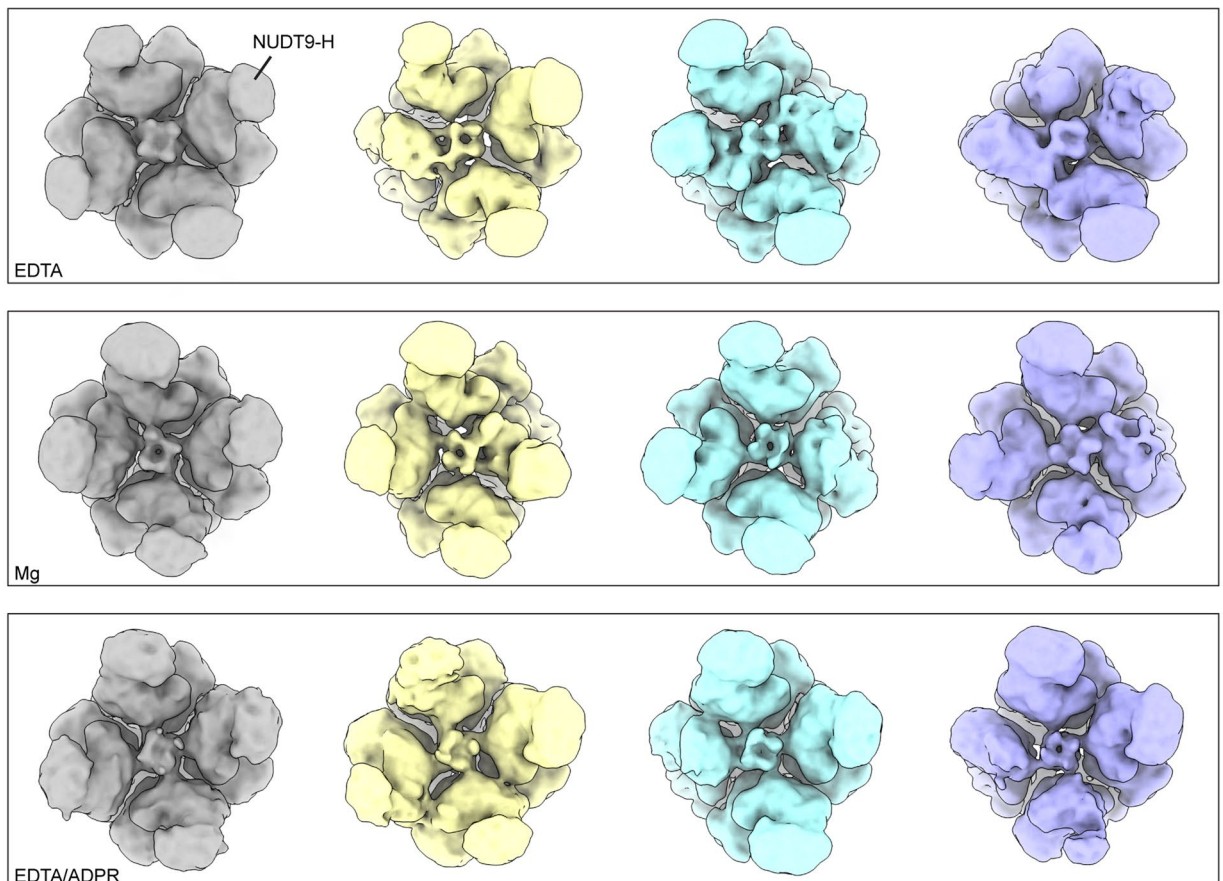

**b**

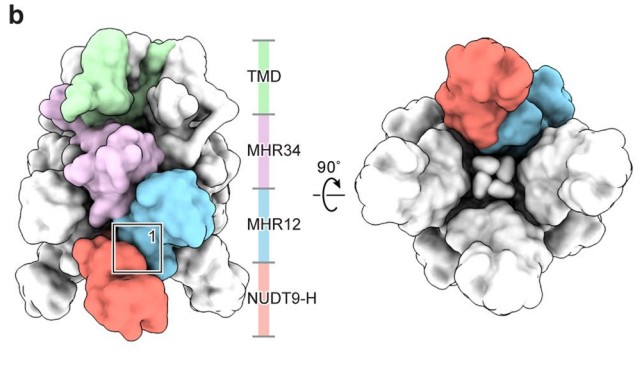

Apo

**c**

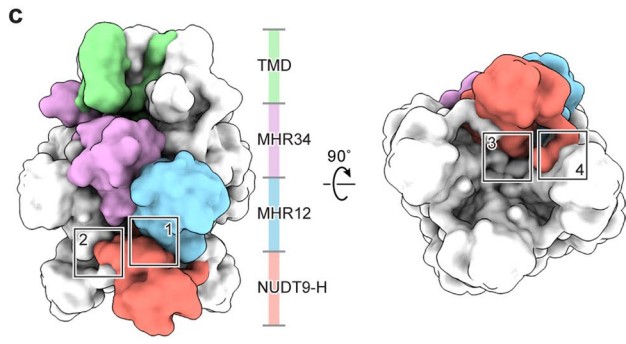

Mg/ADPR-bound

**Extended Data Fig. 9 | The interaction of the NUDT9-H domain with the channel module depends on the ligand-bound state. a**, The NUDT9-H domain is flexible in the apo or partially ligand-bound states, as demonstrated by the substantial heterogeneity of the NUDT9-H domain in the classed obtained from 3D classification. The ligand condition is indicated in the lower left corner of each row. The NUDT9-H domains in the four classes, from left to right, are increasingly disordered. **b**, In the apo or partially ligand-bound states, the NUDT9-H domain forms only one interface with the MHR1/2 domain of the same subunit (marked with a black box). **c**, Upon full ligand binding to the NUDT9-H domain, it forms three additional interfaces with cognate and adjacent subunits (marked with black boxes), reducing its flexibility.

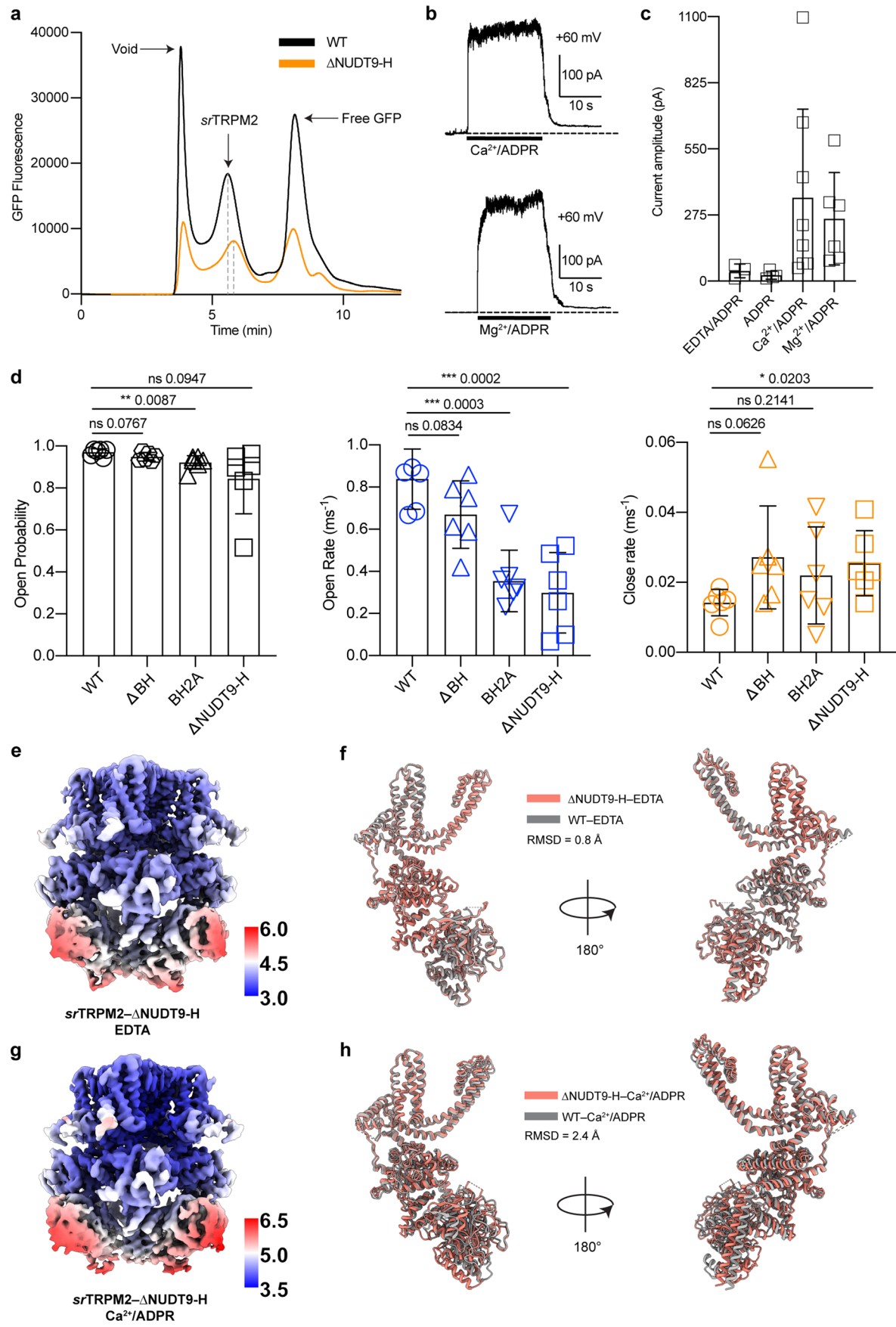

**Extended Data Fig. 10 | See next page for caption.**

**Extended Data Fig. 10 | Electrophysiological and structural analysis of *sr*TRPM2–ΔNUDT9-H. a**, Expression of wild-type *sr*TRPM2 (black) and *sr*TRPM2–ΔNUDT9-H (orange) analyzed by fluorescence size-exclusion chromatography (FSEC) analysis. The elution profile of *sr*TRPM2–ΔNUDT9-H is shifted to the right relative to the wild type (gray dashed lines), consistent with its smaller molecular weight. **b**, Representative traces from membrane patches excised from tsA201 cells overexpressing *sr*TRPM2–ΔNUDT9-H in the presence of $Ca^{2+}$/ADPR, and $Mg^{2+}$/ADPR, respectively, recorded in the inside-out patch-clamp configuration at +60 mV. **c**, Current amplitudes of the experiments in **b**. Each point represents a single membrane patch from independent cells. The bars and error bars denote mean value and s.d., respectively. **d**, Singel channel kinetics of *sr*TRPM2–WT, *sr*TRPM2–ΔBH, *sr*TRPM2–BH2A, *sr*TRPM2–ΔNUDT9-H. Each point represents a single measurement from independent cells. The channel kinetics are reported as mean ± s.d. and the *p* value was derived from two-tailed analysis. **e, g**, Unsharpened cryo-EM maps of *sr*TRPM2–ΔNUDT9-H in the apo state (**e**) and bound with $Ca^{2+}$/ADPR (**g**), colored by local resolution. The color bar indicates resolution in angstrom (Å). **f, h**, Superimpositions of single subunits of wild-type *sr*TRPM2 and *sr*TRPM2–ΔNUDT9-H in the apo state (**f**) and bound with $Ca^{2+}$/ADPR (**h**). The NUDT9-H domain in the wild-type *sr*TRPM2 structure is not shown.

# Reporting Summary

## Statistics

For all statistical analyses, confirm that the following items are present in the figure legend, table legend, main text, or Methods section.

| n/a | Confirmed | |
|---|---|---|
| ☐ | ☒ | The exact sample size (n) for each experimental group/condition, given as a discrete number and unit of measurement |
| ☐ | ☒ | A statement on whether measurements were taken from distinct samples or whether the same sample was measured repeatedly |
| ☐ | ☒ | The statistical test(s) used AND whether they are one- or two-sided *Only common tests should be described solely by name; describe more complex techniques in the Methods section.* |
| ☒ | ☐ | A description of all covariates tested |
| ☒ | ☐ | A description of any assumptions or corrections, such as tests of normality and adjustment for multiple comparisons |
| ☐ | ☒ | A full description of the statistical parameters including central tendency (e.g. means) or other basic estimates (e.g. regression coefficient) AND variation (e.g. standard deviation) or associated estimates of uncertainty (e.g. confidence intervals) |
| ☐ | ☒ | For null hypothesis testing, the test statistic (e.g. F, t, r) with confidence intervals, effect sizes, degrees of freedom and P value noted *Give P values as exact values whenever suitable.* |
| ☒ | ☐ | For Bayesian analysis, information on the choice of priors and Markov chain Monte Carlo settings |
| ☒ | ☐ | For hierarchical and complex designs, identification of the appropriate level for tests and full reporting of outcomes |
| ☒ | ☐ | Estimates of effect sizes (e.g. Cohen's d, Pearson's r), indicating how they were calculated |

*Our web collection on statistics for biologists contains articles on many of the points above.*

## Software and code

Policy information about availability of computer code

| Data collection | SerialEM 3.7, Patchmaster 2x90.5 |
|---|---|
| Data analysis | Gctf-1.06, Gautomatch-0.56, Relion-3.1, Relion-4.0, MolProbity4.4, CryoSparc-v3.0, MotionCorr2-1.4.0, topaz v0.2.4, Phenix v1.19.1-4122, Coot-0.9.8.1, UCSF chimera_1.13.1, UCSF chimeraX_0.91, PyMol v2.5, AlphaFold v2.1.2, GraphPad Prism 7, Nest-o-Patch 2.1. |

For manuscripts utilizing custom algorithms or software that are central to the research but not yet described in published literature, software must be made available to editors and reviewers. We strongly encourage code deposition in a community repository (e.g. GitHub). See the Nature Portfolio guidelines for submitting code & software for further information.

## Data

Policy information about availability of data

All manuscripts must include a data availability statement. This statement should provide the following information, where applicable:
- Accession codes, unique identifiers, or web links for publicly available datasets
- A description of any restrictions on data availability
- For clinical datasets or third party data, please ensure that the statement adheres to our policy

The cryoEM density maps and coordinates of srTRPM2–WT–apo, srTRPM2–WT–Ca2+, srTRPM2–WT–Mg2+, srTRPM2–WT–ADPR, srTRPM2–WT–Ca2+/ADPR, srTRPM2–WT–Mg2+/AMP/R5P, srTRPM2–WT–Mg2+/ADPR/4m, srTRPM2–WT–Mg2+/ADPR/10s_open, srTRPM2–WT–Mg2+/ADPR/10s_closed_intact, srTRPM2–WT–Mg2+/ADPR/10s_closed_hydrolyzed, srTRPM2–E1114A–Mg2+/ADPR/5s_open, srTRPM2–E1114A–Mg2+/ADPR/5s_closed, srTRPM2–ΔNUDT9H–apo, srTRPM2–

ΔNUDT9H–Ca2+-ADPR were deposited in the EMDB (Electron Microscopy Data Bank) under the accession numbers EMD-40722, EMD-40724, EMD-40723, EMD-40725, EMD-40726, EMD-40721, EMD-40730, EMD-40727, EMD-40728, EMD-40729, EMD-40731, EMD-40732, EMD-40733, and EMD-40734 respectively. Atomic models for srTRPM2–WT–apo, srTRPM2–WT–Ca2+, srTRPM2–WT–Mg2+, srTRPM2–WT–ADPR, srTRPM2–WT–Ca2+/ADPR, srTRPM2–WT–Mg2+/AMP/R5P, srTRPM2–WT–Mg2+/ADPR/4m, srTRPM2–WT–Mg2+/ADPR/10s_open, srTRPM2–WT–Mg2+/ADPR/10s_closed_intact, srTRPM2–WT–Mg2+/ADPR/10s_closed_hydrolyzed, srTRPM2–E1114A–Mg2+/ADPR/5s_open, srTRPM2–E1114A–Mg2+/ADPR/5s_closed, srTRPM2–ΔNUDT9H–apo, srTRPM2–ΔNUDT9H–Ca2+-ADPR were deposited in the RCSB PDB (Research Collaboratory for Structural Bioinformatics Protein Data Bank) under accession codes 8SR8, 8SRA, 8SR9, 8SRB, 8SRC, 8SR7, 8SRG, 8SRD, 8SRE, 8SRF, 8SRH, 8SRI, 8SRJ, and 8SRK respectively.

# Research involving human participants, their data, or biological material

Policy information about studies with [human participants or human data](). See also policy information about [sex, gender (identity/presentation), and sexual orientation]() and [race, ethnicity and racism]().

| | |
|---|---|
| Reporting on sex and gender | N/A |
| Reporting on race, ethnicity, or other socially relevant groupings | N/A |
| Population characteristics | N/A |
| Recruitment | N/A |
| Ethics oversight | N/A |

Note that full information on the approval of the study protocol must also be provided in the manuscript.

# Field-specific reporting

Please select the one below that is the best fit for your research. If you are not sure, read the appropriate sections before making your selection.

☒ Life sciences ☐ Behavioural & social sciences ☐ Ecological, evolutionary & environmental sciences

For a reference copy of the document with all sections, see [nature.com/documents/nr-reporting-summary-flat.pdf]()

# Life sciences study design

All studies must disclose on these points even when the disclosure is negative.

| | |
|---|---|
| Sample size | Sample sizes were determined based on preliminary experiments and prior experiences of the investigators. Cryo-EM data has been collected in sufficient quantity to enable reconstruction at a resolution of 4.2 Å or higher. Macroscopic electrophysiology measurements were performed on at least three different cells and single channel measurements were performed on at least six different cells. |
| Data exclusions | No data were excluded from analysis |
| Replication | The number of biologically independent experimental replications is indicated in the figure legend. |
| Randomization | In cryo-EM data processing, particles were randomly allocated to even and odd groups during refinement and resolution estimation using gold-standard Fourier shell correlation (FSC). In electrophysiology experiments, there was no predetermined order of experimentation and the cells used for patching are selected randomly. |
| Blinding | The investigators were not blinded when performing the experiment as it is important to use channels as a control for monitoring factors such as transfection efficiency and cell health, which could influence measurements. |

# Reporting for specific materials, systems and methods

We require information from authors about some types of materials, experimental systems and methods used in many studies. Here, indicate whether each material, system or method listed is relevant to your study. If you are not sure if a list item applies to your research, read the appropriate section before selecting a response.

## Materials & experimental systems

| n/a | Involved in the study |
|---|---|
| ☒ | ☐ Antibodies |
| ☐ | ☒ Eukaryotic cell lines |
| ☒ | ☐ Palaeontology and archaeology |
| ☒ | ☐ Animals and other organisms |
| ☒ | ☐ Clinical data |
| ☒ | ☐ Dual use research of concern |
| ☒ | ☐ Plants |

## Methods

| n/a | Involved in the study |
|---|---|
| ☒ | ☐ ChIP-seq |
| ☒ | ☐ Flow cytometry |
| ☒ | ☐ MRI-based neuroimaging |

## Eukaryotic cell lines

Policy information about cell lines and Sex and Gender in Research

| | |
|---|---|
| Cell line source(s) | Sf9 cells and tsA201 cells were purchased from ATCC. |
| Authentication | These cells were purchased and routinely maintained in our lab, and were not authenticated experimentally in this study. |
| Mycoplasma contamination | sf9 cells and tsA201 cells were tested negative for Mycoplasma contamination. |
| Commonly misidentified lines (See ICLAC register) | No commonly misidentified cell lines were used. |

## Plants

| | |
|---|---|
| Seed stocks | *Report on the source of all seed stocks or other plant material used. If applicable, state the seed stock centre and catalogue number. If plant specimens were collected from the field, describe the collection location, date and sampling procedures.* |
| Novel plant genotypes | *Describe the methods by which all novel plant genotypes were produced. This includes those generated by transgenic approaches, gene editing, chemical/radiation-based mutagenesis and hybridization. For transgenic lines, describe the transformation method, the number of independent lines analyzed and the generation upon which experiments were performed. For gene-edited lines, describe the editor used, the endogenous sequence targeted for editing, the targeting guide RNA sequence (if applicable) and how the editor was applied.* |
| Authentication | *Describe any authentication procedures for each seed stock used or novel genotype generated. Describe any experiments used to assess the effect of a mutation and, where applicable, how potential secondary effects (e.g. second site T-DNA insertions, mosiacism, off-target gene editing) were examined.* |

