## [Peer Review File · Nature Structural & Molecular Biology]

Peer Review Information

Manuscript Title: Coupling enzymatic activity and gating in an ancient TRPM channel and its molecular evolution

Corresponding author name(s): Juan Du, Wei Lü

Reviewer Comments & Decisions:

Decision Letter, initial version:
--

Message: 6th Jul 2023

Dear Dr Du,

Thank you again for submitting your manuscript "Coupling enzymatic activity and gating in an ancient TRPM channel and its molecular evolution". [I apologize for the delay in responding, which resulted from the difficulty in obtaining suitable referee reports. Nevertheless,] we now have comments (below) from the 3 reviewers who evaluated your paper. In light of those reports, we remain interested in your study and would like to see your response to the comments of the referees, in the form of a revised manuscript.

You will see that while reviewers appreciate the results, they raise several concerns which will need to be addressed in a revision. Specifically, in line with reviewers' comments, we would like to ask that the manuscript is made accessible to a wider audience, considering non-TRP channel experts, and structural data presentation improved. As both reviewers #1 and #2 point out, we would expect the electrophysiology experiments revisited, to provide quantification and additional controls where needed. Following to this, we agree with reviewer #3 that longer patch-clamp recordings should be provided to elucidate the effects of ADPR hydrolysis, if feasible. We also think that further investigation into the gating kinetics would be interesting. We agree with reviewer #1 that further discussion on the role of bound cholesterol should be considered, as well as investigation into other lipids, if feasible. However, we do not consider these points essential in the context of the current work.

Please be sure to address/respond to all concerns of the referees in full in a point-by-point response and highlight all changes in the revised manuscript text file.

We appreciate the requested revisions are extensive. We thus expect to see your revised manuscript within 6 months. If you cannot send it within this time, please let us know. We will be happy to consider your revision as long as nothing similar has been accepted for publication at NSMB or published elsewhere. Should your manuscript be substantially delayed without notifying us in advance and your article is eventually published, the received date would be that of the revised, not the original, version.

Reporting Summary:

When submitting the revised version of your manuscript, please pay close attention to our [href="https://www.nature.com/nature-portfolio/editorial-policies/image-integrity">Digital Image Integrity Guidelines](https://www.nature.com/nature-portfolio/editorial-policies/image-integrity). and to the following points below:

Please note that all key data shown in the main figures as cropped gels or blots should be presented in uncropped form, with molecular weight markers. These data can be aggregated into a single supplementary figure. While these data can be displayed in a relatively informal style, they must refer back to the relevant figures.

SOURCE DATA: we request that the authors provide, in tabular form, the data underlying the graphical representations used in figures. This is to further increase transparency in data reporting, as detailed in this editorial (<http://www.nature.com/nsmb/journal/v22/n10/full/nsmb.3110.html>). Spreadsheets can be submitted in excel format. Only one (1) file per figure is permitted; thus, for multi-paneled figures, the source data for each panel should be clearly labeled in the Excel file; alternately the data can be provided as multiple, clearly labeled sheets in an Excel file. When submitting files, the title field should indicate which figure the source data pertains

to. We encourage our authors to provide source data at the revision stage, so that they are part of the peer-review process.

We require deposition of coordinates (and, in the case of crystal structures, structure factors) into the Protein Data Bank with the designation of immediate release upon publication (HPUB). Electron microscopy-derived density maps and coordinate data must be deposited in EMDB and released upon publication. Deposition and immediate release of NMR chemical shift assignments are highly encouraged. Deposition of deep sequencing and microarray data is mandatory, and the datasets must be released prior to or upon publication. To avoid delays in publication, dataset accession numbers must be supplied with the final accepted manuscript and appropriate release dates must be indicated at the galley proof stage. Please find the complete NRG policies on data availability at <http://www.nature.com/authors/policies/availability.html>.

Nature Structural & Molecular Biology is committed to improving transparency in authorship. As part of our efforts in this direction, we are now requesting that all authors identified as 'corresponding author' on published papers create and link their Open Researcher and Contributor Identifier (ORCID) with their account on the Manuscript Tracking System (MTS), prior to acceptance. This applies to primary research papers only. ORCID helps the scientific community achieve unambiguous attribution of all scholarly contributions. You can create and link your ORCID from the home page of the MTS by clicking on 'Modify my Springer Nature account'. For more information please visit please visit www.springernature.com/orcid.

<https://mts-nsmb.nature.com/cgi-bin/main.plex?el=A6J2CnG1A3vCo5J4A9ftdfW1zxkOq974Vq5Z3UX64AZ>

Sincerely,

Katarzyna Ciazynska
(she/her)
Associate Editor
Nature Structural & Molecular Biology
<https://orcid.org/0000-0002-9899-2428>

Referee expertise:

Referee #1: structural biology, TRP channels

Referee #2: structural biology, TRP channels, evolution

Referee #3: TRP channel regulation, structural and molecular biology

Reviewers' Comments:

Reviewer #1:

Remarks to the Author:

Huang et al. provides an insightful view on TRPM2, from structural and evolutionary perspectives. Structural biology of membrane proteins is key, and regarding TRPM channels, it is special because of the dual mechanism of action combining ion channel and enzymatic activities. In this case, this manuscript represents an excellent piece of structural biology work, to comprehend a complex functional mechanism. In fact the manuscript is elegantly prepared and the data strongly supports the conclusions. Certain aspects need clarification and/or further discussion, but the manuscript could be published in present form in NSMB. Specifics that could be addressed to benefit the manuscript in the present form are:

1. The structures show 3 well-defined cholesterol densities per subunit close to the conducting pore region. These densities are present in all 14 structures, suggesting a structural or regulatory role on the channel. This is an interesting observation, however the author do not discuss further on the relevance of this lipid in the mentioned site. In addition, since the structural biology workflow is exhaustive, the presence of differential lipid densities (PIP2, PCs, etc.) in the different maps could be analyzed further in order to define lipid regulatory aspects, if any, on TRPM2.
2. Regarding the novel cation binding site characterization, the changes in channel opening probability and open rate for both mutants E1114A and E1114Q are not significant (Figure S9D). This should be clearly stated in the manuscript, focusing on that the only significative difference is in the channel closed state.
3. Lines 200-203. The authors could discuss further (briefly) what changes, and the importance of these changes, are relevant to to avoid hydrolysis of ADPR, as it is done later for the differences in ADPR binding between srTRPM2 and hsTRPM2 in lines 315-318.
4. Figure 4E seems to show a more significant effect on the channel gating for BH2A and Δ BH mutants of the buckle helix than for the Δ NUDT9-H mutant. The authors suggest that the buckle helix stabilizes the NUDT9-H tetramerization. It seems that this helix might have a more central role in the channel gating regulating, according to these results. Moreover, it is said the buckle helix mutants are expected to weaken the tetramerization, which might account for the long-lived closed states observed in Figure 4, but more concise data analysis should be given to support this claim, since it is quite an important issue, and the number of structures in the present manuscript should suffice to build a stronger argument regarding conformational changes vs. tetramerization of this domain.
5. Line 1014. Figure 9 should be Figure S9

Reviewer #2:

Remarks to the Author:

This is an interesting structural and functional study aimed at understanding how divalent ions and ADPR regulate the activity of TRPM2 channels. Two ADPR molecules bind to distinct sites on the channel, where one appears to be required for channel activation and the other can have an enzymatic action wherein it hydrolyzes ADPR to AMP and R5P. The authors focus on an ancient version of TRPM2 from choanoflagellates that has this enzymatic function to solve structures under 11 different conditions to understand the regulatory interplay between these various sites. Although I am inclined to believe the conclusions advanced by the authors, the presentation is quite dense and will be hard for many to follow unless they work on TRPM2 channels. The following are critically important points the authors should address in revision.

1) If I understand correctly, the authors have solved structures using grids that were prepared using 11 different conditions, all of which were refined using different programs or symmetry constraints and in some cases using different populations of particles. Table S1 is quite confusing in places and needs to be better organized to be readily comprehensible to readers. I suggest using vertical lines to distinguish same conditions or unique conformations. This will help for all four pages of the table but is essential for the last two. Page 3 is almost incomprehensible. The description of what was done should also be expanded in the main text. As it stands now the reader needs to dig through the table to understand and even that is challenging. It would also be helpful if the authors could indicate which refinement was used for different purposes, including formulating different conclusions and making different figures.

2) For all of the key structural findings the authors need to prepare figures using not just cryo-EM density for the ion or ligand, but also for the surrounding protein so that readers can readily judge the quality of the maps and conclusions. As one example, Fig 3B and D look reasonable when only ion and ligand density are shown (although the assignment of water is highly questionable!!), but I am not so sure that will be true once density is included for the surrounding protein given the modest resolution for most of the maps. Showing equivalent views for important control structures like apo would also help to give a much clearer impression of the quality of maps and conclusions. I view this as a systematic issue that needs to be addressed and the authors should find ways to prepare figures that will convince the most skeptical of readers. I am willing to give the authors the benefit of the doubt at this stage, but I shouldn't have to when this is something that can be addressed with a more rigorous presentation.

3) In many cases (Fig 1C,2G,4E) the authors show what appear to be single channel records from one recording without any quantification or presentation of population data. These need to be expanded to include more information on how recordings were obtained and for how long, whether any evidence was seen for multiple channels in a patch, all points histograms to clearly present the underlying complexity and then population data to demonstrate reproducibility. In at least one place the authors make statements about opening and closing kinetics without any presentation of data analysis, how this was determined and how many times it was replicated.

4) I think the authors need to say more about the role of the NUDT9-H domain in regulating channel activation. Including more carefully analyzed data will help, but I can

tell whether ADPR binding to that domain has much of an impact when the NUDT9-H domain is deleted. Is that domain really just there to lower the local concentration of ADPR or does it really influence channel activity. At least to me this wasn't clear from the current presentation.

Reviewer #3:

Remarks to the Author:

In this manuscript, Huang et al use time-resolved cryo-EM to capture 14 structures of a choanoflagellate TRPM2-like channel enzyme in distinct gating and ADPR hydrolysis catalytic states. Some of their structural insights are tested by inside-out patch single channel recordings of wild type and mutant srTRPM2 channels. The authors combine these results to propose a binary model wherein ADPR and cation binding to the MHR1/2 and NUDT9-H domains induce enzymatic domain tetramerization and channel gating while slow ADPR hydrolysis by the NUDT9-H domain reduces local ADPR concentration to close the channel. Finally, the authors compare their srTRPM2 structures to their previously solved human TRPM2 structures to propose an evolution of the NUDT9-H domain from a weak ADPR hydrolase in early species to an enzymatically dead, critical gating module in higher organisms. Despite the impressive number of structures, the authors currently overinterpret their structural data to propose gating and regulatory mechanisms that are not supported by and are poorly integrated with their functional and enzymatic work. For example, the authors claim throughout the manuscript that ADPR binding to and hydrolysis by the NUDT9-H allow rapid switching between the open and closed states to avoid divalent cation overload. However, the authors fail to show whether and how ligand binding to the NUDT9-H domain or substrate hydrolysis accompanies gating-relevant structural transitions. They also fail to correlate their calculated k_{cat} values (seconds) to gating kinetics (milliseconds) to demonstrate that ADPR hydrolysis occurs on a timescale relevant to channel closing. Moreover, the manuscript is currently inaccessible to non-TRPM2 experts as the introduction and discussion sections assume readers are readily familiar with the role and regulation of TRPM2 in vertebrates. In its current form, this manuscript and its findings are not suitable for publication in Nature Structural & Molecular Biology. To strengthen their conclusions, this reviewer recommends the authors address the following major points:

1. The timescales of functional recordings and structural snapshots are poorly integrated. The single channel recordings provide acute effects of ADPR, cations, and mutations on channel function whereas the effects of ADPR hydrolysis by the NUDT9-H domain on channel function inferred from structures will occur on longer timescales. Thus, the authors should include longer recordings where ADPR can be depleted by the NUDT9-H domain. Such recordings would be best conducted in cell-attached or whole cell configuration with endogenous induction of ADPR production versus exogenous application of ADPR and cations to patches. These recordings would also allow the authors to demonstrate whether and how mutations or deletions of the NUDT9-H domain and its tetramerization affect channel activity (to be included in Fig. 4). Does the NUDT9-H domain-mediated ADPR hydrolysis promote channel closing on a physiologically relevant timeframe that would, indeed, prevent divalent cation overload? Does disruption to ADPR hydrolysis result in prolonged channel activation?
2. Relatedly, if ADPR depletion by the NUDT9-H domain hydrolysis sets channel closure, then a TRPM2 "chanzyme" with faster hydrolysis kinetics would be expected to exhibit

enhanced gating kinetics than srTRPM2. The authors should compare the gating kinetics of srTRPM2 to those of TRPM2 orthologues with faster ADPR hydrolysis kinetics.

3. The single channel recordings presented throughout the manuscript should include baseline currents before agonists and cations were applied to show the quality of the recordings. Additionally, histograms to show open probability and dwell times of their replicates should be included. Moreover, the authors should justify why only 3 replicates were performed for patching experiments; literature precedence is 6-8. P0 values should be included for all single channel recordings (currently they are only included in Fig. 1C). Finally, a representative recording or the histogram profiles for WT srTRPM2 channels with Ca/ADPR should be included in Fig. 2G and 4E to allow for better comparison of changes to activity with mutant channels.

4. Their model in Fig. 3A and corresponding hypothesis proposes that reduction of ADPR levels by ADPR hydrolysis leads to ADPR departure from the MHR1/2 sites as opposed to the NUDT9-H domain extracting ADPR from the MHR1/2 site. The authors should explain how is this consistent with srTRPM2 being a chanzyme as opposed to a canonical ligand-gated ion channel wherein agonist departure returns the channel to a closed state (their leading sentences in the abstract).

5. The model in Fig. 3A assumes that the MHR1/2 site has a higher affinity for ADPR than the NUDT9-H domain since the NUDT9-H can be devoid of substrate or products while the MHR1/2 site retains substrate binding. The authors should determine the binding affinity for MHR1/2 and the NUDT9-H domains for ADPR to qualify their model.

6. The authors currently provide an understated explanation for how ADPR binding to MHR1/2 and NUDT9-H lead to channel gating despite having 14 distinct structures. How is ligand binding transmitted to the pore? Include structural overlays to show conformational changes associated with ligand binding from the apo to ligand bound states (apo versus Mg/ADPR-bound, closed) and conformational changes from the closed to open state (Mg/ADPR-bound closed versus Mg/ADPR-bound open). How does enzymatic activity return the channel to the closed state (Intermediate state I or III versus apo state)?

7. The structural landscape presented in Fig. 1D is confusing and subtle conformational changes are difficult to discern in the current presentation. This would benefit from being updated to a model as in Fig. 3A and/or by highlighting which structural features change between neighboring structures. This could be even better strengthened by indicating which structures are compared in subsequent figures and subsections. Additionally, the authors should be clear which structures are used in all figures.

8. The crosslinking data in Fig. 4D needs molecular weight markers to demonstrate confidence that the higher band is the tetramer. Their srTRPM2 construct has at least 1414 residues and a GFP tag, thus a monomer should be ~170 kDa. A tetramer would be 680 kDa, which is close to the resolution limit by SDS-PAGE. Most common molecular weight ladders have the highest standard of 250 kDa. How many times was this experiment repeated? Did the cysteine pairs crosslink without ADPR and calcium addition? Could the authors use these cysteine pair mutants for functional studies to ask whether stabilizing the tetrameric state affects channel gating kinetics independent of ADPR hydrolysis?

Minor points:

1. The authors should elaborate in the introduction and discussion sections why TRPM2 is physiologically important, and they should be clearer that the human TRPM2 is not a chanzyme. What are the relevant endogenous agonists and regulators of TRPM2 and what is known about how they bind the channel? Why would vertebrate TRPM2 not need to self-regulate to ADPR like early species channels?
2. The authors should include an explanation for why they chose specific conditions to solve their distinct structures. The conditions are listed in the methods, but an explanation for conditions and incubation timescales are not clearly indicated.
3. In Fig. 2C and D, the authors compare ADPR binding to the NUDT9-H domain between srTRPM2 and human TRPM2; however, the PDB accession code for the human structure use is not included in the figure legend, main text, methods, or data availability sections.
4. Fig. S9D should be swapped for the representative single channel recordings in Fig. 2G since these data better illustrate differences between WT and mutant channels.
5. Structural callouts in Fig 2D and E should be made clearer to reflect important features. Why is E1062 included since it does not appear to coordinate the divalent cation? Changes to ADPR binding mode between srTRPM2 and human TRPM2 are used to illustrate how human TRPM2 has lost enzymatic activity, however, it is difficult to see how the alpha and beta phosphates are differentially coordinated in the current structural snapshots. It is also difficult to see the catalytic water molecule since it is a similar red color to the protein backbone.
6. In the figure legend title for Fig. 4, the authors should indicate ADPR drives "domain tetramerization". Additionally, clarify what the mutations are for Fig. 4E.
7. In Fig. 6, the authors should compare the human TRPM2 gating ring to the srTRPM2 NUDT9-H domain tetramerization structure. Does the buckle helix partially recapitulate the gating ring?
8. The discussion section would benefit from a mechanistic model to explain how the authors think ADPR binding, domain tetramerization, ADPR hydrolysis, and channel gating are integrated. How does this cycle affect divalent cation levels?

Author Rebuttal to Initial comments

Reviewer #1:

Remarks to the Author:

Huang et al. provides an insightful view on TRPM2, from structural and evolutionary perspectives. Structural biology of membrane proteins is key, and regarding TRPM channels, it is special because of the dual mechanism of action combining ion channel and enzymatic activities. In this case, this manuscript represents an excellent piece of structural biology work, to comprehend a complex functional mechanism. In fact the manuscript is elegantly prepared and the data strongly supports the conclusions. Certain aspects need clarification and/or further

discussion, but the manuscript could be published in present form in NSMB. Specifics that could be addressed to benefit the manuscript in the present form are:

1. The structures show 3 well-defined cholesterol densities per subunit close to the conducting pore region. These densities are present in all 14 structures, suggesting a structural or regulatory role on the channel. This is an interesting observation, however the author do not discuss further on the relevance of this lipid in the mentioned site. In addition, since the structural biology workflow is exhaustive, the presence of differential lipid densities (PIP2, PCs, etc.) in the different maps could be analyzed further in order to define lipid regulatory aspects, if any, on TRPM2.

Response: We appreciate the reviewer's comments and acknowledge the potential significance of cholesterol and other lipids in regulating the channel function of the TRPM2. The three cholesterol-like densities near the ion-conducting pore are well-defined, which allows us to confidently identify them as cholesterol through de novo analysis. Other lipid-like densities, while visible, are less well-defined and thus require further investigation for identification. We agree that these lipid molecules, particularly the cholesterol molecules, may play essential roles in regulating the channel function of TRPM2. However, as this manuscript is specifically focused on and also heavily packed by the coupling mechanism between the enzyme and channel functions, and the structural and functional evolution of the NUDT9-H domain, we hope to address the lipid-dependent channel regulation of TRPM2 in future work.

2. Regarding the novel cation binding site characterization, the changes in channel opening probability and open rate for both mutants E1114A and E1114Q are not significant (Figure S9D). This should be clearly stated in the manuscript, focusing on that the only significant difference is in the channel closed state.

Response: Thanks for this suggestion. To provide a more reliable kinetic analysis, we have increased the number of single channel measurements to 6-7. In the revised manuscript, we have focused on the E1114A mutant (and therefore removed the electrophysiological data for the E1114Q mutant), as we also have structural data confirming this mutant indeed abolished cation binding at this site. Our new data analysis for the E1114A mutant showed a small decrease in channel open probability and a small increase in channel close rate compared to the wild type. We have revised the manuscript accordingly in lines 146-152.

3. Lines 200-203. The authors could discuss further (briefly) what changes, and the importance of these changes, are relevant to avoid hydrolysis of ADPR, as it is done later for the differences in ADPR binding between srTRPM2 and hsTRPM2 in lines 315-318.

Response: We have added a brief discussion in the revised manuscript, lines 195 to 198.

4. Figure 4E seems to show a more significant effect on the channel gating for BH2A and Δ BH mutants of the buckle helix than for the Δ NUDT9-H mutant. The authors suggest that the buckle helix stabilizes the NUDT9-H tetramerization. It seems that this helix might have a more central role in the channel gating regulating, according to these results. Moreover, it is said the buckle helix mutants are expected to weaken the tetramerization, which might account for the long-lived closed states observed in Figure 4, but more concise data analysis should be given to support this claim, since it is quite an important issue, and the number of structures in the present manuscript should suffice to build a stronger argument regarding conformational changes vs. tetramerization of this domain.

Response: We thank the reviewer's comment regarding the role of buckle helix. Following the reviewer #2's suggestion, we have performed additional single-channel recordings for Δ NUDT9-H, BH2A and Δ BH mutants, increasing the number of independent replicates to 6-7. Analysis of the new data revealed that these mutants have similar gating kinetics, albeit with variable open rate (Extended Data Fig. 12d). Moreover, all three mutants exhibited long-lived closed states between bursts of channel opening, which are not observed in the wild-type protein (Figure 1a, 4d). Therefore, our functional data suggest that all three mutants have similar effects on channel gating. However, we do not have sufficient evidence to conclude definitively whether the BH2A and Δ BH mutants have a more pronounced effect than the Δ NUDT9-H mutant. The new single-channel analysis is now included in lines 222-238 and 252-252. We have also discussed a possible mechanism, based on structural analysis, of how Buckle helix-mediated NUDT9-H tetramerization modulates channel activity in lines 242-261.

5. Line 1014. Figure 9 should be Figure S9

Response: Corrected and we thank the reviewer for pointing out this typo.

Reviewer #2:

Remarks to the Author:

This is an interesting structural and functional study aimed at understanding how divalent ions and ADPR regulate the activity of TRPM2 channels. Two ADPR molecules bind to distinct sites on the channel, where one appears to be required for channel activation and the other can have an enzymatic action wherein it hydrolyzes ADPR to AMP and R5P. The authors focus on an ancient version of TRPM2 from choanoflagellates that has this enzymatic function to solve structures under 11 different conditions to understand the regulatory interplay between these various sites. Although I am inclined to believe the conclusions advanced by the authors, the presentation is quite dense and will be hard for many to follow unless they work on TRPM2

channels. The following are critically important points the authors should address in revision.

1) If I understand correctly, the authors have solved structures using grids that were prepared using 11 different conditions, all of which were refined using different programs or symmetry constraints and in some cases using different populations of particles. Table S1 is quite confusing in places and needs to be better organized to be readily comprehensible to readers. I suggest using vertical lines to distinguish same conditions or unique conformations. This will help for all four pages of the table but is essential for the last two. Page 3 is almost incomprehensible. The description of what was done should also be expanded in the main text. As it stands now the reader needs to dig through the table to understand and even that is challenging. It would also be helpful if the authors could indicate which refinement was used for different purposes, including formulating different conclusions and making different figures.

Response: We appreciate the reviewer's comments. Following the reviewer's suggestion, we have revised Extended Data Table 1 to make it more organized. Additionally, in the Methods section, we have now added lines 903 to 912 to include more information on the NUDT9-H domain-focused data processing.

2) For all of the key structural findings the authors need to prepare figures using not just cryo-EM density for the ion or ligand, but also for the surrounding protein so that readers can readily judge the quality of the maps and conclusions. As one example, Fig 3B and D look reasonable when only ion and ligand density are shown (although the assignment of water is highly questionable!!), but I am not so sure that will be true once density is included for the surrounding protein given the modest resolution for most of the maps. Showing equivalent views for important control structures like apo would also help to give a much clearer impression of the quality of maps and conclusions. I view this as a systematic issue that needs to be addressed and the authors should find ways to prepare figures that will convince the most skeptical of readers. I am willing to give the authors the benefit of the doubt at this stage, but I shouldn't have to when this is something that can be addressed with a more rigorous presentation.

Response: We thank the reviewer for the suggestions. We have added additional figures (Fig. S5C) showing the densities of the ligands and the surrounding residues within the NUDT9-H domain, using the same contour levels as in Fig 3. At these levels, the densities of the surrounding residues are properly displayed and are comparable to the ligand densities. It is also noteworthy that the local resolution of the NUDT9-H domain is around 2.8 Å, which is sufficient to resolve ligands such as ADPR or its hydrolysis products. We hope this new information addresses the reviewer's concern about the quality of the ligand densities. We agree with the reviewer that the putative water density in Fig 3B is indeed weak, and we

assigned the density as a water molecule based on the crystal structure of human NUDT5 (PDB ID: 2DSC, doi:10.1016/j.jmb.2008.04.006), which shares a similar catalytic center with the NUDT9-H domain. Nevertheless, we have followed the reviewer's suggestion to qualify this assignment by clearly stating "a putative" water in the main text and figure legend. We are unable to provide an equivalent view of the apo structure because, as discussed in the manuscript, the NUDT9-H domain is a semi-independent domain without of ADPR and is therefore of low resolution due to high flexibility. The models for the NUDT9-H domain in the ADPR-free states were generated by rigid-body fitting of an Alphafold model into the cryo-EM maps.

3) In many cases (Fig 1C,2G,4E) the authors show what appear to be single channel records from one recording without any quantification or presentation of population data. These need to be expanded to include more information on how recordings were obtained and for how long, whether any evidence was seen for multiple channels in a patch, all points histograms to clearly present the underlying complexity and then population data to demonstrate reproducibility. In at least one place the authors make statements about opening and closing kinetics without any presentation of data analysis, how this was determined and how many times it was replicated.

Response: We appreciate this suggestion. All single-channel measurements analyzed were from recordings with only one channel active, and the duration of the measurements is indicated in the figure legends. To strengthen our analysis, we have performed additional single-channel recordings, increasing the number of independent replicates to 6-7. We have added Extended Data Figs. 14-15 to provide a detailed analysis of single-channel kinetics, including the identification of single channel activity, amplitude histograms, and opening and closing kinetics. We have also modified the figures related to single-channel data to reflect updated statistics resulting from the increased number of replicates.

4) I think the authors need to say more about the role of the NUDT9-H domain in regulating channel activation. Including more carefully analyzed data will help, but I can tell whether ADPR binding to that domain has much of an impact when the NUDT9-H domain is deleted. Is that domain really just there to lower the local concentration of ADPR or does it really influence channel activity. At least to me this wasn't clear from the current presentation.

Response: We apologize for not explaining the dual roles of the NUDT9-H domain clearly in the previous version of the manuscript. We have provided both structural and functional evidence demonstrating that the NUDT9-H domain not only indirectly regulates channel gating by lowering the local ADPR concentration, but also directly affects channel gating by inducing conformational changes in the channel module when the NUDT9-H domains tetramerize upon ADPR binding. In the revised the manuscript, we have updated the electrophysical data related

to the NUDT9-H domain (Fig. 4e) and made revisions to the text to better depict the direct role of the NUDT9-H domain in regulating channel function (see section "The srNUDT9-H modulates channel activity via tetramerization").

Reviewer #3:

Remarks to the Author:

In this manuscript, Huang et al use time-resolved cryo-EM to capture 14 structures of a choanoflagellate TRPM2-like channel enzyme in distinct gating and ADPR hydrolysis catalytic states. Some of their structural insights are tested by inside-out patch single channel recordings of wild type and mutant srTRPM2 channels. The authors combine these results to propose a binary model wherein ADPR and cation binding to the MHR1/2 and NUDT9-H domains induce enzymatic domain tetramerization and channel gating while slow ADPR hydrolysis by the NUDT9-H domain reduces local ADPR concentration to close the channel. Finally, the authors compare their srTRPM2 structures to their previously solved human TRPM2 structures to propose an evolution of the NUDT9-H domain from a weak ADPR hydrolase in early species to an enzymatically dead, critical gating module in higher organisms. Despite the impressive number of structures, the authors currently overinterpret their structural data to propose gating and regulatory mechanisms that are not supported by and are poorly integrated with their functional and enzymatic work. For example, the authors claim throughout the manuscript that ADPR binding to and hydrolysis by the NUDT9-H allow rapid switching between the open and closed states to avoid divalent cation overload. However, the authors fail to show whether and how ligand binding to the NUDT9-H domain or substrate hydrolysis accompanies gating-relevant structural transitions. They also fail to correlate their calculated k_{cat} values (seconds) to gating kinetics (milliseconds) to demonstrate that ADPR hydrolysis occurs on a timescale relevant to channel closing. Moreover, the manuscript is currently inaccessible to non-TRPM2 experts as the introduction and discussion sections assume readers are readily familiar with the role and regulation of TRPM2 in vertebrates. In its current form, this manuscript and its findings are not suitable for publication in Nature Structural & Molecular Biology. To strengthen their conclusions, this reviewer recommends the authors address the following major points:

1. The timescales of functional recordings and structural snapshots are poorly integrated. The single channel recordings provide acute effects of ADPR, cations, and mutations on channel function whereas the effects of ADPR hydrolysis by the NUDT9-H domain on channel function inferred from structures will occur on longer timescales. Thus, the authors should include longer recordings where ADPR can be depleted by the NUDT9-H domain. Such recordings would be best conducted in cell-attached or whole cell configuration with endogenous induction of ADPR production versus exogenous application of ADPR and cations to patches. These recordings

would also allow the authors to demonstrate whether and how mutations or deletions of the NUDT9-H domain and its tetramerization affect channel activity (to be included in Fig. 4). Does the NUDT9-H domain-mediated ADPR hydrolysis promote channel closing on a physiologically relevant timeframe that would, indeed, prevent divalent cation overload? Does disruption to ADPR hydrolysis result in prolonged channel activation?

Response: We agree with the reviewer that it would be insightful to address the correlation between our calculated k_{cat} value and gating kinetics under physiological conditions.

However, performing the proposed longer recordings in cell-attached or whole-cell configurations with endogenous induction of ADPR production poses significant challenges. This is because in our system, srTRPM2 is recombinantly expressed using human cell lines, and in these cells, the interplay of multiple enzymes and cytosolic factors that would influence ADPR metabolism and TRPM2 channel gating is complex and not fully understood. Moreover, there is a lack of tools to reliably control/measure the level of srTRPM2 expression and ADPR production in the human cell lines. In fact, we do not know specific data on the copy number of srTRPM2 and the concentration of ADPR in *Salpingoeca rosetta*. The relationship between these two parameters is absolutely critical to assess if the rate of ADPR hydrolysis by srTRPM2 would align with the timescales for channel closing under “real” physiological settings. We propose that further research, possibly involving direct measurements in *Salpingoeca rosetta*, would be needed to resolve this question.

In response to the reviewer’s suggestion, we spent a substantial amount of effort attempting whole-cell recordings with exogenous application of ADPR through the pipette solution. The ADPR concentrations in the pipette solution ranged from 0.01 to 100 μM in 10-fold increments. These concentrations were chosen to establish a broad range of gradients for ADPR diffusion into the cell. Theoretically, a higher concentration in the pipette should facilitate a quicker equilibration to a target intracellular concentration (e.g. activation threshold). Interestingly, we never observed ADPR-activated currents, which suggests a faster hydrolysis of ADPR by srTRPM2 than its diffusion into the cell under all tested ADPR concentrations. Of note, we consistently observed channel activation in the inside-out configuration using cells from the same batches. This result further questions the feasibility of inducing endogenous ADPR production to activate the channel in human cell lines, and supports that the physiological relevance of ADPR hydrolysis is dependent on the specific copy number of srTRPM2 on the cell membrane and the intracellular concentrations of ADPR.

In summary, our manuscript provides substantial insights into the coupling mechanisms between the channel and enzyme functions of srTRPM2 and the molecular evolution of the NUDT9-H domain. The additional experiments suggested by the reviewer, while of great value, represent a substantial and separate body of work beyond our current biophysical focus.

2. Relatedly, if ADPR depletion by the NUDT9-H domain hydrolysis sets channel closure, then a TRPM2 “chanzyme” with faster hydrolysis kinetics would be expected to exhibit enhanced gating kinetics than srTRPM2. The authors should compare the gating kinetics of srTRPM2 to those of TRPM2 orthologues with faster ADPR hydrolysis kinetics.

Response: As addressed in the response to the first comment, the physiological correlation between the hydrolysis kinetics and gating kinetics depends on the relationship between the copy number of the TRPM2 chanzyme on the cell membrane and the intracellular concentration of ADPR. Therefore, a TRPM2 “chanzyme” with faster hydrolysis kinetics does not necessarily exhibit enhanced gating kinetics than srTRPM2. Nevertheless, we fully agree with the reviewer that it would be insightful to further explore the unique TRPM2 chanzyme system, which is still understudied. We are indeed interested in characterizing additional TRPM2 chanzymes with different biophysical properties in future work, using the structural and biophysical approaches established in our current study.

3. The single channel recordings presented throughout the manuscript should include baseline currents before agonists and cations were applied to show the quality of the recordings. Additionally, histograms to show open probability and dwell times of their replicates should be included. Moreover, the authors should justify why only 3 replicates were performed for patching experiments; literature precedence is 6-8. P0 values should be included for all single channel recordings (currently they are only included in Fig. 1C). Finally, a representative recording or the histogram profiles for WT srTRPM2 channels with Ca/ADPR should be included in Fig. 2G and 4E to allow for better comparison of changes to activity with mutant channels.

Response: We appreciate the reviewer's suggestions. We have performed additional single-channel recordings to increase the number of independent replicates to 6-7. We have carefully revised the single-channel related figures and methods based on the reviewer's feedback to ensure clarity and accuracy in presenting the results. Specifically, we have included the P_0 values, baseline and histograms. Due to space limitations, we have chosen to present the detailed analysis in Extended Data Figs. 14-15.

4. Their model in Fig. 3A and corresponding hypothesis proposes that reduction of ADPR levels by ADPR hydrolysis leads to ADPR departure from the MHR1/2 sites as opposed to the NUDT9-H domain extracting ADPR from the MHR1/2 site. The authors should explain how is this consistent with srTRPM2 being a chanzyme as opposed to a canonical ligand-gated ion channel wherein agonist departure returns the channel to a closed state (their leading sentences in the abstract).

Response: We appreciate this comment. Given the significant spatial separation (~40-50 Å) and the lack of a closed pathway between the ADPR binding sites in the MHR1/2 and NUDT9-H domains, a direct handover of ADPR between these sites is unlikely. Instead, we believe that ADPR binding to each site is independent and in dynamic equilibrium with the environmental ADPR pool. The purpose of Fig 3A is to illustrate the key steps involved in the hydrolysis cycle that were captured in our structural analysis. To clarify this following the reviewer's suggestion, we have revised the legend of Fig. 3a to include the statement: Note that ADPR binding to the N- and C-terminal sites occurs independently, and each is in a dynamic equilibrium with the environmental ADPR pool.

5. The model in Fig. 3A assumes that the MHR1/2 site has a higher affinity for ADPR than the NUDT9-H domain since the NUDT9-H can be devoid of substrate or products while the MHR1/2 site retains substrate binding. The authors should determine the binding affinity for MHR1/2 and the NUDT9-H domains for ADPR to qualify their model.

Response: As noted in the response to comment 4, the models presented in Figure 3A are not intended to suggest that MHR1/2 has a higher affinity for ADPR. Instead, they serve to illustrate the key structural snapshots during ADPR hydrolysis. We have revised the legend of Fig. 3A for clarification.

6. The authors currently provide an understated explanation for how ADPR binding to MHR1/2 and NUDT9-H lead to channel gating despite having 14 distinct structures. How is ligand binding transmitted to the pore? Include structural overlays to show conformational changes associated with ligand binding from the apo to ligand bound states (apo versus Mg/ADPR-bound, closed) and conformational changes from the closed to open state (Mg/ADPR-bound closed versus Mg/ADPR-bound open). How does enzymatic activity return the channel to the closed state (Intermediate state I or III versus apo state)?

Response: This is a great suggestion! We have now provided a movie to illustrate the remarkable conformational changes induced by ligand binding, ultimately resulting in channel opening. In the previous version of the manuscript, we intentionally limited our discussion on this aspect to maintain a focus on the interplay between the enzymatic and channel functions of srTRPM2, as well as the structural evolution of the enzyme domain. Recognizing the importance of visualizing the gating mechanism, we believe that the new movie effectively complements this aspect.

7. The structural landscape presented in Fig. 1D is confusing and subtle conformational changes are difficult to discern in the current presentation. This would benefit from being updated to a model as in Fig. 3A and/or by highlighting which structural features change between

neighboring structures. This could be even better strengthened by indicating which structures are compared in subsequent figures and subsections. Additionally, the authors should be clear which structures are used in all figures.

Response: Thanks for this constructive comment! We recognize that the conformational differences between the structures in Fig. 1D were not effectively conveyed. To rectify this, we have enlarged the labels to more clearly highlight the conformational states, and removed trivial labels to make the presentation less crowded. Additionally, we have clarified which specific structures are used in each figure throughout the manuscript.

8. The crosslinking data in Fig. 4D needs molecular weight markers to demonstrate confidence that the higher band is the tetramer. Their srTRPM2 construct has at least 1414 residues and a GFP tag, thus a monomer should be ~170 kDa. A tetramer would be 680 kDa, which is close to the resolution limit by SDS-PAGE. Most common molecular weight ladders have the highest standard of 250 kDa. How many times was this experiment repeated? Did the cysteine pairs crosslink without ADPR and calcium addition? Could the authors use these cysteine pair mutants for functional studies to ask whether stabilizing the tetrameric state affects channel gating kinetics independent of ADPR hydrolysis?

Response: For the crosslinking experiments in Fig. 4D, we utilized the HiMark protein standard from Invitrogen, which includes a marker at 460 kDa, therefore suitable for detecting proteins with very high molecular weights. We have included the protein ladder in a revised figure. It is important to note the membrane proteins usually migrate faster than the theoretical molecular weight, which means the tetrameric srTRPM2 would appear smaller than 680 kDa. The band of the cross-linked srTRPM2 was higher than the 460 kDa marker, indicating it most likely corresponds to the srTRPM2 tetramer. However, we recognize the inherent limitations of SDS-PAGE in precisely determining molecular weight and wish to emphasize that the main objective of this experiment was to demonstrate protein crosslinking when cysteine pairs—designed based on our structural model—were introduced, thereby supporting our structural model. Our result unambiguously demonstrated protein crosslinking. This experiment was conducted multiple times, yielding consistent results. In the absence of ADPR and Ca, we noted a distinctly weaker tetramer band. The presence of this band even in the absence of ligand suggests a basal level of tetramerization that may be attributed to the principles of conformational selection and population shift, where the protein samples different conformational states, including those favoring tetramerization, even in the absence of ligands. This basal tetramerization could also be due to endogenous levels of ADPR, Mg, and Ca present in the whole-cell lysates.

Our electrophysiological data on the mutants affecting NUDT9-H tetramerization have provided essential insights into the regulatory role of the NUDT9-H domain in srTRPM2 gating. While we agree with the reviewer that investigating the effects of stabilizing tetrameric states through

cysteine pair mutations on channel gating kinetics, independent of ADPR hydrolysis, would be an interesting biophysical study, we hope to address this along with many other intriguing questions that may arise from our findings, in future studies.

Minor points:

1. The authors should elaborate in the introduction and discussion sections why TRPM2 is physiologically important, and they should be clearer that the human TRPM2 is not a channel-enzyme. What are the relevant endogenous agonists and regulators of TRPM2 and what is known about how they bind the channel? Why would vertebrate TRPM2 not need to self-regulate to ADPR like early species channels?

Response: Thanks for the suggestion. We have revised the introduction to clearly indicate that human TRPM2 is not a channel-enzyme, along with relevant literature references (lines 68 to 70).

We have discussed the most well-established endogenous agonist of TRPM2, ADPR, in the introduction, as it is directly related to the dual functions of srTRPM2, which is the main objective of this work. We have intentionally focused on this key agonist, as an exhaustive review of all known endogenous agonists and regulators, along with their binding mechanisms, would detract from the main objective of this paper and would be better suited to a dedicated literature review, especially considering space constraints.

We have also added speculation as to why vertebrate TRPM2 would not need to self-regulate to ADPR like early species channels (lines 386 to 398). Perhaps because vertebrate TRPM2 has evolved a rapid desensitization mechanism (Iordanov et al, 2019). Therefore, it is no longer necessary to degrade ADPR to close the channel.

2. The authors should include an explanation for why they chose specific conditions to solve their distinct structures. The conditions are listed in the methods, but an explanation for conditions and incubation timescales are not clearly indicated.

Response: We have included the relevant details in the Method section from lines 842 to 856.

3. In Fig. 2C and D, the authors compare ADPR binding to the NUDT9-H domain between srTRPM2 and human TRPM2; however, the PDB accession code for the human structure use is not included in the figure legend, main text, methods, or data availability sections.

Response: We have added the PDB ID for the human TRPM2 in complex with Ca²⁺ and ADPR.

4. Fig. S9D should be swapped for the representative single channel recordings in Fig. 2G since these data better illustrate differences between WT and mutant channels.

Response: We thank and agree with the review's comment and have updated Fig. 2 accordingly.

5. Structural callouts in Fig 2D and E should be made clearer to reflect important features. Why is E1062 included since it does not appear to coordinate the divalent cation? Changes to ADPR binding mode between srTRPM2 and human TRPM2 are used to illustrate how human TRPM2 has lost enzymatic activity, however, it is difficult to see how the alpha and beta phosphates are differentially coordinated in the current structural snapshots. It is also difficult to see the catalytic water molecule since it is a similar red color to the protein backbone.

Response: Thanks for this valuable feedback. E1062 is located in the TRP helix and is conserved across Ca²⁺-sensitive TRPM2, 4, 5, and 8 channels. In our lab's previous study of the TRPM5 channel, we observed that this glutamate residue interacts with Ca via a water molecule and plays an essential role in TRPM5 channel gating (Ruan *et al*, 2021). We included this residue to highlight its conservation within the Ca²⁺-sensitive TRPM channels. We have added black lines to indicate the interaction of alpha and beta phosphates with neighboring residues in the two figure panels. This was not done in the previous version as we felt that these two figure panels were overcrowded. We have also changed the color of the water molecules to better distinguish them from the protein backbones.

6. In the figure legend title for Fig. 4, the authors should indicate ADPR drives "domain tetramerization". Additionally, clarify what the mutations are for Fig. 4E.

Response: We have revised the title for Fig. 4 and added information about the mutants in Fig. 4e.

7. In Fig. 6, the authors should compare the human TRPM2 gating ring to the srTRPM2 NUDT9-H domain tetramerization structure. Does the buckle helix partially recapitulate the gating ring?

Response: In human TRPM2, the NUDT9-H domains and MHR1/2 domains of all four subunits interact extensively with each other, forming an integrated gating ring in both apo and ligand-bound states; both the cap and core regions of the NUDT9-H domain are engaged in intersubunit interactions. On the other hand, in srTRPM2, the NUDT9-H domain remains a semi-independent domain regardless of ligand binding, as the cap region of the NUDT9-H domain does not participate in any interaction. Therefore, while the buckle helix in srTRPM2 may appear to recapitulate the gating ring in human TRPM2 to some extent, their roles in

channel gating are not equivalent. We also believe that the fundamental difference in how the NUDT9-H domain is integrated with the channel module in srTRPM2 and human TRPM2 can be better visualized using the apo states due to their obvious difference.

8. The discussion section would benefit from a mechanistic model to explain how the authors think ADPR binding, domain tetramerization, ADPR hydrolysis, and channel gating are integrated. How does this cycle affect divalent cation levels?

Responses: Thanks for the suggestion. We have briefly discussed the integration of ADPR-induced channel gating and ADPR hydrolysis within the unique srTRPM2 channel-enzyme in the first paragraph of the Discussion section. We refrained from speculating on how the enzyme-gating cycle might impact intracellular divalent cation levels, as the physiological roles of TRPM2 chanzyme and ADPR in the choanoflagellate *Salpingoeca rosetta* are not well understood at this time.

Decision Letter, first revision:

Message: Our ref: NSMB-A47635A

22nd Dec 2023

Dear Dr. Du,

Thank you for submitting your revised manuscript "Coupling enzymatic activity and gating in an ancient TRPM chanzyme and its molecular evolution" (NSMB-A47635A). It has now been seen by the original referees and their comments are below. The reviewers find that the paper has improved in revision, and therefore we'll be happy in principle to publish it in Nature Structural & Molecular Biology, pending minor revisions to satisfy the referees' final requests and to comply with our editorial and formatting guidelines.

To facilitate our work at this stage, it is important that we have a copy of the main text as a word file. If you could please send along a word version of this file as soon as possible, we would greatly appreciate it; please make sure to copy the NSMB account (cc'ed above).

Sincerely,

Carolina Perdigoto, on behalf of

Katarzyna Ciazynska, PhD
(she/her)
Associate Editor
Nature Structural & Molecular Biology
<https://orcid.org/0000-0002-9899-2428>

Reviewer #1 (Remarks to the Author):

The authors have addressed all points raised, the manuscript is elegant and solid and my recommendation is publication in NSMB in the present form.

Reviewer #2 (Remarks to the Author):

I read the revised manuscript and response to reviewers and in my view the authors have done a really nice job of revising the manuscript to address the concerns raised by each of the reviewers.

I have a few minor remaining points and one more potentially substantive point that may influence how the current findings are contextualized for the reader.

Minor:

- 1) The TLC results shown in Fig.2f concerning Ca inhibition of ADPR hydrolysis are n=1. While it may be challenging to document population data in these experiments, could the authors at least state how many times this experiment was repeated, and a similar inhibition of hydrolysis was observed?
- 2) I have a similar comment for the crosslinking results shown in Fig.4d. How many times was a similar result obtained?
- 3) For the data plotted in Ext Fig. 6c, it would be easier to see that there are two overlapping datapoints if the green triangles are shown on top of the purple squares.

Potentially substantive:

The ability of Ca to inhibit ADPR hydrolysis seems to me to be quite fundamental to understanding the physiological context/role of enzymatic activity. My knowledge of choanoflagellates is not vast, but I presume that intracellular Mg is in the mM range and stable while Ca is low except when it permeates the plasma membrane or is released from stores. I have no idea how or when ADPR changes. When ADPR ribose is present at sufficient levels to activate TRPM2, intracellular Mg could support activation of the channel and enzymatic activity, giving rise to transient activation. However, as Ca permeates the channel and intracellular concentrations around the channel rise, it would inhibit enzymatic activity and thereby sustain persistent activation. If choanoflagellates are anything like a mammalian cell, the properties you describe would be expected to give rise

to potentially quite complex activity as a function of changes in intracellular Ca and ADPR. It would seem that framing the enzymatic activity as responsible for preventing Mg and Ca overload is somewhat off base. What am I missing here? I also think the finding that Ca inhibits hydrolysis while Mg supports it, and that these can be rationalized by the structures solve, should be elevated in how they are presented.

While some of the additional experiments suggested by reviewer 3 are indeed interesting and would be ways of further understanding how the hydrolase regulates channel activity in a physiological setting, they represent entirely distinct lines of investigation that will be extremely challenging, with many potential problems that would need to be solved.

The authors have already provided extensive evidence, including time-resolved cryo-EM, single channel and macroscopic current recordings and ADPR hydrolysis experiments, to support reasonable core conclusions. To my thinking this study is almost ready for publication, and in my view, it represents nothing less than a tour-de-force. The authors should be proud of their achievement, and they are to be congratulated.

Reviewer #3 (Remarks to the Author):

In this revised manuscript, Huang et al addressed reviewer comments from the first round of reviews. This includes increased numbers of replicates for single channel recordings accompanied by data analysis, as well as some improvement in structural data and story presentation. However, this reviewer is still unclear on the importance of ADPR hydrolysis to channel regulation, which affects the impact or significance of the current findings.

The authors suggest that their collective data support ADPR binding to the NUDT9-H domain promotes channel activity (via distinct conformational changes, albeit perhaps somewhat analogous to the gating ring in human TRPM2) and that ADPR hydrolysis affects local ADPR concentrations, which closes the channel. Indeed, in the abstract the authors say this allows the channel to "alternate rapidly" between open and closed states. What is the functional evidence that ADPR hydrolysis by the NUDT9-H domain influences channel gating/closure? The authors find that truncating the NUDT9-H domain or the buckle helix affected the length of closed states between open bursts, but that the channel properties were otherwise like wild type channels. Moreover, Iordanov et al (eLife 2019) found that mutations to the ancient TRPM2 NUDT9-H domain had no effect on channel activity. The authors should be clear on whether there is functional data to support this regulatory role, or whether this is a hypothesis. If the latter, the language should be clear and consistent throughout the manuscript. Relatedly, the video submitted with the manuscript does not appear to reveal whether and how ADPR hydrolysis resets the channel to the closed state.

Moreover, what is the evidence that rapid gating is necessary to avoid divalent cation overload? In the abstract, the authors state "This coupling enables the ion-conducting pore to alternate rapidly between open and closed states, avoiding Mg²⁺ and Ca²⁺ overload", however in the rebuttal the authors indicated that "We refrained from speculating on how the enzyme-gating cycle might impact intracellular divalent cation levels, as the physiological roles of TRPM2 chanzyme and ADPR in the choanoflagellate *Salpingoeca rosetta* are not well understood at this time." If this is also speculative, then the authors should soften the language in the abstract and throughout the manuscript.

Two other minor notes are:

1. Figure 1 would benefit from demarcations for the TMD, signal transduction layer and ADPR-sensing layer to help orient readers to the relevant sections of the large structures.
2. Figures 2C and 3C seem largely redundant. Are they both necessary?

Author Rebuttal, first revision

Reviewer #1:

Remarks to the Author:

The authors have addressed all points raised, the manuscript is elegant and solid and my recommendation is publication in NSMB in the present form.

Response: We thank the reviewer for the encouraging remarks regarding our manuscript.

Reviewer #2:

Remarks to the Author:

I read the revised manuscript and response to reviewers and in my view the authors have done a really nice job of revising the manuscript to address the concerns raised by each of the reviewers.

Response: We thank the reviewer for the encouraging comments on our revised manuscript and the constructive feedback.

I have a few minor remaining points and one more potentially substantive point that may influence how the current findings are contextualized for the reader.

Minor:

1) *The TLC results shown in Fig.2f concerning Ca inhibition of ADPR hydrolysis are n=1. While it may be challenging to document population data in these experiments, could the authors at least state how many times this experiment was repeated, and a similar inhibition of hydrolysis was observed?*

Response: We performed the TLC analysis for the Ca inhibition effect three times. This is consistent with the observation of Ca²⁺/ADPR in the NUDT9-H domain in our structural studies.

2) *I have a similar comment for the crosslinking results shown in Fig.4d. How many times was a similar result obtained?*

Response: We performed the crosslinking experiments three times and consistent results were observed.

3) *For the data plotted in Ext Fig. 6c, it would be easier to see that there are two overlapping datapoints if the green triangles are shown on top of the purple squares.*

Response: We have revised Extended Fig. 6c according to the reviewer's suggestion.

Potentially substantive:

The ability of Ca to inhibit ADPR hydrolysis seems to me to be quite fundamental to understanding the physiological context/role of enzymatic activity. My knowledge of choanoflagellates is not vast, but I presume that intracellular Mg is in the mM range and stable while Ca is low except when it permeates the plasma membrane or is released from stores. I have no idea how or when ADPR changes. When ADPR ribose is present at sufficient levels to activate TRPM2, intracellular Mg could support activation of the channel and enzymatic activity, giving rise to transient activation. However, as Ca permeates the channel and intracellular concentrations around the channel rise, it would inhibit enzymatic activity and thereby sustain persistent activation. If choanoflagellates are anything like a mammalian cell, the properties you describe would be expected to give rise to potentially quite complex activity as a function of changes in intracellular Ca and ADPR. It would seem that framing the enzymatic activity as responsible for preventing Mg and Ca overload is

somewhat off base. What am I missing here? I also think the finding that Ca inhibits hydrolysis while Mg supports it, and that these can be rationalized by the structures solve, should be elevated in how they are presented.

While some of the additional experiments suggested by reviewer 3 are indeed interesting and would be ways of further understanding how the hydrolase regulates channel activity in a physiological setting, they represent entirely distinct lines of investigation that will be extremely challenging, with many potential problems that would need to be solved.

The authors have already provided extensive evidence, including time-resolved cryo-EM, single channel and macroscopic current recordings and ADPR hydrolysis experiments, to support reasonable core conclusions. To my thinking this study is almost ready for publication, and in my view, it represents nothing less than a tour-de-force. The authors should be proud of their achievement, and they are to be congratulated.

Response: Thank you for your insightful comments and for highlighting the importance of understanding the physiological context of enzyme activity in relation to Ca and ADPR concentrations. Based on our biophysical characterisation, we observed that the presence of sufficient levels of ADPR, together with Mg, leads to the activation of TRPM2. At the same time, the enzyme domain begins to hydrolyse ADPR, albeit at a rate that appears to be slower than that of channel gating, until ADPR levels decrease sufficiently to induce TRPM2 closure. We propose that this enzymatic activity may serve to limit the duration of TRPM2 activation, thereby preventing Mg and Ca overload. The effect of Ca influx during TRPM2 activation, potentially increasing cytosolic Ca to levels high enough to compete with Mg (presumably in mM concentration) and consequently inhibit TRPM2—implying that cytosolic Ca must also reach mM concentrations (see Extended Data Fig. 2f)—remains uncertain. Although we consider this scenario unlikely, we acknowledge that this aspect warrants further investigation. We agree with the reviewer that understanding the exact physiological role of this enzymatic activity, in particular how it is integrated with channel gating, requires further investigation, ideally in its native environment.

Reviewer #3:

Remarks to the Author:

In this revised manuscript, Huang et al addressed reviewer comments from the first round of reviews. This includes increased numbers of replicates for single channel recordings accompanied by data analysis, as well as some improvement in structural data and story presentation. However, this reviewer is still unclear on the importance of ADPR hydrolysis to channel regulation, which affects the impact or significance of the current findings.

The authors suggest that their collective data support ADPR binding to the NUDT9-H domain promotes channel activity (via distinct conformational changes, albeit perhaps somewhat analogous to the gating ring in human TRPM2) and that ADPR hydrolysis affects local ADPR concentrations, which closes the channel. Indeed, in the abstract the authors say this allows the channel to "alternate rapidly" between open and closed states. What is the functional evidence that ADPR hydrolysis by the NUDT9-H domain influences channel gating/closure? The authors find that truncating the NUDT9-H domain or the buckle helix affected the length of closed states between open bursts, but that the channel properties were otherwise like wild type channels. Moreover, Iordanov et al (eLife 2019) found that mutations to the ancient TRPM2 NUDT9-H domain had no effect on channel activity. The authors should be clear on whether there is functional data to support this regulatory role, or whether this is a hypothesis. If the latter, the language should be clear and consistent throughout the manuscript. Relatedly, the video submitted with the manuscript does not appear to reveal whether and how ADPR hydrolysis resets the channel to the closed state.

Response: In our manuscript, we propose two functional roles of the NUDT9-H domain in regulating the channel function of srTRPM2. Firstly, through structural comparison, we demonstrate that the NUDT9-H domain and its binding with ligands, along with the buckling of the helix, promote the rotation of the intracellular domain of srTRPM2 toward the open conformation. This is further supported by the observation of long-lived closed states in single-channel analysis of Δ NUDT9-H and buckle helix mutants. Secondly, we suggest that the hydrolysis of ADPR in NUDT9-H domain indirectly regulates the channel function, based on the premise that lowering ligand concentration will diminish channel activation.

In the paper by Iordanov et al. (eLife, 2019), functional studies of mutations in the NUDT9-H domain were conducted on *Nematostella vectensis* TRPM2, which may have different gating and coupling mechanisms when compared to srTRPM2.

Moreover, what is the evidence that rapid gating is necessary to avoid divalent cation overload? In the abstract, the authors state "This coupling enables the ion-conducting pore to alternate rapidly between open and closed states, avoiding Mg²⁺ and Ca²⁺ overload", however in the rebuttal the authors indicated that "We refrained from speculating on how the enzyme-gating cycle might impact intracellular divalent cation levels, as the physiological roles of TRPM2 chanzyme and ADPR in the choanoflagellate Salpingoeca rosetta are not well understood at this time." If this is also speculative, then the authors should soften the language in the abstract and throughout the manuscript.

Response: We have revised the language to ensure consistency with our responses.

Two other minor notes are:

1. Figure 1 would benefit from demarcations for the TMD, signal transduction layer and ADPR-sensing layer to help orient readers to the relevant sections of the large structures.

Response: We appreciate the reviewer's suggestion. Although we did not mark these layers in Figure 1, we labeled these structural layers in Figure 2a, Extended Figures 6a, 8a,d, and 9b,c.

2. Figures 2C and 3C seem largely redundant. Are they both necessary?

Response: These two figures serve different purposes: Figure 2C illustrates the ligand binding details in the NUDT9-H domain, while Figure 3C illustrates the enzymatic mechanisms of the NUDT9-H domain. Combining all the information into one figure would make it challenging to clearly illustrate these two different types of information. Therefore, we believe it is necessary to keep these two figures separate.

Final Decision Letter:

Message: 12th Apr 2024

Dear Dr. Du,

We are now happy to accept your revised paper "Coupling enzymatic activity and gating in an ancient TRPM channel and its molecular evolution" for publication as an Article in Nature Structural & Molecular Biology.

Your paper will be published online soon after we receive proof corrections and will appear

in print in the next available issue. You can find out your date of online publication by contacting the production team shortly after sending your proof corrections.

Please note that *Nature Structural & Molecular Biology* is a Transformative Journal (TJ). Authors may publish their research with us through the traditional subscription access route or make their paper immediately open access through payment of an article-processing charge (APC). Authors will not be required to make a final decision about access to their article until it has been accepted. Find out more about Transformative Journals

Authors may need to take specific actions to achieve compliance with funder and institutional open access mandates. If your research is supported by a funder that requires immediate open access (e.g. according to Plan S principles) then you should select the gold OA route, and we will direct you to the compliant route where possible. For authors selecting the subscription publication route, the journal's standard licensing terms will need to be accepted, including self-archiving policies. Those licensing terms will supersede any other terms that the author or any third party may assert apply to any

version of the manuscript.

Sincerely,

Katarzyna Ciazynska, PhD
(she/her)
Associate Editor
Nature Structural & Molecular Biology
<https://orcid.org/0000-0002-9899-2428>